# Achieving high-efficiency emission depletion nanoscopy by employing cross relaxation in upconversion nanoparticles

Qiuqiang Zhan [1], Haichun Liu [2], Baoju Wang[1], Qiusheng Wu[1], Rui Pu[1], Chao Zhou[1], Bingru Huang[1], Xingyun Peng[1], Hans Ågren[2] & Sailing He[1,3,4]

Stimulated emission depletion microscopy provides a powerful sub-diffraction imaging modality for life science studies. Conventionally, stimulated emission depletion requires a relatively high light intensity to obtain an adequate depletion efficiency through only light–matter interaction. Here we show efficient emission depletion for a class of lanthanide-doped upconversion nanoparticles with the assistance of interionic cross relaxation, which significantly lowers the laser intensity requirements of optical depletion. We demonstrate two-color super-resolution imaging using upconversion nanoparticles (resolution ~ 66 nm) with a single pair of excitation/depletion beams. In addition, we show super-resolution imaging of immunostained cytoskeleton structures of fixed cells (resolution ~ 82 nm) using upconversion nanoparticles. These achievements provide a new perspective for the development of photoswitchable luminescent probes and will broaden the applications of lanthanide-doped nanoparticles for sub-diffraction microscopic imaging.

[1] Centre for Optical and Electromagnetic Research, South China Academy of Advanced Optoelectronics, South China Normal University, 510006 Guangzhou, P.R. China. [2] Division of Theoretical Chemistry and Biology, Royal Institute of Technology, S-10691 Stockholm, Sweden. [3] Centre for Optical and Electromagnetic Research, State Key Laboratory of Modern Optical Instrumentation, Zhejiang University, 310058 Hangzhou, P.R. China. [4] JORCEP, Department of Electromagnetic Engineering, Royal Institute of Technology, 10044 Stockholm, Sweden. Qiuqiang Zhan, Haichun Liu and Baoju Wang contributed equally to this work. Correspondence and requests for materials should be addressed to Q.Z. (email: zhanqiuqiang@m.scnu.edu.cn) or to S.H. (email: sailing@kth.se)

Luminescent biomarkers, which include fluorescent dyes and proteins[1], semiconductor quantum dots[2], carbon dots[3], aggregation-induced emission molecules[4], and lanthanide-doped upconversion nanoparticles (UCNPs)[5–10], offer powerful tools for optical imaging and life science studies since they make it possible to visualize events and structures that are otherwise invisible. While continuously expanding the toolbox of luminophores, researchers are keen on exploring their photophysical properties to improve the quality and scope of imaging modalities. Exploitation of optical switching to control the transition of luminophores between two optically distinguishable states (on/off) is an exciting and popular way to circumvent the diffraction limit[11–13]. This approach has led to many far-field super-resolution fluorescence microscopy techniques, such as stimulated emission depletion microscopy (STED)[14–16], single-molecule localization nanoscopy (such as PALM/STORM and MINFLUX)[17–19], and saturated structured illumination microscopy[20, 21], which breaks the diffraction limit and makes it possible to image at a spatial resolution of down to a few nanometers. A major category of these nanoscopy methods, including pulse STED and PALM/STORM, provides sub-diffraction resolution essentially by transiently switching the luminophores between the bright (on) and dark (off) states, causing those within the same diffraction range to emit successively, rather than simultaneously[22].

Lanthanide-doped UCNPs have been developed as an important group of luminescent biomarkers during the last decade. UCNPs convert low-intensity near-infrared (NIR) excitation light to shorter-wavelength NIR, visible, and ultraviolet (UV) emission. Such unique luminescence properties of UCNPs enable superior bioimaging without many of the constraints associated with conventional optical biomarkers, including photobleaching, photoblinking, tissue autofluorescence, limited imaging depth, and high light toxicity[23]. However, efficient optical modulation of UCNPs has not been well established, which hinders their use in advanced luminescence imaging techniques, such as STED microscopy. In the few studies on optical modulation of UCNPs, highly efficient optical depletion of upconversion luminescence has not been reported in major groups of UCNPs (NaYF$_4$:Yb$^{3+}$, Er$^{3+}$/Tm$^{3+}$/Ho$^{3+}$) that provide bright luminescence upon NIR excitation, except a demonstration of optical inhibition of upconversion luminescence in low-efficiency YAG:Pr$^{3+}$ nanoparticles emitting toxic UV light under a complicated excitation scheme with limited applications in biomedical areas[24]. The research efforts exploring co-irradiation by additional wavelengths have usually resulted in luminescence enhancement rather than silencing due to the presence of complicated photon recycling pathways among the lanthanide dopants under multi-wavelength excitation[25, 26]. In our previous study on another upconversion system (NaYF$_4$:Yb$^{3+}$, Er$^{3+}$), the first optical depletion of Er$^{3+}$ emission was shown by inducing excited-state absorption (ESA), intrinsically different from the mechanism we propose in the present study, but the efficiency needed improvement for applications[27].

In this work, the cross relaxation between lanthanide ions provides an auxiliary mechanism for the optical depletion

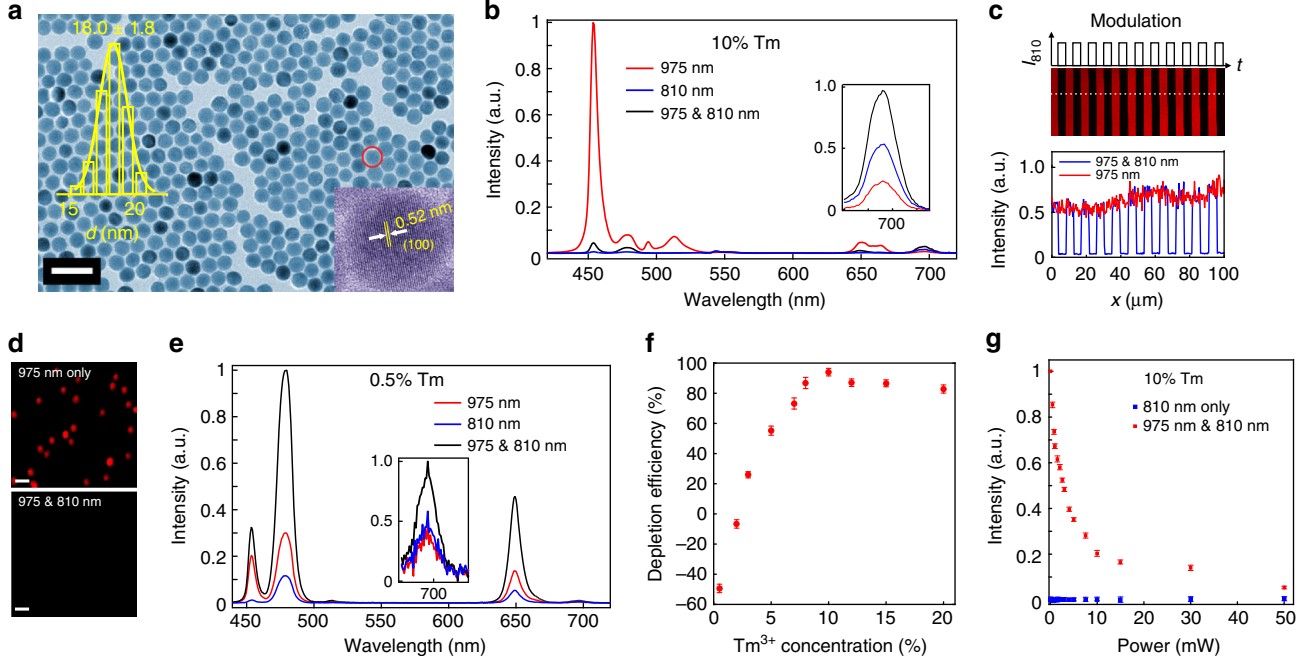

**Fig. 1** Optical depletion of the 455-nm upconversion luminescence of NaYF$_4$:18% Yb$^{3+}$, 10% Tm$^{3+}$ UCNPs. **a** TEM images of NaYF$_4$:18% Yb$^{3+}$, 10% Tm$^{3+}$ UCNPs. Insets: the distribution of the nanoparticle diameters and a high-resolution TEM image, revealing the highly crystalline nature of the nanocrystals. Scale bar: 50 nm. **b** Ninety-six percent depletion efficiency (DE, $(I_{455}^{975} - I_{455}^{975\&810})/I_{455}^{975}$) of the 455-nm emission from NaYF$_4$:18% Yb$^{3+}$, 10% Tm$^{3+}$ ($\lambda_{exc}$ = 975 nm (CW), $\lambda_{depletion}$ = 810 nm (CW)). **c** The upconversion imaging of NaYF$_4$:18% Yb$^{3+}$, 10% Tm$^{3+}$ UCNPs film on slides with modulated 810-nm laser co-irradiation. **d** The upconversion imaging of single or sparse nanoparticles on slides with/without 810-nm laser co-irradiation. Scale bar: 2.5 μm. **e** The emission spectra of NaYF$_4$:18% Yb$^{3+}$, 0.5% Tm$^{3+}$ nanoparticles on glass slides excited by 975-nm CW solely, 810-nm CW solely, 975-nm and 810-nm CW simultaneously. **f** The depletion efficiencies of the 455-nm emission of NaYF$_4$:18% Yb$^{3+}$, x% Tm$^{3+}$ UCNPs with different Tm$^{3+}$ doping concentrations (0.5, 2, 3, 5, 7, 8, 10, 12, 15, and 20%). Negative depletion efficiency represents emission enhancement. The power density of the 975-nm laser is 700 kW cm$^{-2}$, and the power density of 810-nm laser is 17.7 MW cm$^{-2}$. Error bars represent ± 1 s.d. **g** The power-dependent depletion efficiency and the background 455-nm emission intensity excited solely by the 810-nm beam of NaYF$_4$:18% Yb$^{3+}$, 10% Tm$^{3+}$ UCNPs. The saturation intensity $I_{sat}$ (50% off) is estimated to be 849 kW cm$^{-2}$ (about 2.4 mW). Error bars represent ± 1 s.d.

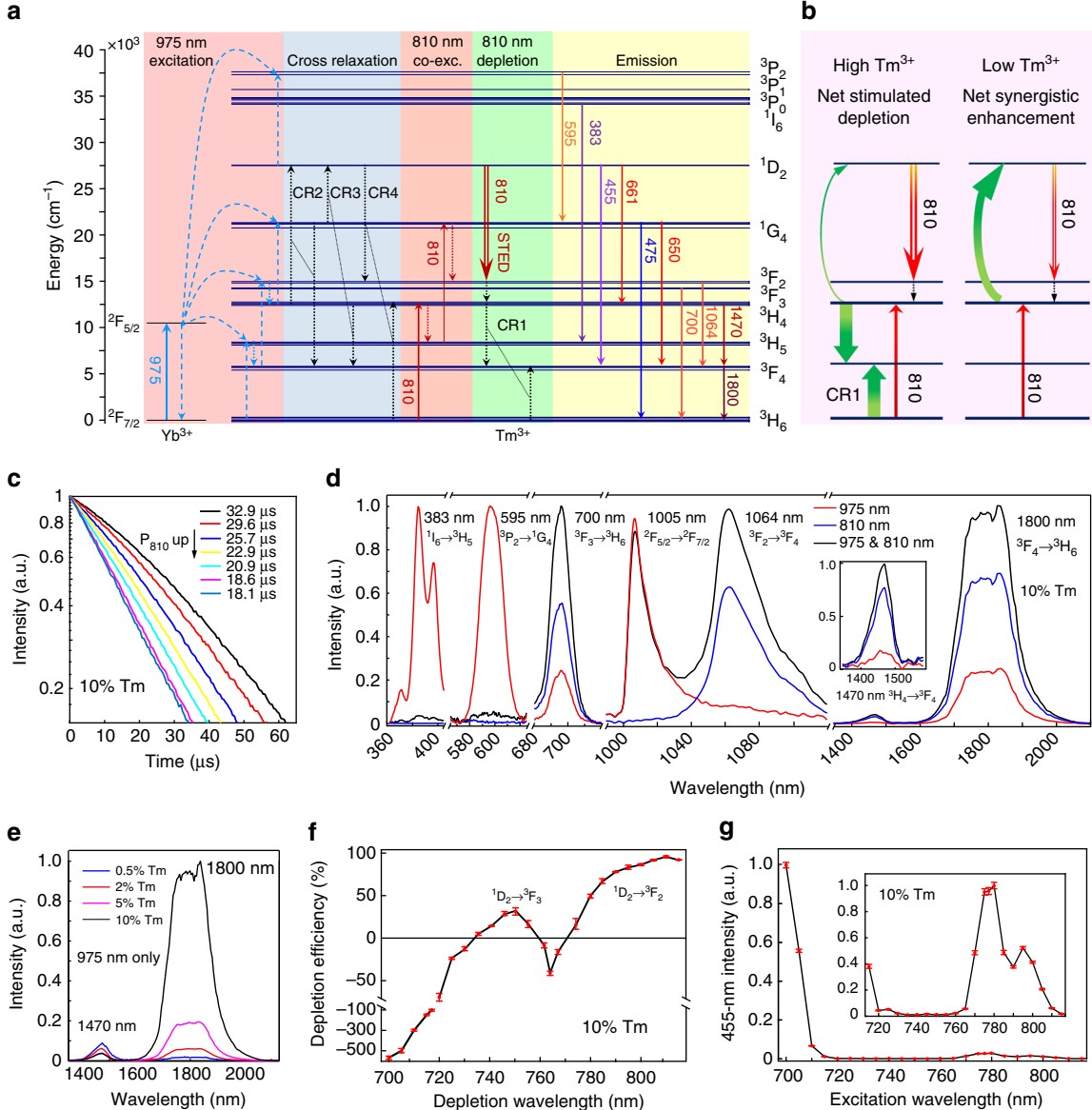

**Fig. 2** Optical depletion mechanism of the 455-nm upconversion luminescence in NaYF$_4$:18% Yb$^{3+}$, 10% Tm$^{3+}$ nanoparticles. **a** Proposed optical emission depletion mechanism of the 455-nm upconversion band of Tm$^{3+}$ of NaYF$_4$:18% Yb$^{3+}$, 10% Tm$^{3+}$ nanoparticles. **b** Schematics of net stimulated depletion (left) and net synergistic enhancement (right) induced by 810-nm irradiation in high and low Tm$^{3+}$-doped UCNPs, respectively. **c** Decay lifetimes of the 455-nm emission of NaYF$_4$:18% Yb$^{3+}$, 10% Tm$^{3+}$ UCNPs at different power densities (Supplementary Table 1) of the depleting 810-nm laser. **d** The intensity change of different emission bands of NaYF$_4$:18% Yb$^{3+}$, 10% Tm$^{3+}$ UCNPs under 975-nm excitation with/without 810-nm irradiation. The spectra were individually normalized in each spectral range, including 360–400, 570–615, 680–715, 980–1140, and 1300–2100 nm, separated by cutoff marks, in order to clearly display the relative intensity changes of different emission bands with and without the 810-nm irradiation. **e** The Tm$^{3+}$ doping concentration-dependent 1470-nm and 1800-nm downconversion emission spectra of NaYF$_4$:18% Yb$^{3+}$, x% Tm$^{3+}$ UCNPs ($\lambda_{exc}$ = 975 nm). **f** Dependence of 455-nm depletion efficiency (DE, $(I_{455}^{975} - I_{455}^{975\&STED})/I_{455}^{975}$)) of NaYF$_4$:18% Yb$^{3+}$, 10% Tm$^{3+}$ on the depletion laser wavelength ($\Delta\lambda$ = 4 nm) ranging from 700 to 815 nm. Negative depletion efficiency represents emission enhancement. The intensity of the 975-nm excitation laser and the depletion laser were kept at 700 kW cm$^{-2}$ and 17.7 MW cm$^{-2}$, respectively. **g** The excitation spectrum for the 455-nm emission of NaYF$_4$:18% Yb$^{3+}$, 10% Tm$^{3+}$ solely excited by the depletion laser with wavelength ranging from 700 to 815 nm; the inset is an enlarged view. Error bars represent ± 1 s.d.

pathway of upconversion luminescence. We establish an efficient optical depletion approach for the blue upconversion luminescence at 455 nm of Tm$^{3+}$ ions in high Tm$^{3+}$-doped NaYF$_4$:Yb$^{3+}$, Tm$^{3+}$ UCNPs with enhanced inter-Tm$^{3+}$ interaction. The almost complete emission silencing is achieved by inducing stimulated emission from the emitting state through a second laser beam, while simultaneously suppressing the synergistic emission enhancement effect of the same laser beam by making full use of the concentration-dependent cross relaxation between the doped

Tm$^{3+}$ ions. This novel emission depletion approach enables us to implement super-resolution luminescence microscopy. We further extend the super-resolution imaging to a two-color mode using the same pair of excitation/depletion beams. In addition, we implement cytoskeleton protein super-resolution imaging of immunolabeled HeLa cells using antibody-conjugated UCNPs and demonstrate the ability of this established UCNP-mediated nanoscopy technique for imaging subcellular structures.

## Results

**Reversible optical depletion of blue emission of Tm-UCNPs.**
The hexagonal-phase NaYF$_4$ nanoparticles used in this study were synthesized using previously reported protocols with modifications (Methods)[28, 29]. Transmission electron microscope micrographs reveal a hexagonal shape of the as-synthesized high Tm$^{3+}$-doped (NaYF$_4$:18% Yb$^{3+}$, 10% Tm$^{3+}$) nanoparticles with an average diameter of 18.0 ($\pm$ 1.8) nm (Fig. 1a). When excited by a 975-nm continuous-wave (CW) laser (700 kW cm$^{-2}$), these nanoparticles produce a dominant blue emission band at 455 nm together with several other relatively weak emission bands at 475 nm, 512 nm, 650 nm, and 700 nm (Fig. 1b, Supplementary Fig. 1), similar to the results in previous reports in high Tm$^{3+}$-doped UCNPs under excitation of comparable intensity[30]. These emission bands can be assigned to the transitions of $^1D_2 \rightarrow {}^3F_4$ (455 nm), $^1G_4 \rightarrow {}^3H_6$ (475 nm), $^1D_2 \rightarrow {}^3H_5$ (512 nm), $^1G_4 \rightarrow {}^3F_4$ (650 nm), and $^3F_3 \rightarrow {}^3H_6$ (700 nm) of Tm$^{3+}$ ions, respectively[30]. Intriguingly, when co-irradiated by an 810-nm CW laser beam ($\Delta\lambda = 4$ nm, 17.7 MW cm$^{-2}$), the most intense 455-nm blue emission is greatly quenched, with a depletion efficiency approaching 96% (Fig. 1b). The emission bands at 475 nm and 650 nm are also depleted, but less significantly (Fig. 1b), while the emission band at around 700 nm ($^3F_3 \rightarrow {}^3H_6$) is distinctly enhanced (Fig. 1b).

A concentrated UCNP dispersion was spin-coated onto a glass slide repeatedly to form a uniform film of nanoparticles and then imaged with a multiphoton laser scanning microscope (Supplementary Fig. 2). Figure 1c shows the multiphoton luminescence images of the UCNP film. Here the intensity of the CW 810-nm laser beam, $I_{810}$, was modulated by a chopper, which resulted in dark lines in the scanned image when the luminescence was periodically inhibited. This emphasizes the degree of optical control over the 455-nm upconversion luminescence as well as its reversible nature. The optical modulation of the 455-nm luminescence for dispersed nanoparticles on a slide was measured. As shown in Fig. 1d, compared to 975-nm single excitation, the 455-nm blue luminescence was almost completely depleted by the co-excitation with 975-nm and 810-nm laser light. Interestingly, the depletion efficiency of the 455-nm upconversion luminescence exhibits strong dependence on the Tm$^{3+}$ concentration (Fig. 1e, f). With low Tm$^{3+}$ doping (0.5%) (Supplementary Figs. 1 and 3), strong enhancement was observed for the 455-nm emission with the addition of the second laser at 810 nm (Fig. 1e). Emission depletion for the 455-nm band requires an adequately high Tm$^{3+}$ concentration (Fig. 1f). For different-sized UCNPs with the same composition (NaYF$_4$:18% Yb$^{3+}$, 10% Tm$^{3+}$), all show efficient depletion of the 455-nm upconversion luminescence ($\lambda_{exc} = 975$ nm) by the 810-nm laser beam, with slightly different saturation intensities (Supplementary Fig. 4). The degree of luminescence inhibition was measured for various values of $I_{810}$ (Fig. 1g). About 4% residual luminescence can be detected for $I_{810} = 17.7$ MW cm$^{-2}$, confirming the effectiveness of the luminescence inhibition process. In our study $I_{sat}$ is estimated to be 849 kW cm$^{-2}$ (power about 2.4 mW). It should be noted that optical depletion efficiency data of upconversion luminescence cannot be well fitted with the depletion equation $\eta = 1/(1 + I_{STED}/I_{sat})$[27, 31], which was derived from a simplified Jablonski diagram of downconversion fluorescence[32, 33]. The inapplicability of this equation to optical depletion of lanthanide upconversion emission is probably due to the fact that many complicated interionic energy transfer processes are not considered in the derivation. However, we still adopt the parameter—saturation intensity ($I_{sat}$) to characterize the emission depletion. The saturation power here is significantly lower than those used in CW-STED microscopy using other biomarkers and those used in the depletion of other lanthanide

ions, typically tens to hundreds of milliwatts[11, 34–37]. Our saturation power is larger than that reported in a similar work about STED microscopy of UCNPs[31], probably due to the difference in the nanoparticles. The 810-nm laser itself can only generate very weak upconversion emission (Fig. 1b). The background luminescence intensity generated by single 810-nm excitation (17.7 MW cm$^{-2}$) is about 2% of that generated by single 975-nm excitation (700 kW cm$^{-2}$) (Fig. 1b).

**Emission depletion mechanism assisted with cross relaxation.**
Our spectroscopic results suggest an emission depletion mechanism involving stimulated emission supported by interionic cross relaxation that suppresses the synergistic enhancement effect of the second laser beam (810 nm). As illustrated in Fig. 2a, the $^1D_2$ emitting state, is populated by the 975-nm excitation beam through a five-photon upconversion process[38–40], and can be depleted via a stimulated emission process induced by the 810-nm beam to populate the $^3F_2$ state, namely, $^1D_2 \xrightarrow{810\,nm} {}^3F_2$. It should be noted that the $^1D_2 \rightarrow {}^3F_2$ transition is very easy to be overlooked and was indeed often neglected in the literature. The $^3F_2$ and $^3F_3$ states were commonly treated as a single state (denoted as $^3F_{2,3}$) while ignoring their conspicuous energy difference since ions at the $^3F_2$ state quickly decays to the $^3F_3$ state nonradiatively, and the transition of $^1D_2 \rightarrow {}^3F_{2,3}$ was usually recognized to generate a single emission band centered at 740–750 nm[30]. Notably the $^1D_2 \rightarrow {}^3F_2$ transition has an even larger branching ratio than the $^1D_2 \rightarrow {}^3F_3$ transition according to previous theoretical studies[41–44]. The $^1D_2 \rightarrow {}^3F_2$ transition spectrally overlaps with other transitions, including $^3H_4 \rightarrow {}^3H_6$ and $^1G_4 \rightarrow {}^3H_5$[39, 45, 46], making it barely distinguishable from others and thus causing negligence. The matching of the STED laser light at 810 nm with the emission spectrum (red-shifted relative to the absorption spectrum) of the $^1D_2 \rightarrow {}^3F_2$ transition is supported by many previous reports that indicate the absorption spectrum of $^3F_2 \rightarrow {}^1D_2$ transition is centered around 800 nm and could cover quite a few nanometers[39, 46–49]. Subsequently, the ions at the $^3F_2$ state quickly decay to the lower-lying $^3F_3$ state and then to the $^3H_4$ state nonradiatively, which increases the population densities of these states.

The occurrence of the stimulated emission process is well supported by the enhanced emission originating from the $^3F_{2,3}$ states at around 700 nm ($^3F_3 \rightarrow {}^3H_6$) and 1064 nm ($^3F_2 \rightarrow {}^3F_4$) after the addition of the 810-nm beam (Figs. 1b and 2d), indicating population increase of the $^3F_{2,3}$ states. It is also supported by the observed decreased lifetime of the 455-nm luminescence with increasing the power of 810 nm beam in UCNPs with different Tm$^{3+}$ concentrations (Fig. 2c, Supplementary Fig. 5, and Supplementary Table 1). Given that the emission cross sections of Tm$^{3+}$ transition are typically in the range of 10$^{-21}$ to 10$^{-20}$ cm$^2$ [42, 50–52], the current depletion laser intensity yields a stimulated emission lifetime from the $^1D_2$ state, $\tau_{STED}$, on the order of a few microseconds (Supplementary Fig. 5), which are three orders of magnitude longer than the stimulated emission lifetime in organic fluorescent dyes (ns vs. ps). The less dramatic lifetime change in some samples was probably attributable to the multiple effects of the depletion laser (discussed below), as well as the stepwise upconversion pumping approach in the lifetime measurement, where the measured decay lifetimes are often not solely determined by the lifetime of the emitting state but also reflect the radiative properties of many long-lived intermediate states[53]. In addition, in UCNPs with low doping (0.5%), where the 475-nm emission (three-photon process) was enhanced with the addition of the depletion laser at 810 nm, the 455-nm emission (five-photon process) was enhanced much less significantly (Fig. 1e). This

result also indicates that stimulated emission occurred for the $^1D_2$ state, because higher-order multiphoton emission from the $^1D_2$ state (455 nm) would be enhanced more significantly than the lower-order multiphoton emission from the $^1G_4$ state (475 nm) if stimulated emission did not occur for the $^1D_2$ state.

Along with the optical depletion through stimulated emission effect of the 810-nm laser beam, a synergistic enhancement effect by the same laser beam is also active due to the presence of a ground-state absorption (GSA) process matching 810 nm, $^3H_6 \xrightarrow{810\,nm} {}^3H_4$, which effectively populates a key intermediate state ($^3H_4$) of the upconversion process (Supplementary Fig. 6). The increase of the population density at the $^3H_4$ state, well supported by the observed enhanced emission at around 1470 nm originating from the transition $^3H_4 \rightarrow {}^3F_4$ (Fig. 2d, Supplementary Fig. 7), is in favor of upconversion emission originating from higher states above the $^3H_4$ state (including the $^1G_4$ and $^1D_2$ states) under the co-irradiation of the 975-nm excitation beam, although single 810-nm excitation can only generate very weak upconversion luminescence (Fig. 1b, Supplementary Fig. 6). This synergistic enhancement effect competes with the STED effect, compromising the optical depletion efficiency induced by the 810-nm beam (Fig. 2b). A key cross relaxation (CR1) process between $Tm^{3+}$ ions could suppress the aforementioned emission enhancement effect of the 810-nm beam and facilitate the optical depletion, $^3H_4 + {}^3H_6 \rightarrow {}^3F_4 + {}^3F_4$, through transferring electrons at the $^3H_4$ state to the $^3F_4$ state (Fig. 2a, b). The occurrence of this $Tm^{3+}$ concentration-dependent CR1 process is confirmed by the infrared emission measurements on nanoparticles with varied $Tm^{3+}$ concentrations under 975-nm excitation, where the emission originating from the $^3F_4 \rightarrow {}^3H_6$ transition (1800 nm) is significantly enhanced, while that from the $^3H_4$ state (1470 nm) is significantly quenched as $Tm^{3+}$ concentration increases (Fig. 2e). Our experimental results have shown that with the increase of the $Tm^{3+}$ doping concentration (from 0.5 to 10%), the ratio $I_{1470}/I_{1800}$ shows a very significant decrease (from about 5:1 to 1:32), indicating an enhancement factor of 160 for $I_{1800}$ with respect to $I_{1470}$ (Fig. 2e, Supplementary Fig. 8). The importance of this cross relaxation process in the blue luminescence inhibition process in our experiments is supported by the large enhancement of the infrared emission at 1800 nm upon the co-irradiation of the 810-nm beam in high $Tm^{3+}$-doped UCNPs (Fig. 2d), and also by the strong dependence of the depletion efficiency on the $Tm^{3+}$ concentration shown in Fig. 1f. These results suggest that the CR1 process can weaken the synergistic enhancement effect of the 810-nm beam and render the STED effect dominant in high $Tm^{3+}$-doped UCNPs. Otherwise, the absence of this CR1 process as in low $Tm^{3+}$-doped UCNPs makes the emission enhancement effect dominant (Fig. 2b, Supplementary Fig. 9).

Furthermore, it should be noted that the CR1 process can simultaneously enhance the STED effect of the 810-nm beam and thus lower the depletion laser intensity requirement. As shown in Fig. 2b, in the case of low $Tm^{3+}$ doping (without the CR1 process), the relaxing electrons from $^1D_2$ to $^3F_2$ via stimulated emission and then to $^3H_4$ would be readily re-pumped back to the $^1D_2$ state via the synergistic excitation of the two lasers. In contrast, in the case of high $Tm^{3+}$ doping the electrons at the $^1D_2$ state are first transferred to the $^3H_4$ state (via $^3F_2$) by the depletion laser, and then the CR1 process consecutively transfers the relaxing electrons from the $^3H_4$ state further to the $^3F_4$ state, where the CR1 process introduces a second energy-dissipating pathway $^3F_4 \rightarrow {}^3H_6$. The cooperation of the STED effect of the 810-nm laser beam and the CR1 process makes the population center of gravity shift toward lower energy

states and helps suppress the re-population rate of the emitting state under two laser irradiation, thus enhancing the stimulated emission-induced depletion efficacy of the depletion laser. This downward shift of the population center of gravity, yielding a new equilibrium among the population densities of different energy states, can also explain the passive depletion of the $^1G_4$ state (475-nm emission) just below the $^1D_2$ state at high $Tm^{3+}$ doping (Fig. 1b).

The slight decrease of the depletion efficiency induced by the 810-nm beam with the $Tm^{3+}$ doping concentration above 10% (Fig. 1f) could be due to the activation of the otherwise inefficient cross relaxations between $Tm^{3+}$ ions at adequately high $Tm^{3+}$ concentration, for example, $^1G_4 + {}^3F_4 \rightarrow {}^1D_2 + {}^3H_6$ and $^3F_{2,3} + {}^3H_4 \rightarrow {}^1D_2 + {}^3H_6$, as reported in a previous report[39], which enhances the synergistic excitation effect of the 810-nm laser beam. The critical role of the $Tm^{3+}$ concentration-dependent CR1 process is also supported by the results of our numerical simulations (Supplementary Figs. 10 and 11, Supplementary Table 2). The associated quenching of the emission from higher-lying states (370/383/394/595 nm) above the $^1D_2$ state[38] precludes depletion pathways through ESA to higher excited states (Fig. 2d, Supplementary Fig. 12).

The proposed optical depletion mechanism of the 455-nm emission for high $Tm^{3+}$-doped (NaYF$_4$:18% $Yb^{3+}$, 10% $Tm^{3+}$) nanoparticles is also well supported by the depletion-laser-wavelength-dependent depletion efficiency data, which were obtained by fixing the wavelength (975 nm) and power of the excitation laser and scanning the wavelength of the second beam from 700 nm to 810 nm while keeping the power constant, as shown in Fig. 2f. A depletion efficiency peak at around 810 nm (corresponding to the $^1D_2 \rightarrow {}^3F_2$ transition) was observed. This broad depletion region is in consistence with the $^1D_2 \rightarrow {}^3F_2$ emission spectrum[49, 54–56], indicating that this transition can have a rather broad emission band in various host materials due to spectrum-broadening effects. The existence of this efficient depletion peak is also consistent with the observed excitation spectrum for the 455-nm emission under sole irradiation of the second beam (Fig. 2g), where a second laser beam at around 810 nm gives rise to weaker luminescence background than other wavelengths. Interestingly, an extra depletion region centered at around 750 nm was observed (Fig. 2f), which is associated with the $^1D_2 \rightarrow {}^3F_3$ transition[31]. This reveals that the population of the $^1D_2$ state can also be efficiently depleted to the $^3F_3$ state via stimulated emission with laser irradiation of proper wavelength, analogous to our proposed stimulated emission pathway by an 810-nm laser associated with the $^1D_2 \rightarrow {}^3F_2$ transition (Supplementary Fig. 13). The difference in the depletion efficiency at these two depletion peaks could be partially due to the different emission cross sections of the $^1D_2 \rightarrow {}^3F_2$ and $^1D_2 \rightarrow {}^3F_3$ transitions[41–44]. In addition, laser light at around 750 nm could have a larger synergistic enhancement effect (cooperating with the 975-nm excitation light) on the 455-nm emission than the 810-nm light, due to its matching with other energy gaps, for example, the $^3H_5 \rightarrow {}^1G_4$ transition[49, 54, 55], which would also degrade the depletion efficiency at this wavelength (Supplementary Fig. 13). Notable 455-nm emission enhancement was induced by addition of a second beam approaching 700 nm (Fig. 2f), which could be caused by the action of the $^3H_6 \rightarrow {}^3F_3$ GSA process that increases the population of the $^3F_3$ state, in favor of multiphoton upconversion of $Tm^{3+}$ ions (Supplementary Fig. 13). In addition, the 455-nm emission enhancement observed under the co-irradiation of a laser beam around 765 nm and the 975-nm excitation laser (Fig. 2f) could be due to the matching of the former with the $^3H_5 \rightarrow {}^1G_4$ ESA

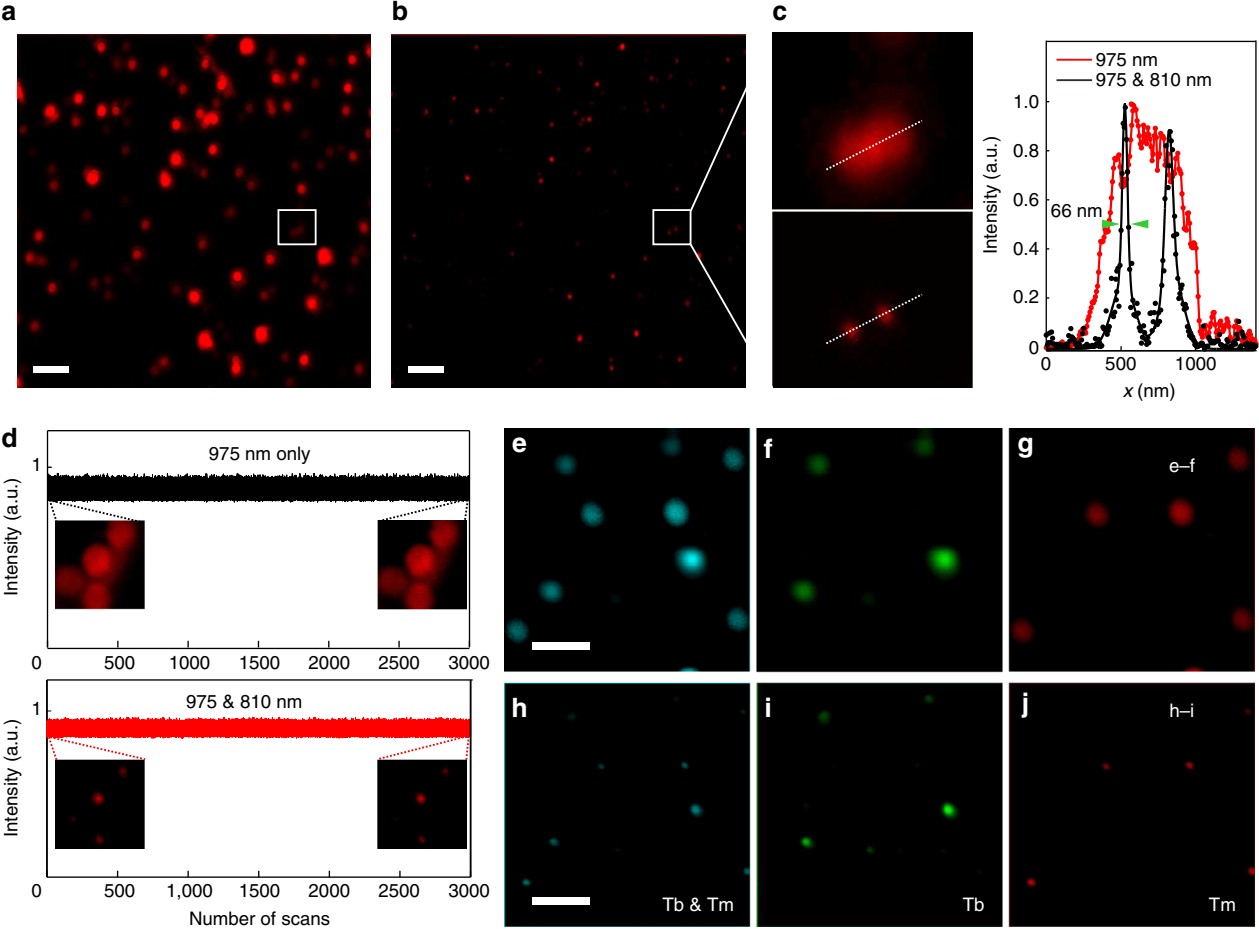

**Fig. 3** Dual-color super-resolution microscopy using UCNPs. **a** NaYF$_4$:18% Yb$^{3+}$, 10% Tm$^{3+}$ (18.0 ± 1.8 nm in diameter) imaged by the multiphoton laser scanning microscope under 975-nm excitation solely. Scale bar: 2.5 μm. **b** The corresponding super-resolution image of the same area under co-irradiation of a 975-nm Gaussian excitation beam and an 810-nm donut-shaped depletion beam. **c** Magnified areas selected from **a** and **b** (marked by white squares) and the corresponding line profiles of the images, fitted with Gaussian functions. **d** Intensity recording over the number of scans under 975-nm excitation solely and 975-nm and 810-nm co-irradiation. **e–j** Dual-color imaging of NaYF$_4$:18% Yb$^{3+}$, 10% Tm$^{3+}$ mixed with NaGdF$_4$:40% Yb$^{3+}$, 10% Tm$^{3+}$@NaGdF$_4$:15% Tb$^{3+}$ (28.0 ± 2.1 nm in diameter). Scale bar: 1 μm. **e–g** Images under single 975-nm excitation; **h–j** images under 975-nm and 810-nm co-irradiation; **e**, **h** the 455-nm emission imaging from both Tm-UCNPs and Tb-UCNPs. **f**, **i** The 547-nm emission imaging from Tb-UCNPs; **g**, **j** the images acquired by subtracting the images of the 547-nm Tb-UCNPs channel (**f**, **i**) from the images of the 455-nm emission channel (**e**, **h**)

process[31, 49, 55], facilitating multiphoton upconversion luminescence by increasing the population of the $^1G_4$ state (Supplementary Fig. 13).

**Two-color optical nanoscopy with high photostability**. This efficient optical depletion mechanism enabled subsequent STED super-resolution microscopy using NaYF$_4$:18% Yb$^{3+}$, 10% Tm$^{3+}$ UCNPs with two NIR laser beams centered at 975 nm and 810 nm. The 975-nm laser beam with a Gaussian distribution intensity profile was the excitation beam, while the 810-nm laser of a donut-shaped intensity distribution provided the depletion beam. In comparison with the single 975-nm excitation, a significant improvement in the spatial resolution was achieved with the addition of the 810-nm depletion laser beam, as demonstrated in Fig. 3a, b. The line profile analysis for a selected area containing two single nanoparticles indicates that the lateral imaging resolution reaches about 66 nm using the excitation wavelength of 975 nm with the 810-nm depletion beam (Fig. 3c). The spatial resolution can likely be further improved by optimizing our lab-made super-resolution imaging system. The power densities for the two CW NIR beams were much lower than those used in

typical CW-STED imaging[11, 34–37, 57], which is attractive for biological studies and technique commercialization. Another attractive advantage offered by this UCNP-mediated nanoscopy technique is the outstanding photostability of the UCNPs. In the test of photostability, the region of interest was scanned 3000 times using both the 975-nm excitation and 810-nm depletion beam simultaneously, with a control region scanned using only the 975-nm beam. The results show that the UCNPs did not show any noticeable photobleaching and photoblinking (Fig. 3d, Supplementary Fig. 14). Such an excellent photostability is highly desired for STED nanoscopic imaging, and the utilization of UCNPs would enable long time, continuous super-resolution imaging[58].

The proposed emission depletion mechanism can be exploited in multiple types of UCNPs for multicolor super-resolution imaging using the same pair of laser beams. This can be achieved by incorporating other activators such as Tb$^{3+}$ and Eu$^{3+}$ into the Yb$^{3+}$-Tm$^{3+}$ system to generate luminescent nanoprobes emitting in complementary spectral channels, taking advantage of energy migration-mediated upconversion constructed on the population of higher excited states of Tm$^{3+}$ above the $^1D_2$ state

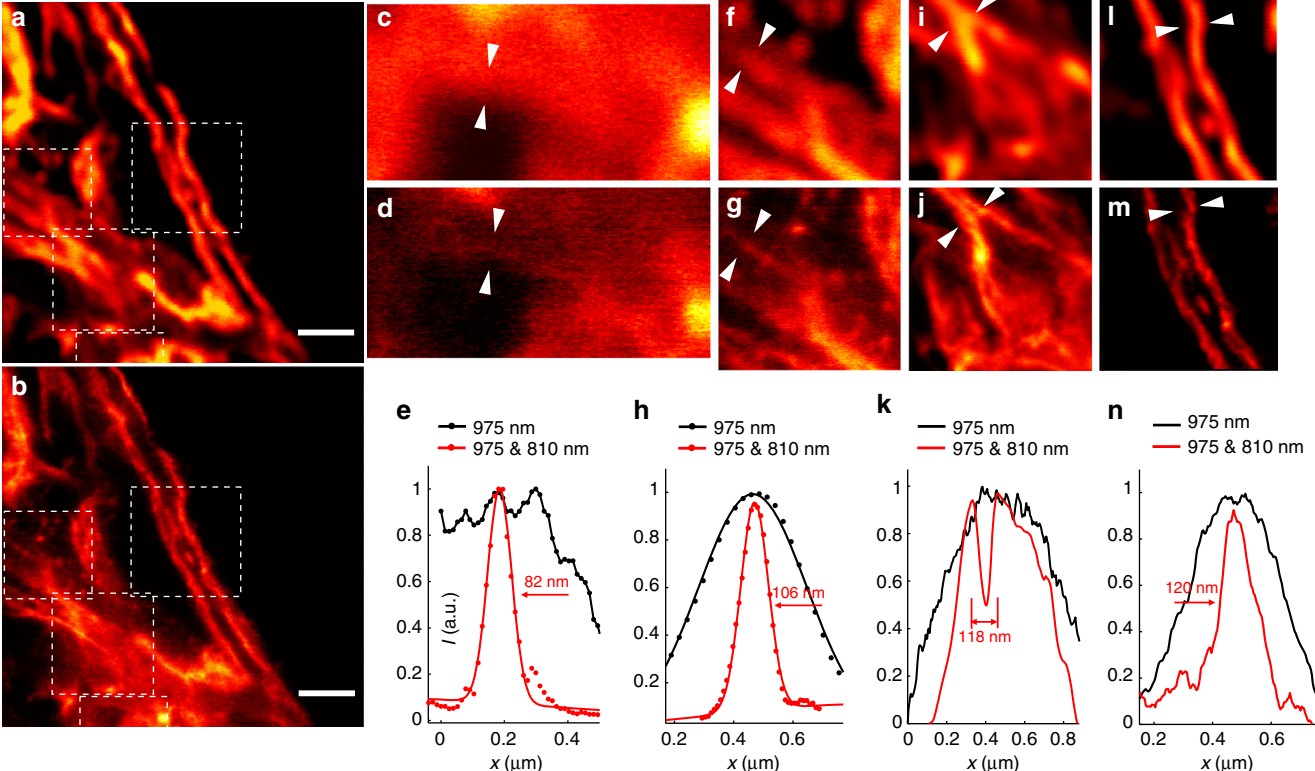

**Fig. 4** Immunofluorescence labeling of cellular cytoskeleton protein desmin with antibody-conjugated UCNPs and super-resolution imaging. **a** The multiphoton imaging under 975-nm excitation of some cytoskeleton structures and desmin proteins in HeLa cancer cells incubated with anti-desmin primary antibody and immunostained with UCNPs ($11.8 \pm 2.2$ nm in diameter) bioconjugated with goat anti-rabbit IgG secondary antibody. **b** The same region with **a** imaged in the super-resolution mode (975-nm excitation and the 810-nm STED laser beam). Scale bars: 2 µm. **c–n** Magnified areas selected from **a** and **b** (marked by white dotted squares) and line profile analyses; images in **c**, **f**, **i**, and **l** are taken from the white dotted squares in **a**; Images in **d**, **g**, **j**, and **m** are taken from the white dotted squares in **b**. **e**, **h**, **k**, **n** Line profiles analyses of several areas indicated by arrow heads in **c** and **d**, **f** and **g**, **i** and **j**, and **l** and **m**, respectively

(Supplementary Fig. 15)[59]. As a proof of concept, a two-color super-resolution imaging was carried out using $NaYF_4$:18% $Yb^{3+}$, 10% $Tm^{3+}$ nanoparticles (type 1) together with $NaGdF_4$:40% $Yb^{3+}$, 10% $Tm^{3+}$@$NaGdF_4$:15% $Tb^{3+}$ nanoparticles (type 2). Type 2 nanoparticles, upon 975-nm excitation, produce bright green emission at around 547 nm of $Tb^{3+}$, indirectly pumped via the $^1I_6$ state of $Tm^{3+}$ (Supplementary Fig. 16). As expected, the green emission of type 2 can be efficiently depleted using the 810-nm beam, here with efficiency of 88% (Supplementary Fig. 16). The mixture of both types of nanoparticles was spin-coated onto a glass coverslip for two-channel imaging. The target was first imaged in the 455-nm channel, which yielded images of both types of nanoparticles (Fig. 3e, h), since both types emitted in this blue channel. Subsequent imaging in the 547-nm channel resulted in images of solely type 2 nanoparticles (Fig. 3f, i), where type 1 nanoparticles were absent because they did not give rise to green emission under excitation of either laser beam or the combination of both. By comparing the images taken in the blue and green channels, type 1 nanoparticles can be completely distinguished (Fig. 3g, j). This super-resolution imaging strategy can be readily extended to even more spectral channels by introducing, for example, $Eu^{3+}$- and $Sm^{3+}$-activated nanoparticles, pumped through a high excited state of $Tm^{3+}$. This strategy eliminates the need for one pair of excitation and depletion beams for each color channel, which currently constitutes a major obstacle for multicolor STED due to the challenge of aligning multiple laser beams[60]. The pixel dwelling time (100 µs) employed in the laser

scanning super-resolution imaging is comparable to that used in traditional STED imaging (50–200 µs) and is significantly shorter than that reported in a very recent work regarding nanoscopy using UCNPs (4–6 ms)[31]. The image acquisition speed can be increased with STED-parallelization strategies[61].

**Immunofluorescence nanoscopy of cytoskeleton with UCNPs.** In the recent decade lanthanide-doped UCNPs have been extensively developed to be an important group of biomarkers for various biological applications. However, there is a long-standing question of whether UCNPs can be used as an alternative fluorophore for subcellular fine structure imaging. In this study, we managed to image the cytoskeleton of HeLa cancer cells with UCNP-mediated nanoscopy. High $Tm^{3+}$-doped UCNPs ($NaGdF_4$:18% $Yb^{3+}$, 10% $Tm^{3+}$) with an average diameter of 11.8 ($\pm$2.2) nm were synthesized (Supplementary Fig. 4). After surface modification with poly(acrylic acid) (PAA) (Supplementary Fig. 17), these UCNPs were conjugated with the secondary antibody (goat anti-rabbit IgG antibody) (Supplementary Fig. 18). Fixed HeLa cancer cells were incubated with anti-desmin monoclonal antibody (cytoskeleton marker)[62], and then stained with the secondary antibody IgG-conjugated UCNPs. Such immunostained HeLa cells were imaged in the super-resolution mode under 975-nm and 810-nm co-irradiation, and with a reference under irradiation of only the 975-nm laser.

As shown in Fig. 4, subcellular structures were clearly imaged, and a significant improvement in the spatial resolution was

achieved with the assistance of the 810-nm depletion laser beam (Fig. 4a–n). To evaluate the resolution of STED imaging, line profiles of several areas were extracted and fitted with Gaussian functions. The results reveal that the lateral imaging resolution has been improved to about 82 nm, less than $\lambda/10$ ($\lambda_{ex} = 975$ nm) (Fig. 4c–e). To the best of our knowledge, this is the first demonstration of successful immunolabeling of fine subcellular structures as well as the first demonstration of super-resolution cell imaging using UCNPs.

## Discussion

Optical depletion of upconversion luminescence has long remained a challenge, since the depletion laser wavelength easily matches with multiple energy gaps, complicating the intended role of the depletion laser with both emission enhancement and depletion effects active. In this work, we utilize ion–ion interaction to suppress the enhancement effect and enhance the depletion effect of the depletion laser beam, and achieve large optical depletion of upconversion luminescence in a major group of UCNPs. Specifically, we have successfully achieved 96% depletion of the blue emission at 455 nm ($\lambda_{exc} = 975$ nm) in NaYF$_4$:18% Yb$^{3+}$, 10% Tm$^{3+}$ nanoparticles with an 810-nm beam. The strong cross-relaxation between low-lying states of the emitters (Tm$^{3+}$), $^3H_4 + {^3H_6} \rightarrow {^3F_4} + {^3F_4}$, plays a critical role in the optical inhibition of the 455-nm emission. We have successfully implemented STED super-resolution imaging using these UCNPs. By adopting nanoparticles emitting in other color channels, which were indirectly pumped via a high excited state of Tm$^{3+}$ through energy migration upconversion, we have implemented two-color super-resolution imaging using a single excitation/depletion laser beam pair. In addition, we have achieved a successful immunolabeling of fine subcellular structures and super-resolution cellular imaging using UCNPs. The upconversion emission depletion nanoscopy developed here significantly reduces the requirement of the depletion power density, as also demonstrated in a very recent similar work[31], and removes the need for bulky and expensive pulse lasers and makes STED microscopy more biocompatible, providing alternatives to traditional STED imaging using organic dyes and other luminescent nanoparticles. We believe that our work will provide new insight into developing optical modulation methods for luminophores toward broad applications in super-resolution imaging and beyond and will significantly extend the range of applications of UCNPs in nanoscale photonics and life science.

## Methods

**Optical setup for spectroscopic and microscopic study**. A lab-made optical system was built to perform spectroscopic measurement and super-resolution imaging, as shown in Supplementary Fig. 2. The 975-nm laser beam was generated by a CW, single-mode diode laser (Laser 1, B&A Technology Co. Ltd., Shanghai). A band-pass filter (F2, LD01-975/10-25, Semrock) was used to purify the laser spectrum. The 810-nm CW laser beam was provided by a Ti:sapphire laser (Laser 2, Mira 900, Coherent) and filtered with a band-pass filter (F1, FF01-810/10-25, Semrock). A pinhole in combination with two lenses was used as a spatial filter to optimize the beam profile. Two pairs of half-wave plate and polarization beam splitter were placed in the optical path to control the laser power. These two NIR laser beams were then spatially overlapped using a 950-nm short-pass dichroic mirror (DM1, ZT950spxrxt, Chroma), and then directed into a multiphoton laser scanning microscope. In the super-resolution imaging, a vortex phase plate (VPP-1a, RPC Photonics) was employed to perform a $2\pi$ helix phase modulation of the 810-nm beam, generating a donut-shaped beam. Another half-wave plate (P3) combined with a quarter-wave plate attached to the objective lens was used to make the 810-nm laser beam in circular polarization. The coaxial laser beams were reflected by a 775-nm short-pass dichroic mirror (DM2, ZT775sp-2p-UF3, Chroma), and focused on the sample using an oil-immersed objective (OL1, PL-APO ×60/1.35, Olympus).

The visible emission (400–755 nm) was collected through the same objective and filtered by a 775-nm short-pass filter (F3, FF01–775/SP-25, Semrock) and then captured by a spectrometer (QE65000, Ocean Optics) or a photomultiplier tube

(PMT) for spectroscopic or imaging studies. The NIR emission was collected by another objective (OL2, ×20/0.5, Olympus) set on the condenser holder above the sample and filtered by a 980-nm long-pass filter (F6, BLP01-980R-25, Semrock), detected by two NIR spectrometers: NIRQuest, Ocean Optics (spectral range: 1100–2100 nm); and Shamrock 303i-B, iDus Spectroscopy Cameras, Andor (spectral range: 800–1500 nm).

**Luminescence lifetime measurement**. Emission lifetime measurements of UCNPs were performed by modulating the 975-nm excitation laser with a chopper (model SR540, Stanford) and recording the time-resolved luminescence intensity. The emission photons, passing through a 448-nm band-pass filter (F4, ET448/×19, Chroma), were detected by a PMT, and the trigger signal from the chopper was synchronized with a time-correlated single-photon counter (NanoHarp, Pico-quant). The chopper modulation frequencies were varied with the Tm$^{3+}$ doping concentrations, 480, 530, 796, 890, 1000, 1276, 2000, and 3500 Hz for UCNPs doped with 0.5% Tm$^{3+}$, 2% Tm$^{3+}$, 5% Tm$^{3+}$, 7% Tm$^{3+}$, 10% Tm$^{3+}$, 15% Tm$^{3+}$, and 20% Tm$^{3+}$, respectively. To minimize the influence of the instrument response function, for example, the falling edge of the mechanically modulated light pulse, the beam was suitably focused and the chopper was placed in the focal plane. In addition, iterative deconvolution was performed with the instrument response function, measured with a rhodamine sample with a lifetime on order of nanoseconds.

**Dual-color super-resolution imaging**. In the dual-color super-resolution micro-scopic imaging, a 495-nm long-pass dichroic mirror (DM3, FF495-Di03-25 × 36, Semrock) was used to separate the luminescence into two detectors. The 448-nm band-pass filter F4 was placed in front of PMT1 to collect the 455-nm emission signal from NaYF$_4$:18% Yb$^{3+}$/10% Tm$^{3+}$ and NaGdF$_4$:40% Yb$^{3+}$/10% Tm$^{3+}$@NaGdF$_4$:15% Tb$^{3+}$, and a 550-nm band-pass filter (F5, ET550/×20, Chroma) was placed in front of PMT2 to collect the 547-nm emission signal only from NaGdF$_4$:40% Yb$^{3+}$/10% Tm$^{3+}$@NaGdF$_4$:15% Tb$^{3+}$. Image subtraction using the data from the two channels allowed for separate imaging of the two types of UCNPs, enabling dual-color super-resolution imaging.

**Synthesis of Tm$^{3+}$-NaYF$_4$ UCNPs**. The designed UCNPs were synthesized following previously reported protocols with some modifications[28, 29]. In a typical synthesis, 5 mL Ln(CH$_3$CO$_2$)$_3$ (Ln = Y/Yb/Tm) stock solution (0.2 M) was added into a 100-mL round bottom flask, followed by the addition of 7.5 mL oleic acid (OA) and 17.5 mL 1-octadecene (ODE). The mixture was heated to 150 °C under stirring for 40 min to form the lanthanide-oleate precursor solution, and then cooled down to room temperature. Then 10 mL of NH$_4$F-methanol solution (0.4 M) and 2.5 mL of NaOH-methanol solution (1.0 M) were pipetted into a 15-mL centrifuge tube, and then the mixture solution was quickly injected into the flask. Subsequently, the mixture was heated to 50 °C and kept at that temperature for at least 0.5 h, and then was heated to 110 °C under vacuum to remove methanol. After methanol was evaporated, the mixture was heated to 300 °C and incubated at that temperature for 1.5 h under an argon atmosphere, and then cooled down to room temperature. The UCNPs were precipitated by the addition of ethanol, followed by centrifugation at 7500 r.p.m. for 5 min. The obtained UCNPs were washed several times using ethanol and cyclohexane, and were finally re-dispersed into cyclohexane for subsequent use. Besides, the NaYF$_4$:18% Yb$^{3+}$, 10% Tm$^{3+}$ UCNPs with an average diameter of 49.5($\pm$1.6) nm (Supplementary Fig. 4) were synthesized with minor modification, and the reaction was iso-thermally kept at 320 °C for 2 h.

**Synthesis of core-shell Tm$^{3+}$/Tb$^{3+}$-NaGdF$_4$ UCNPs**. The fabrication of the core NaGdF$_4$:40% Yb$^{3+}$/10% Tm$^{3+}$ UCNPs with an average diameter of 15.7($\pm$1.9) nm used the same general synthesis protocol with modifications. In the synthesis, 5 mL Ln(CH$_3$CO$_2$)$_3$ (Ln = Gd/Yb/Tm) stock solution (0.2 M) was added into a 100-mL round bottom flask, followed by the addition of 10 mL OA and 15 mL ODE. The mixture was heated to 150 °C under stirring for 40 min to form the lanthanide-oleate precursor solution, and then cooled down to room temperature. Then 11 mL of NH$_4$F-methanol stock solution and 2.5 mL of NaOH-methanol stock solution were pipetted into a 15-mL centrifuge tube, and then the mixture solution was quickly injected into the flask. Subsequently, the mixture was kept at 50 °C for at least 0.5 h, and then was heated to 110 °C under vacuum to remove methanol. After methanol was evaporated, the mixture was heated to 290 °C and incubated at that temperature for 1.5 h under an argon atmosphere, and then cooled down to room temperature. The UCNPs were precipitated by the addition of ethanol, followed by centrifugation at 7500 r.p.m. for 5 min. The obtained UCNPs were washed for several times using ethanol and cyclohexane and were finally re-dispersed in 8 mL cyclohexane for subsequent use. In addition, the NaGdF$_4$:18% Yb$^{3+}$/10% Tm$^{3+}$ UCNPs with an average diameter of 11.8($\pm$2.2) nm used in the immunostaining experiments (Fig. 4 in the main text) were synthesized using the same procedure, except that the reaction temperature was adjusted to 280 °C.

The core-shell NaGdF$_4$:40% Yb$^{3+}$/10% Tm$^{3+}$@NaGdF$_4$:15% Tb$^{3+}$ nanoparticles were synthesized following a similar procedure to the synthesis of the NaGdF$_4$:40% Yb$^{3+}$/10% Tm$^{3+}$ UCNPs. In the synthesis, the shell precursor was prepared by

mixing 2.5 mL Ln(CH$_3$CO$_2$)$_3$ (Ln$^{3+}$ = Gd$^{3+}$/Tb$^{3+}$) stock solution with 5 mL of OA and 7.5 mL of ODE in a 100-mL flask. The mixture was heated to 150 °C under stirring for 40 min to form a homogeneous solution, and then cooled down to room temperature. Then, 4 mL of as-prepared core nanoparticle suspension was injected into the reaction flask, followed by the addition of 5 mL NH$_4$F-methanol stock solution and 1.25 mL NaOH-methanol stock solution. Subsequent steps were the same as in the synthesis of the NaGdF$_4$:40% Yb$^{3+}$/10% Tm$^{3+}$ UCNPs.

**Preparation of hydrophilic Tm$^{3+}$-UCNPs.** Two methods were used to prepare hydrophilic Tm-UCNPs. One is the modified method of ligand removal with hydrochloric acid (HCl)[63, 64]. In a typical experiment, 60 mg dried OA-UCNPs were re-dispersed into 3 mL 1% HCl solution, and then was sonicated for 1 h. The desorbed OA and naked UCNPs were separated by extraction with ether, followed by centrifugation at 13,000 r.p.m. for 30 min. The supernatant was discarded and the obtained pellets were dispersed into 1 mL deionized water by sonication. Another hydrophilic processing method is ligand exchange to replace the oleate on the nanoparticle surface with PAA[65, 66]. In a typical experiment, 30 mL diethylene glycol and 0.5 g PAA were added to a 100-mL bottom flask. The mixture was heated to 110 °C with vigorous stirring under an argon atmosphere for 1 h. Then 4 mL OA-UCNPs (ca. 60 mg) was slowly injected into the hot solution and the mixture was kept under vacuum for 30 min to evaporate cyclohexane. Subsequently, the mixture was heated to 240 °C and kept for 3 h under argon protection until the solution became clear. Then the mixture was cooled down to room temperature. After adding 15 mL 1% HCl aqueous solution and stirring for 15 min, a transparent gelatinous system was obtained. The UCNPs were precipitated by centrifugation at 15,000 r.p.m. for 30 min, followed by washing with ethanol and deionized water (v/v = 1:1) for several times. Finally, the products were re-dispersed in 4 mL fresh deionized water to form a clear solution for subsequent use.

**Upconversion luminescence immunolabeling of cytoskeleton.** Activation of the carboxyl groups on the surface of UCNPs were carried out according to the commonly used methods with modifications[67, 68]. First, PAA-UCNP aqueous solution (1 mg mL$^{-1}$) was prepared, and N-(3-dimethylaminopropyl)-N′-ethylcarbodiimide hydrochloride (EDC) and N-hydroxysuccinimide (NHS) solids were dissolved in 2-(N-morpholino)ethanesulfonic solution at concentrations of 0.2 and 0.3 mg μL$^{-1}$, respectively. After adding 20 μL NHS and 20 μL EDC solution into 1 mL PAA-UCNPs aqueous solution, the mixture was stirred for 2 h at room temperature. Subsequently, the UCNPs were obtained via centrifugation at 15,000 r.p.m. for 30 min and then re-dispersed in 1 mL ultrapure water. The pH value of the solution was adjusted to 7.2–7.5 to ensure an efficient reaction between the antibody and the NHS-activated UCNPs. After the addition of 500 μg goat anti-rabbit antibody IgG (dissolved in antibody dilution), the mixture was stored in a refrigerator at 4 °C overnight, followed by centrifugation at 14,000 r.p.m. for 20 min. The obtained precipitate was dispersed into phosphate-buffered saline (PBS) buffer for subsequent use.

A commonly used immunostaining technique was used to perform cytoskeleton labeling using UCNPs[67, 69]. The HeLa cells were first cultured in 96-well plates with about 15,000 cells in each unit and then were rinsed three times using PBS buffer to remove culture media. Subsequently, the cells were fixed using 4% paraformaldehyde (PFA) for 15 min at room temperature. Next, the fixed cells were rinsed three times again to remove the residual PFA. The HeLa cells were permeabilized in 0.2% Triton X-100 for 30 min at room temperature and were rinsed again using PBS. Quick Block blocking buffer was then used to block the cells for 20 min at room temperature. The antibody dilutions were prepared using the Quick Block primary antibody dilution buffer. The primary antibody (anti-desmin monoclonal antibody) solution of 5 μg mL$^{-1}$ was used to incubate the fixed cells at 4 °C overnight. Then, the incubated sample was rinsed three times using PBS buffer. Then, the as-prepared UCNPs bioconjugated with secondary antibodies were dissolved in PBS buffer after centrifugation. A volume of 100 μL UCNPs solution (200 μg mL$^{-1}$) was added to 96-well plates with the primary antibody-treated fixed cells. The immunostaining reaction of cytoskeleton desmin protein was kept at room temperature for 2 h. The cells were then rinsed gently with pre-warmed (37 °C) PBS buffer twice. Then, the HeLa cells were imaged using the multiphoton laser scanning microscope. The super-resolution imaging was performed with a pixel dwelling time of 100 μs.

**Theoretical modeling and numerical simulation.** The proposed optical depletion mechanism of the Tm$^{3+}$ emission in NaYF$_4$:Yb$^{3+}$/Tm$^{3+}$ nanocrystals is shown in Supplementary Fig. 10. The upconversion process in Yb$^{3+}$-Tm$^{3+}$ co-doped system can be described by the rate equations of optical direct excitation/depletion and interionic energy transfer. According to the proposed energy transfer upconversion process shown in Supplementary Fig. 10, the rate equations of each energy state are derived as follows:

$$\text{Tm}^{3+}\left(^3\text{H}_6\right): \frac{dn_0}{dt} = -\sum_{i=1}^{8}\frac{dn_i}{dt} \tag{1}$$

$$\begin{aligned}\text{Tm}^{3+}\left(^3\text{F}_4\right): \frac{dn_1}{dt} =& \beta_2 n_2 + 2c_1 n_0 n_3 + c_2 n_3 n_5 + c_3 n_3 n_5 \\ &+ b_{31}\frac{n_3}{\tau_3} + b_{51}\frac{n_5}{\tau_5} + b_{61}\frac{n_6}{\tau_6} \\ &+ b_{71}\frac{n_7}{\tau_7} + b_{81}\frac{n_8}{\tau_8} - w_2 n_{\text{Yb1}}n_1 - \frac{n_1}{\tau_1}\end{aligned} \tag{2}$$

$$\begin{aligned}\text{Tm}^{3+}\left(^3\text{H}_5\right): \frac{dn_2}{dt} =& \frac{\sigma_{d2}^{se}I_d}{h\nu_d}n_6 - \frac{\sigma_{d2}^{a}I_d}{h\nu_d}n_2 \\ &+ w_1 n_{\text{Yb1}}n_0 + b_{62}\frac{n_6}{\tau_6} \\ &+ b_{82}\frac{n_8}{\tau_8} + \beta_3 n_3 - \beta_2 n_2 - \frac{n_2}{\tau_2}\end{aligned} \tag{3}$$

$$\begin{aligned}\text{Tm}^{3+}\left(^3\text{H}_4\right): \frac{dn_3}{dt} =& \frac{\sigma_{d1}^{a}I_d}{h\nu_d}n_0 - \frac{\sigma_{d1}^{se}I_d}{h\nu_d}n_3 + b_{63}\frac{n_6}{\tau_6} + b_{73}\frac{n_7}{\tau_7} \\ &+ \beta_4 n_4 + c_4 n_0 n_7 - c_1 n_0 n_3 - c_2 n_3 n_6 \\ &- c_3 n_3 n_6 - w_3 n_{\text{Yb1}}n_3 - \beta_3 n_3 - \frac{n_3}{\tau_3}\end{aligned} \tag{4}$$

$$\text{Tm}^{3+}\left(^3\text{F}_3\right): \frac{dn_4}{dt} = \beta_5 n_5 + b_{74}\frac{n_7}{\tau_7} - \beta_4 n_4 - \frac{n_4}{\tau_4} \tag{5}$$

$$\begin{aligned}\text{Tm}^{3+}\left(^3\text{F}_2\right): \frac{dn_5}{dt} =& \frac{\sigma_{d3}^{se}I_d}{h\nu_d}n_7 - \frac{\sigma_{d3}^{a}I_d}{h\nu_d}n_5 + w_2 n_{\text{Yb1}}n_1 \\ &+ c_4 n_0 n_7 + b_{75}\frac{n_7}{\tau_7} + \beta_6 n_6 - \beta_5 n_5 - \frac{n_5}{\tau_5}\end{aligned} \tag{6}$$

$$\begin{aligned}\text{Tm}^{3+}\left(^1\text{G}_4\right): \frac{dn_6}{dt} =& \frac{\sigma_{d2}^{a}I_d}{h\nu_d}n_2 - \frac{\sigma_{d2}^{se}I_d}{h\nu_d}n_6 + w_3 n_{\text{Yb1}}n_3 \\ &- c_2 n_3 n_6 - c_3 n_3 n_6 - \beta_6 n_6 - \frac{n_6}{\tau_6}\end{aligned} \tag{7}$$

$$\begin{aligned}\text{Tm}^{3+}\left(^1\text{D}_2\right): \frac{dn_7}{dt} =& -\frac{\sigma_{d3}^{se}I_d}{h\nu_d}n_7 + \frac{\sigma_{d3}^{a}I_d}{h\nu_d}n_5 \\ &+ c_2 n_3 n_6 + c_3 n_3 n_6 - w_4 n_{\text{Yb1}}n_7 - c_4 n_0 n_7 - \frac{n_7}{\tau_7}\end{aligned} \tag{8}$$

$$\text{Tm}^{3+}\left(^1\text{I}_6\right): \frac{dn_8}{dt} = w_4 n_{\text{Yb1}}n_7 - \frac{n_8}{\tau_8} \tag{9}$$

$$\text{Yb}^{3+}\left(^2\text{F}_{7/2}\right): \frac{dn_{\text{Yb0}}}{dt} = -\frac{dn_{\text{Yb1}}}{dt} \tag{10}$$

$$\begin{aligned}\text{Yb}^{3+}\left(^2\text{F}_{5/2}\right): \frac{dn_{\text{Yb1}}}{dt} =& \frac{\sigma_p^a I_p}{h\nu_p}n_{\text{Yb0}} \\ &- (w_1 n_0 + w_2 n_1 + w_3 n_3 + w_4 n_7)n_{\text{Yb1}} - \frac{n_{\text{Yb1}}}{\tau_{\text{Yb1}}}\end{aligned} \tag{11}$$

Here $n_i$ ($i = 0$–8) and $\tau_i$ ($i = 1$–8) represent the population densities and radiative lifetimes of the $^3\text{H}_6$, $^3\text{F}_4$, $^3\text{H}_5$, $^3\text{H}_4$, $^3\text{F}_3$, $^3\text{F}_2$, $^1\text{G}_4$, $^1\text{D}_2$, and $^1\text{I}_6$ states of Tm$^{3+}$ ions, respectively. $w_i$ ($i = 1$–4) is the energy transfer rate from the $^2\text{F}_{5/2}$ state of Yb$^{3+}$ to the $^3\text{H}_6$, $^3\text{F}_4$, $^3\text{H}_4$, and $^1\text{D}_2$ states of Tm$^{3+}$, respectively. $c_i$ ($i = 1$, 2, 3, 4) denotes the Tm$^{3+}$-Tm$^{3+}$ cross-relaxation processes $^3\text{H}_4 + {}^3\text{H}_6 \rightarrow {}^3\text{F}_4 + {}^3\text{F}_4$, $^1\text{G}_4 + {}^3\text{H}_4 \rightarrow {}^1\text{D}_2 + {}^3\text{F}_4$, $^1\text{G}_4 + {}^3\text{H}_4 \rightarrow {}^3\text{F}_4 + {}^1\text{D}_2$, and $^1\text{D}_2 + {}^3\text{H}_6 \rightarrow {}^3\text{F}_2 + {}^3\text{H}_4$, respectively. $\beta_i$ ($i = 2$, 3, 4, 5, 6) is the nonradiative decay rates of the $^3\text{H}_5$, $^3\text{H}_4$, $^3\text{F}_3$, $^3\text{F}_2$, and $^1\text{G}_4$ states of Tm$^{3+}$, respectively. $b_{ij}$ is the branching ratio for the radiative transition from the initial state $i$ to the terminal state $j$ of Tm$^{3+}$ ($j < i$). $\sigma_{d1}^a$, $\sigma_{d2}^a$, and $\sigma_{d3}^a$ are the cross sections for the absorption processes $^3\text{H}_6 \xrightarrow{810\,\text{nm}} {}^3\text{H}_4$, $^3\text{H}_5 \xrightarrow{810\,\text{nm}} {}^1\text{G}_4$, and $^3\text{F}_2 \xrightarrow{810\,\text{nm}} {}^1\text{D}_2$ of Tm$^{3+}$, respectively. $\sigma_{d1}^{se}$, $\sigma_{d2}^{se}$, and $\sigma_{d3}^{se}$ are the cross sections for the stimulated emission processes $^3\text{H}_4 \xrightarrow{810\,\text{nm}} {}^3\text{H}_6$, $^1\text{G}_4 \xrightarrow{810\,\text{nm}} {}^3\text{H}_5$, and $^1\text{D}_2 \xrightarrow{810\,\text{nm}} {}^3\text{F}_2$ of Tm$^{3+}$, respectively. $n_{\text{Yb0}}$ and $n_{\text{Yb1}}$ are the population densities of the ground state and the excited state of Yb$^{3+}$. $\sigma_p^a$ is the absorption cross section of Yb$^{3+}$ at 975 nm. $\tau_{\text{Yb1}}$ is the lifetime of the excited state of Yb$^{3+}$. $h$ is Planck's constant. $\nu_{\text{p/d}}$ is the frequency of the 975-/810-nm laser light. $I_{\text{p/d}}$ is the excitation intensity of the 975-/810-nm laser beam.

Due to the matching of the 810-nm laser (full width at half maximum of 4 nm) with multiple transitions, including $^3H_6 \overset{810\,nm}{\longleftrightarrow} {}^3H_4$, $^3H_5 \overset{810\,nm}{\longleftrightarrow} {}^1G_4$, and $^3F_2 \overset{810\,nm}{\longleftrightarrow} {}^1D_2$[39, 45, 46], three light absorption and three stimulated emission processes at 810 nm are all considered in this model, with different cross sections (Supplementary Table 2). Regarding the net effect of the 810-nm laser on the population density of the $^3H_4$ state, fundamentally determined by the magnitude relationship between two items, namely, $\frac{\sigma_{d1}^{a} I_d}{h\nu_d} n_0$ and $\frac{\sigma_{d1}^{st} I_d}{h\nu_d} n_3$, our spectroscopic result of the 1470-nm band ($^3H_4 \rightarrow {}^3F_4$) (Fig. 2d) indicates that the GSA process $^3H_6 \overset{810\,nm}{\longrightarrow} {}^3H_4$ is dominant over the stimulated emission process ($^3H_4 \overset{810\,nm}{\longrightarrow} {}^3H_6$). In addition, the stimulated emission process ($^1D_2 \overset{810\,nm}{\longrightarrow} {}^3F_2$) is supposed to be dominant over the ESA process $^3F_2 \overset{810\,nm}{\longrightarrow} {}^1D_2$, due to a larger population density at the $^1D_2$ state than at the $^3F_2$ state and a larger emission cross section than the absorption cross section at this wavelength. For simplicity, only the net effects of the 810-nm laser on the $^3H_6 \overset{810\,nm}{\longleftrightarrow} {}^3H_4$, $^3H_5 \overset{810\,nm}{\longleftrightarrow} {}^1G_4$, and $^3F_2 \overset{810\,nm}{\longleftrightarrow} {}^1D_2$ transitions are illustrated in the mechanism diagram (Fig. 2a). The $Tm^{3+}$ concentration-dependence of the depletion efficiency of the 455-nm emission was investigated by varying the parameter values, particularly those for cross-relaxation processes. The values used for the main parameters for both low and high $Tm^{3+}$-doped samples are tabulated in Supplementary Table 2.

**Data availability**. All the relevant data that support the findings of this work are available from the correspondence authors upon reasonable request.

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

## Acknowledgements

This work was supported by the National Natural Science Foundation of China (61675071, 61405062, and 91233208), the Pearl River Nova Program of Guangzhou (201710010010), the Guangdong Innovative Research Team Program (201001D00104799318 and 2011D039), the Natural Science Foundation of Guangdong province (2014A030313445), the China Postdoctoral Science Foundation (2016M600659, XJ2015018, and 2014T70818), Joint International Research Laboratory of Optical Information, and the Key Laboratory of Optoelectronic Devices and Systems of Ministry of Education and Guangdong Province (Shenzhen University). H.L. acknowledges the financial support from an International Postdoc grant (2015–00160) and a Starting Grant (2016–03804) from Swedish Research Council. We thank Dr. Fuhong Cai at Hainan University and Prof. Cuifang Kuang at Zhejiang University for helpful advices on the STED technique; Prof. Guanying Chen at Harbin Institute of Technology and Dr. Kai Huang at University of Massachusetts Medical School for helpful advices on the surface functionalization; Dr. Jing Liu at Gent University, Dr. Shuiqin Niu at Southern University of Science and Technology, and Miss Nana Li at Foshan for their helpful advices on the cell immunolabelling; and Dr. Julian Evans at Zhejiang University for the proof reading of the manuscript.

## Author contributions

Q.Z. conceived and designed this project. Q.Z. directed the experiments with contribution from H.L. B.W., Q.W., and Q.Z. built the optical system. B.W., Q.W., Q.Z., and X.P. acquired and processed data. B.H., C.Z., R.P., and Q.Z. were responsible for synthesis, surface functionalization, and characterization of nanoparticles. C.Z. and Q.Z. prepared biosamples. Q.Z., H.L., R.P., B.W., Q.W., and S.H. analyzed data. H.L., Q.Z., and R.P. were responsible for the theoretical analysis and simulation with contribution from H.Å. and S.H. Q.Z. and S.H. supervised the project. H.L., Q.Z., and S.H. wrote the paper with contribution from all authors.

## Additional information

**Competing interests:** The authors declare no competing financial interests.

