## [Peer Review File · Nature Communications]

Reviewers' comments:

Reviewer #1 (Remarks to the Author):

Zhan et al. report upconversion nanoparticles (UCNPs) with controllable emission for use in super-resolution STED nanoscopy by using 975 nm excitation and 810 nm depletion. They have also included an elaborate analysis of the switching mechanism. I am not an expert on lanthanide photoluminescence; therefore, I shall not comment on the mechanistic issues but rather focus on the application to microscopy.

Although I find the presented scheme of luminescence switching interesting, I cannot recommend publication in Nature Communications because the manuscript does not make a convincing case for clear advantages of using these UCNPs for imaging. First of all, they are huge (18/19/22/28 nm as reported in the MS and Fig. S1) in comparison with typical proteins, which severely limits their use as biomarkers. Second, like other nanocrystals, UCNPs are difficult to conjugate to interesting biomolecules. Third, the use of near-IR light means that there is a clear disadvantage for achieving high resolution, and even the reported 66 nm (which I question) are only slightly better than what can be achieved with linear SIM. I will comment on the quoted resolution enhancement below. Moreover, the UCNP luminescence lifetimes are 4 orders of magnitude greater than those of the usual fluorophores, which limits the emission intensity. All of these issues should be thoroughly discussed and compared with established approaches to present a well-balanced account.

I have a number of major and minor specific points of criticism that should be addressed before resubmitting this manuscript, possibly to a well-edited specialty journal (J. Luminescence, J. Microscopy etc.).

Major points of criticism:

1. The Line 97, Fig. S1. How were the sizes determined? No size distributions are reported, no errors and statistics are provided. The same applies to the water soluble UCNPs, which have not been characterized physicochemically (spectra, brightness, DLS etc.).

2. Line 178 "...lateral imaging resolution reaches 66 nm...". This number, which is also used in the abstract, is essentially meaningless. The cut through the image is very noisy, and just selecting the narrowest feature is not scientifically sound. There are sophisticated, Fourier-based algorithms to determine resolution. At least, one would expect statistics over a large number of cross-sections, including a statistical analysis. In fact, the FWHM of the neighboring UCNP is already much larger. Also, it appears that there is a broad halo of non-depleted low-resolution background intensity, which is suppressed in the image.

3. The authors claim that the cell imaging studies reveal tracks of single nanoparticles. I have severe doubts that this is true. It was not characterized how these particles enter the cell. Most likely, they were endocytosed, which means that multiple particles reside in one vesicle. This is also suggested by the apparently greater size dispersion in the image. Again, there is only a single cross section shown. Without appropriate statistics, this is meaningless.

4. The manuscript and especially the supplement could greatly benefit from careful editing. Examples of mistakes from the supplement: "2π-heli phase", "photon multiple tube (PMT)", "muliphoton laser scanning", no units given in Table S1.

Minor points of criticism:

1. Line 19: "...which generally requires dangerously high light intensity..." What is "dangerously" supposed to mean? I find it completely inappropriate here. Moreover, the authors use 17.7 MW/cm^2 to reach 96% depletion, corresponding to 50 mW (at the sample?), if I understand their data correctly. Is that dangerous as well in the view of the authors? How did they calculate the power density?
2. Line 45: SIM works in the realm of diffraction, as long as it is used in a linear way (REF 18). The sentence should be stated clearly. Also, MINFLUX should be mentioned as a new super-resolution approach.
3. Lifetimes should be reported on semi-log plots (Fig. 3e, Fig. S7).

Reviewer #2 (Remarks to the Author):

The current manuscript describes a strategy for high-efficiency emission depletion by introducing auxiliary matter-matter interaction inside the highly doped upconversion nanoparticles. The observed low power depletion (upconversion emission switch off at 455 nm) is attributed to the transfer electrons to side transition pathway to avoid the re-excitation of the emitting state by the 810 nm depletion laser. As a result, the authors applied this new discovered material property for low-power super resolution STED microscopy applications and demonstrated an improved resolution of 66 nm to image single NaYF₄:Yb,Tm nanoparticles.

Developing more efficient photo-switchable luminescent probes for low-power super resolution microscopy is highly important as it is widely known that the current STED microscopy approaches using organic dyes and other luminescent nanoparticles commonly suffer from the high power depletion lasers to achieve sub-diffraction-limit super resolution imaging. Currently, in order to switch each individual pixel on and off for super-resolution imaging, a high-power bulky laser is needed in all the commercial STED system. This ends up with very expensive equipment, typically over \$1 million. And with such a high-powered laser shining on a fragile biological sample, the sample essentially becomes 'cooked'. Significantly reducing the power requirement removes the need for bulky and expensive lasers and makes super resolution microscopy approach much more biocompatible.

The main credit of this work includes observing the optical switch off of 450 nm upconversion emissions using a low power 810 nm depletion laser, and to introduce the emerging luminescent materials, the highly doped upconversion nanoparticles, to solve the key bottleneck issue in super resolution, which is a highly novel approach.

Interestingly, we have a similar work (Amplified stimulated emission in upconversion nanoparticles for super-resolution nanoscopy; doi:10.1038/nature21366) to be published online by Nature on 22nd February 2017 (submitted to Nature on 23rd August 2016; accepted on 4th January 2017). It is fair to regard this work by He and co-workers as an independent study. Three additional experiments by He and his co-workers are highly complementary to our work: 1. Demonstration of multi color super resolution imaging by design and synthesis of Tb based upconversion nanoparticles using the energy migration effect (reported by Xiaogang Liu and co-workers in 2011); 2. Whole spectrum (deep UV and near infrared to infrared) measurement to monitor the upconversion and depletion process; 3. Demonstration of single nanoparticle tracking (super resolution) in living cells.

Before other minor and grammar suggestions, the authors need to thoroughly consider our main points

to improve the quality of this work before it should be eventually accepted by Nature Communication.

1. The mechanism explained by the authors seems incorrect. The observed stimulated depletion should happen from the 3H4 level. It is really the net stimulated depletion surpasses absorption, after the population inversion is established in the highly doped upconversion nanoparticles, and when 810 nm depletion laser is introduced, matching the transition from 3H4 to 3H6 (see the detailed mechanism analysis from our work).

We think fundamentally the authors are confused by the energy levels. The 1D2 level of Tm is the excited state accumulating four 980 nm photons sensitized by Yb, which level is responsible for the 455 nm, 480 nm, 660 nm and 740 nm emissions. There is no transition bandgap for 810 nm absorption from this level. See line 124, page 6, they attributed 1D2 transitions to six-photon upconversion process according to a literature. We checked the literature and found that 5-photon process was involved in Y2O3: 0.2 mol. % Tm³⁺ and 3 mol. % Yb³⁺ in the reference. That's totally different host and doping concentration cases. Actually, the 1D2 population in conventional NaYF₄ nanoparticles (18-20 Yb; <2 mol% Tm³⁺) always attribute to 4-photon process. To verify the 6 photon process, claimed by the authors, we suggest to measure the power-intensity curve to confirm the transition process.

To further exclude the possibility of stimulated depletion (emission), and the side-way emission depletion mechanism described by the authors, the lifetime measurement (Fig S7) still shows the slow decay lifetime (>50 us) when the blue emission at 80% depletion power is applied. It is very obvious that there is no stimulated emission depletion at 455, otherwise, the lifetime should be dramatically reduced to sub microsecond range.

2. the super resolution imaging of nanoparticles on cells is not a convincing demonstration, as it does not show any specific imaging of sub-cellular structures. The typical demonstration, acceptable by super resolution community, should be to show the resolution improvement in imaging the cytoskeleton of a cell, made up of microtubules and actin filaments. The current demonstration of imaging the nanoparticles on the cell does not deliver additional value than the imaging of single nanoparticles.

3. Low frame rate (every 5 second) for single particle tracking experiment needs a bit more work. Additionally, how do we know it is the single nanoparticle instead of the particle cluster? as it is often the case that single monodispersed nanoparticles become aggregates in the biological physiological conditions due to their sticky surface. Also, according to the authors "there is no essential decrease in the intensity", but the intensity of the particles was increased from 0.3 to 0.8, why? The long pixel dwell time (6-10 ms) will lead to relatively long image acquisition times compared to traditional STED (50 -200 μs). So how to avoid the cell damage (photo toxicity) in this case? Additional data may be needed here to prove the cell viability after the long time tracking.

Other minor suggestions:

In the main text, the transitions of two emissions at 700 nm and 1064 nm should be indicated when the single beam 810 nm is applied.

The emission peaks in Fig. 3 should be labelled accordingly with transition notes to facilitate readers. Error bars are needed in figure 2e and in figure 3c.

The samples have different particle sizes, e.g. 22 nm for NaYF₄:18%Yb³⁺,20%Tm³⁺, 18 nm for NaYF₄:18%Yb³⁺,10%Tm³⁺. How to exclude the size effect that affecting the depletion efficiency?

The whole spectra including the NIR part should be given in the supplementary Fig S5.

Can the decay lifetimes of 1800-nm emission at different depletion efficiencies be measured? This will provide more powerful evidence than that of 455 nm.

Supplementary figure S11: "with our experimental data (Fig. 2(b))" should be "with our experimental data (Fig. 2(e))"?

Reviewer #4 (Remarks to the Author):

I read this manuscript with considerable interest, given the implied strong claim on new mechanism for the depletion in STED-like microscopy in the abstract. The idea is to lock electrons at lower energy states (or to let the ion switch between these states, not allowing switching to 455 nm emitting state) to prohibit re-population of the emitting state. To my knowledge, this idea with the use of lanthanide-based UCNPs in STED-like microscopy is new. The idea can indeed potentially impact STED-microscopy by reducing requirement on STED-laser power, in additions advantaging in no photobleaching, and dual- and potentially multi-color STED imaging using single excitation and STED beams. The paper might be of interest not only for super-resolution microscopy field, but also for lanthanide/up-conversion materials.

However, I also noticed several essential drawbacks which preclude the publication unless eliminated. 1. Main claim was not properly supported by experimental data and analysis. 2. The claims are not appropriately discussed in the context of previous literature.

I will discuss them in detail.

1. In the simplest two-level scheme, the depletion efficiency depends on the spontaneous and the stimulated emission (SE) lifetimes of the emitting state. The latter is determined by the SE cross-section and the STED-laser power/focal area. The authors continuously claim that they locked the Tm³⁺ in between lower energy states, thus not allowing Tm³⁺ electrons to return back to the emitting state, basically making the competition between spontaneous and stimulated emission not/less important. This, however, was not properly supported neither by experiment not by analysis.

Examples of authors arguments:

Line 126-127. "Subsequently, the ions at the 3F_{2,3} states quickly decay to the 3H₄ state nonradiatively." How quickly? What is the non-radiative decay life-time? If it is quick, why Fig. 3d demonstrates strong 700 nm and 1064 nm emission bands due to radiative transitions from 3F_{2,3}? What about cross-section for 810-ESA back to 1D₂?

Lines 128-130. "The following cross-relaxation process between Tm³⁺ ions contributes to the concentration of excitation energy into the lower 3F₄ state and eventually dissipates it in the form of infrared emission, efficiently padlocking the repopulation of the 1D₂ state under excitation of the 975 nm beam." I do not buy it unless I see the lifetimes of all the above-mentioned processes.

Simulating 10 rate equations with >20 unknown parameters including the cross-sections and the lifetimes, is not convincing.

It can well be that with low Tm³⁺ concentration (low CR rate), re-population rate of the 1D₂ state by both lasers is stronger than de-excitation by only depletion laser, and that is why you have increase of luminescence at low Tm³⁺ concentrations. And with presence of CR, the depletion efficiency comes to its theoretical maximum based purely on the 1D₂ radiative lifetime and the SE cross section. This might actually be not far from the truth based on my next comment.

2. Lines 119-120. "The saturation excitation intensity I_{sat} (50% emission off) is estimated to be 1.78 MW/cm² (power ~5 mW), which is orders of magnitude lower than that in common STED cases." This is not correct, the claim should be removed. I_{sat} typically amounts for 10 mW in standard STED microscopy (Willig et al, STED microscopy with continuous wave beams, Nat Meth, 2007). Moreover,

for Tb³⁺ and Eu³⁺ 4f-4f transitions P_{sat} is 10-20 mW as well (Alekhin et al, STED properties of Ce³⁺, Tb³⁺, and Eu³⁺ doped inorganic scintillators, Opt express, 2017). It can well be that SE cross section of the Tm³⁺ 1D₂ - 3F_{2,3} transitions and the radiative lifetime of the Tm³⁺ 1D₂ - 3F₄ transition will alone result in $P_{sat} = 5$ mW, allowing 96% depletion at 50 mW. All the speculation about locking the electrons at the lower Tm³⁺ levels will then be invalid. Unless the authors have solid proof for the opposite, the claims should not be present in the publication.

Other critical comments:

3. Lines 64-65. "we propose a novel low-energy state trap assisted mechanism for emission depletion (Fig. 1d)". The previous work of the authors (Wu et al, "Optical depletion mechanism of upconverting luminescence and its potential for multi-photon STED-like microscopy", Opt. express, 2015), where they investigated similar mechanisms of similar class of UCNPs, is not cited. This questions a bit the novelty of the current research.

4. Line 124. "A six-photon upconversion process into 1D₂ level" In Fig 3a, only a four-photon process is illustrated.

5. Lines 178-179. "A factor of 8 improvement to 66 nm". It means that without the depletion beam the resolution was 530 nm. More or less the same is demonstrated in Fig. 4. Theoretically, $R = \lambda/2NA$ is 160 nm for 455 nm emission and 1.35NA objective used in the experiment. How comes then 530 nm resolution?

Minor comments:

6. Lines 46-49. Trying to explain the main super-resolution methods in one sentence confuses more than explains. The statement in line 48 contradicts the principles of cw-STED.

7. Lines 58-63. "...of great significance to create efficient optical depletion pathways". You mentioned light toxicity, photobleaching, but what about STED-parallelization ("Nanoscopy with more than 100,000 'doughnuts'", Nat Meth, Chmyrov, 2013). Faster imaging is also a very good motivation.

8. It would be much easier to follow the articles if the following is implemented:

- Numbering all the relevant transitions and referring to those numbers in the text.
- Perhaps making 1-2 large electron-energy diagrams instead of more smaller ones.
- Lines 50-75. Make text and illustration more consistent with each other. Emitting/dissipating states with S1/S0 in illustration.

All-in-all, the presented idea and the experimental results are of considerable interest and worth for publishing. But either the claims should be reduced, or they should be supported by stronger data/analysis to be published.

This paper entitled “Achieving high-efficiency emission depletion nanoscopy by inhibiting re-excitation to the emitting-state of upconversion nanoparticles” by Zhan et al. describes a very exciting approach to achieve super-resolution microscopy. Through the cross-relaxation in the upconversion nanoparticles, the saturation intensity can be decreased by 2-3 orders of magnitude, leading to super-resolution with low power. This interesting finding has led to the demonstration of 66-nm super-resolution. The authors have further demonstrated that the UCNPs can be used to stain the subcellular organelle, for super-resolution and single particle tracking. Overall, I think the novelty and breath of the manuscript meets well the publication requirements. However, the depth of this work should be further improved, with substantial further experiments, to justify its publication in Nature Communications. Below are my comments/suggestions:

1. According to the depletion equation, $\eta = \frac{1}{1+I_{STED}/I_s}$. The $I_s=1.78\text{MW cm}^{-2}$, and

the STED intensity $=17.7\text{MW cm}^{-2}$, which is 10x the I_s . Based on this, it is unlikely that the intensity can be depleted to 2%, and similarly, the resolution of 66 nm cannot be obtained, with such I_s , as the resolution of STED obeys a square

root law: $d_{STED} = d_c \frac{1}{\sqrt{1+\alpha I_{STED}/I_s}}$. And more importantly, here the $\alpha < 1$, and

I_{STED} is only one quarter that of Gaussian focal spot, as it is modulated to a donut.

Therefore, I encourage the authors to double check the measurement of the saturation intensity.

- a) In page 7, the two statements “Less than 2% residual luminescence can be achieved for $I_{810} = 17.7 \text{ MW/cm}^2$ ” and “when co-irradiated by an 810 nm CW laser beam (17.7 MW/cm^2), the most intense 455 nm blue emission was greatly quenched, with a depletion efficiency approaching 96%” is inconsistent.
2. It is very interesting to see that when the Tm^{3+} concentration is greater than 10% in the 18% Yb doping nanoparticle, the depletion efficiency decreases with the doping concentration.
 - a) Any explanation to this?
 - b) What is the relationship between the doping concentration and the depletion efficiency? Is it related with the cross-relaxation?
 3. The mechanism of stimulated depletion on $^1\text{D}_2$ level is not well supported. The stimulated emission is a very fast process which has lifetime of $\sim\text{ps}$ level. If the stimulated emission is from the highest energy level, one should see very dramatic lifetime change. While in Fig. 2e, the lifetime is from 45 us to 32 us.
 - a) In the methods section, the authors only stated that, the lifetime is measured with a 1KHz chopper, with window of 1ms. How long is the duration of the

pulsed excitation pulse? The effect of the excitation pulse should be carefully measured to avoid any bias to the lifetime measurement.

4. The Fig. 2e should be more carefully studied. Currently, it only tells us that when increasing the 808 nm intensity, for one specific doping concentration, the lifetime decreases. I suggest the authors to give more thorough study on this, by using UCNPs with different doping concentration, to study the effect of stimulated emission to the lifetime. This can also help to understanding the relationship between the doping concentration and the spontaneous emission/stimulated emission.
5. In Fig. S11 a simulated dependency of the fluorescence intensity, and 810 nm laser power has been given. Please correlate this figure with Fig. 2e to show how well the theory matches with the experiment.
6. This work should demonstrate a real biological super-resolution imaging, by specifically labeling the UCNPs on one of the subcellular organelle. The current Fig. 5 a and b are just showing that the UCNPs can enter the cell non-specifically, and super-resolution imaging can see a smaller UCNP spots, but no biological information can be revealed.
7. Likewise, the current Fig. 5i is only traces of the UCNPs, not necessarily any of the cellular molecules. It is only when the UCNP label is proved with other single molecule techniques, that one can claim “single particle tracking for cancer cell”. Specificity in the UCNP labeling should be demonstrated with e. g. correlative study of labeling the organelle with organic dye. Ideally, if this experiment can be performed in super-resolution, it will broaden the application of super-resolution in cellular study significantly.
8. It is unclear to me of the yellow line in Fig. 2d. Why is it curved, like a manually drawing?
9. In my opinion Fig. 1 should be moved to SI, since the mechanism in this work doesn't belong to any of the mechanisms illustrated. I encourage the authors to improve the diagram for better understanding.
10. The authors have claimed that the 1800 nm is significantly enhanced with 975+810nm simultaneous illumination. However, in Fig. 3 the 700nm, 1064 nm, and 1800 nm (black line) are shown just like the sum of the contribution of separate 975nm (red) and 810nm (green). Please explain it.
11. From Fig. 5, it is interesting to see that often when the intensity drops, the velocity increases. Further statistical analysis should be performed on this.

Manuscript No. NCOMMS-17-01804

Title: Achieving high-efficiency emission depletion nanoscopy by **employing interionic interaction** in upconversion nanoparticles

Response Letter to Reviewers

Dear Reviewers,

Thank you very much for your careful consideration and valuable comments on our manuscript, which have helped us significantly improve the manuscript. In the following, we provide a point-by-point response to the comments, together with the corresponding changes in the manuscript. As below, the reviewers' comments are written in **black** and our responses to them in **blue**. The important amendments or changes to the manuscript are given after the response in **red**.

Reviewer 1:

General comments:

Zhan *et al.* report upconversion nanoparticles (UCNPs) with controllable emission for use in super-resolution STED nanoscopy by using 975 nm excitation and 810 nm depletion. They have also included an elaborate analysis of the switching mechanism. I am not an expert on lanthanide photoluminescence; therefore, I shall not comment on the mechanistic issues but rather focus on the application to microscopy.

Response: We thank the reviewer for his/her general comments on our work.

Specific comments:

(1) Although I find the presented scheme of luminescence switching interesting, I cannot recommend publication in Nature Communications because the manuscript does not make a convincing case for clear advantages of using these UCNPs for imaging.

Response: We thank the reviewer for the comment regarding the advantages of UCNPs. During the last decade, lanthanide doped UCNPs have been developed as an important group of luminescent biomarkers. Compared to traditional fluorophores, they have many outstanding advantages as imaging contrast agents, including excellent chemical-/photo-stability, nil autofluorescence noise, and biocompatible and high penetrating excitation/emission wavelengths [1]. Particularly, the non-photobleaching and non-photoblinking properties of UCNPs are highly desired for STED imaging, which enables long term and repetitive imaging with high temporal resolution [2].

Photobleaching has remained a severe issue in microscopic imaging when using organic fluorophores, especially in STED imaging where generally high depletion laser intensity is required to achieve high spatial resolution [3, 4]. This issue precludes long-term, continuous and repetitive imaging. Although many efforts

have been devoted and progress has been made, *e.g.*, the development of the so-called “protected STED” technique pushing the fluorophores into a second off state [5, 6] and faster laser scanning strategy [7], this issue has not been adequately and fundamentally resolved. Photoblinking of fluorophores is another issue that needs to be addressed, which often bothers some non-photobleaching fluorophores, such as quantum dots [8], and limits the temporal resolution of imaging. Compared to organic fluorophores and other nanocrystals, UCNPs are completely photobleaching- and photoblinking-free, as demonstrated in the present study (Figure 3, Figure S14 in the revised manuscript) and in other reports [2, 9], suitable for long term and repetitive imaging and nanoparticle tracking studies.

Interestingly, an independent work with a similar topic was published in Nature very recently (“Amplified stimulated emission in upconversion nanoparticles for super-resolution nanoscopy”, Nature **2017**, 543(7644), 229-233) [10]. As demonstrated in this report and in the present study, the significantly reduced requirement on depletion power density removes the need for bulky and expensive pulse lasers and makes super-resolution microscopy approach much more biocompatible. The large shifts among the wavelengths (Ex/STED/Em: 975/810/455 nm) also simplify the optical filtering and enhances the detection sensitivity without any emission spectral waste, which is inevitable in traditional STED [11]. By combining the efficient multiphoton emission (MPE) of UCNPs with STED, the upconversion MPE-STED is also advantageous over single photon emission STED for deep tissue studies in turbid biosamples, since the scattering of the NIR excitation/depletion light is much less significant [12]. In addition, the nonlinearity of UCNPs brings higher signal-background-noise ratio in MPE-STED [13]. In view of this, we believe it is meaningful to develop UCNPs as a group of luminescent probes for STED imaging, providing alternatives to organic dyes and other luminescent nanoparticles.

Change in the manuscript:

We have added more discussion in the Introduction and Conclusion sections to highlight the advantages of using UCNPs for bioimaging, particular for STED imaging, correlated with our experimental results.

Sentences added in Introduction section (lines 44-49, page 3): “Lanthanide UCNPs have been developed as an important group of luminescent biomarkers during the last decade. UCNPs convert low-intensity near-infrared (NIR) excitation to shorter-wavelength NIR, visible and ultraviolet (UV) emissions. Such unique luminescence properties of UCNPs enable superior bioimaging without many of the constraints associated with conventional optical biomarkers, including photobleaching, photoblinking, tissue autofluorescence, limited imaging depth and high light toxicity^{24, 25, 26}.”

Sentences added in Conclusion section (lines 267-271, page 12): “The upconversion emission depletion nanoscopy developed here significantly reduces the requirement of the depletion power density, as also

demonstrated in a very recent similar work⁴⁷, and removes the need for bulky and expensive pulse lasers and makes STED microscopy more biocompatible, providing alternatives to traditional STED imaging using organic dyes and other luminescent nanoparticles.”

Figure numbering changed: The numbering of Fig. 4d (nonbleaching demonstration of STED), Figs. 5i-j (continuous nanoparticle tracking), and Fig. S12 (nonblinking demonstration in STED) in the original manuscript have been changed to Fig. 3d, Figs. S16 d-f, and Fig. S14, respectively, in the revised manuscript.

(2) First of all, they are huge (18/19/22/28 nm as reported in the MS and Fig. S1) in comparison with typical proteins, which severely limits their use as biomarkers.

Response: We thank the reviewer for the comment on the nanoparticle size. It is true that the scales 18/19/22/28 nm is bigger than typical proteins. We also agree with that smaller nanoparticle would be better for bioimaging applications. We would like to bring up that many efforts have been devoted to regulating the synthesis to obtain UCNPs with tunable sizes, in order to address the nanoparticle size concern. To date, UCNP synthesis protocols have been well established, making bright sub-10 nm UCNPs easily accessible [14-18]. Such size is comparable to some typical proteins (~10 nm) [19, 20].

In the revised manuscript, as requested, we have modified our protocol and successfully synthesized UCNPs ($\text{NaGdF}_4:18\% \text{Yb}^{3+}, 10\% \text{Tm}^{3+}$) of much smaller size, with an average diameter of 11.8 nm, as shown in Figure R1 (in this Response Letter). The size of these nanoparticles is smaller or at least not larger than other nanocrystals that have been employed in super-resolution imaging, including quantum dots (~12 nm) [8], nanodiamonds (~34 nm) [21] and gold nanoparticles (~100 nm) [22-24]. The smaller sized nanoparticles, after surface modification and antibody bioconjugation, are used in cellular imaging study.

Change in the manuscript:

The following figure and the corresponding synthesis method and characterization have been added.

Figure R1 (Fig. S5(b) in the revised Supplementary Information) TEM images of newly synthesized $\text{NaGdF}_4:18\% \text{Yb}^{3+}, 10\% \text{Tm}^{3+}$ UCNPs with an average diameter of 11.8 nm.

(3) Second, like other nanocrystals, UCNPs are difficult to conjugate to interesting biomolecules.

Response: Inorganic nanocrystals including UCNPs have been developed as an important category of biomarkers and are extensively used in life science studies due to their excellent physicochemical properties (such as good photostability), providing alternatives and complementary to organic fluorescent dyes and proteins. Bioconjugation of nanocrystals with biomolecules is generally required for their biological applications such as targeted imaging, immunolabelling and biosensing and thus always remains a key topic. After persistent efforts of numerous researchers, many effective protocols about nanocrystal surface modification and functionalization have been developed [20, 25], leading to easy access to water-dispersible and biocompatible nanocrystals decorated with reactive groups suitable for subsequent bioconjugation to biomolecules. Specific to UCNPs, many surface modification strategies have been established, including ligand exchange [26-28], ligand oxidation [29, 30] and surface silanization [31-33], and subsequent bioconjugation with biomolecules such as antibody [26], peptide [34, 35], folic acid [36], DNA/RNA have been demonstrated [37, 38].

In the revised manuscript, as requested, we have made a lot of effort in the super-resolution bio-imaging using UCNPs and finally achieved and updated the biological super-resolution imaging data in Figure 4 in the revised manuscript. We have successfully conjugated the 11.8 nm UCNPs with secondary antibody (goat anti-rabbit IgG antibody) and achieved high quality immunolabelling of the cytoskeleton protein desmin of HeLa cancer cell incubated with primary antibody (Anti-desmin antibody (rabbit)), imaged by the upconversion nanoscopy technique developed in this study, as shown in Fig. R2. The line profile analysis for a local area indicates that the lateral imaging resolution is increased to 82 nm, less than $\lambda/10$ ($\lambda_{\text{ex}}=975$ nm) (Fig. R2). Cytoskeleton protein labeling with inorganic nanoparticles is generally challenging and had never been achieved using UCNPs previously. To the best of our knowledge, this is the first demonstration of immunolabelling of fine subcellular structures as well as the first demonstration of super-resolution cellular imaging using UCNPs. We have updated the biological super-resolution imaging data using the newly obtained immunolabelling results in Figure 4 in the revised manuscript.

Change in the manuscript:

Sentences added (lines 235-250, page 11): “Finally, we imaged the cytoskeleton of HeLa cancer cells with UCNP-mediated nanoscopy. High Tm^{3+} -doping UCNPs ($\text{NaYF}_4:18\% \text{Yb}^{3+}, 10\% \text{Tm}^{3+}$) with an average diameter of $11.8(\pm 2.2)$ nm were synthesized (Fig. S5). After surface modification with poly(acrylic acid) (PAA) (Fig. S19), these UCNPs were conjugated with the secondary antibody (goat anti-rabbit IgG antibody) (Fig. S20). Fixed HeLa cancer cells were incubated with anti-desmin monoclonal antibody (cytoskeleton marker)⁴⁶, and then stained with the secondary antibody IgG conjugated UCNPs. Such immunostained HeLa cells were imaged in the super-resolution mode under 975-nm and 810-nm co-irradiation, and with a reference

under irradiation of just the 975-nm laser. As shown in Fig. 4, subcellular structures were clearly imaged, and a significant improvement in the spatial resolution was achieved with the assistance of the 810-nm depletion laser beam (Figs. 4a-4n). To evaluate the resolution of STED imaging, line profiles of several areas were extracted and fitted with Gaussian functions. The results reveal that the lateral imaging resolution has been improved to ~82 nm, less than $\lambda/10$ ($\lambda_{ex} = 975$ nm) (Fig. 4c-4e). To the best of our knowledge, this is the first demonstration of successful immunolabelling of fine subcellular structures as well as the first demonstration of super-resolution cell imaging using UCNP.

Figure R2 (Fig. 4 in the revised manuscript) Immunofluorescence labeling of cellular cytoskeleton protein desmin with antibody conjugated UCNPs and super-resolution imaging. (a) The multiphoton imaging under 975 nm excitation of some cytoskeleton structures and desmin proteins in HeLa cancer cells incubated with anti-desmin primary antibody and immunostained with UCNP (~11.8 nm in diameter) bioconjugated with goat Anti-rabbit IgG secondary antibody. (b) The same region with (a) imaged in the super-resolution mode (975 nm excitation and the 810 nm STED laser beam), Scale bar: 2 μm. (c-n) Magnified areas selected from a, b (marked by white dotted squares) and line profile analyses; Images in c, f, i and l are taken from the white dotted squares in a; Images in d, g, j and m are taken from the white dotted squares in b; (e, h, k, n) Line profiles analyses of several areas indicated by arrow heads in c and d, f and g, i and j, and l and m, respectively.

(4) Third, the use of near-IR light means that there is a clear disadvantage for achieving high resolution, and even the reported 66 nm (which I question) are only slightly better than what can be achieved with linear SIM. I will comment on the quoted resolution enhancement below. Moreover, the UCNP luminescence lifetimes are

4 orders of magnitude greater than those of the usual fluorophores, which limits the emission intensity. All of these issues should be thoroughly discussed and compared with established approaches to present a well-balanced account.

Response: We agree that NIR light is not advantageous for achieving high resolution in normal confocal microscopy in comparison with using shorter excitation wavelengths, due to the diffraction limit. However, in STED imaging, the resolution is not strictly limited by the wave nature of light but instead depends on the efficiency of the optical depletion [12]. The disadvantage of relatively long wavelength of the NIR light can be compensated by increasing the depletion laser intensity, as the resolution is scaled with the depletion intensity [39]. In addition, the nonlinearity of multi-photon upconversion luminescence will also compensate the resolution loss, as in the case of coherent multiphoton fluorescence microscopy. It should also be noted that the reported resolution of ~66 nm (achieved with 18 nm UCNPs) in the original manuscript is not the theoretical limit. By optimizing the imaging system, increasing depletion intensity and employing smaller nanoparticles, the resolution can be further improved. Regarding the comparison of lateral resolution, STED technique is advantageous over the linear SIM, which could theoretically achieve two-fold enhancement in lateral imaging resolution compared to traditional fluorescence microscopy [40, 41]. In the case of linear SIM, an excitation wavelength as short as 292 nm may be required in order to achieve an equivalent lateral resolution reported here (with a similar objective with $NA=1.35$, $d=0.61\lambda/NA$). Such a short wavelength would cause a lot of issues in the imaging and is impractical. It is worthwhile mentioning that near-IR light enables larger imaging depth compared to the strongly scattered visible light in the turbid biological tissue [42].

We agree that the relatively long lifetime of luminescence of lanthanides (μ s-ms) limits the emission intensity of single emitting centers (*i.e.*, single ions). However, the large number of emission centers within a single nanoparticle, typically in order of a few thousands [43], will greatly compensate the low emission rate of single ions. It is also worth mentioning that the quantum efficiency of UCNPs is several orders of magnitude higher than that of traditional multiphoton fluorophores [42, 44], which are being extensively utilized in biomedical imaging.

In addition, it is worthwhile to mention that the signal-to-background ratio is the most critical parameter for high quality imaging. UCNPs enable autofluorescence-free imaging, inherently guaranteeing a high signal-to-noise ratio and high imaging sensitivity [45-47]. With the advance of upconversion nanochemistry in the last decade, sufficiently bright UCNPs can be readily synthesized, and single nanoparticle detection sensitivity using sub-10 nm nanoparticles has been achieved in this field [14, 48].

Change in the manuscript:

Sentences added in the Introduction section (lines 44-49, page 3): “Lanthanide UCNPs have been developed as an important group of luminescent biomarkers during the last decade. UCNPs convert low-intensity near-infrared (NIR) excitation to shorter-wavelength NIR, visible and ultraviolet (UV) emissions. Such unique luminescence properties of UCNPs enable superior bioimaging without many of the constraints associated with conventional optical biomarkers, including photobleaching, photoblinking, tissue autofluorescence, limited imaging depth and high light toxicity^{24, 25, 26}.”

Sentence added (lines 196-197, page 9): “The spatial resolution can likely be further improved by optimizing our lab-made super-resolution imaging system.”

I have a number of major and minor specific points of criticism that should be addressed before resubmitting this manuscript, possibly to a well-edited specialty journal (J. Luminescence, J. Microscopy etc.). Major points of criticism:

(5) The Line 97, Fig. S1. How were the sizes determined? No size distributions are reported, no errors and statistics are provided. The same applies to the water soluble UCNPs, which have not been characterized physicochemically (spectra, brightness, DLS etc.).

Response: The average diameter of nanoparticles was determined by sampling hundreds of nanoparticles from the TEM raw data. As requested, we have added size distributions, errors and statistics in TEM images in the revised manuscript and supplementary information, as shown in Figure R3.

As requested, we have also added physicochemical characterization data (spectra, brightness, and DLS) for water dispersible nanoparticles used in the experiments to the supplementary information, as shown in Figs. R4 and R5.

Change in the manuscript: Size distributions, errors and statistics have been added in TEM images in Figure 1a in the revised manuscript and Figures S4-S5 in the revised supplementary information. Physicochemical characterization data for water dispersible nanoparticles are added as Figures S15 and S19 in the supplementary information.

Figure R3 (Fig. S4 in the Supplementary Information) **Transmission electron microscopy (TEM) images and Selected area electron diffraction (SAED) patterns of the as-synthesized UCNPs.** (a) The SAED pattern of the $\text{NaYF}_4:18\% \text{Yb}^{3+}/10\% \text{Tm}^{3+}$ used for optical depletion and super-resolution imaging, matching a hexagonal NaYF_4 lattice (JCPDS file number 16-0334). (b) TEM image of the as-prepared low Tm^{3+} -doping UCNPs samples: $\text{NaYF}_4:18\% \text{Yb}^{3+}/0.5\% \text{Tm}^{3+}$ (average size: 19.5 ± 2.7 nm in diameter). (c) TEM image of the as-prepared high Tm^{3+} -doping UCNPs samples: $\text{NaYF}_4:18\% \text{Yb}^{3+}/20\% \text{Tm}^{3+}$ (average size: 22.5 ± 2.7 nm in diameter) (d) TEM image of the as-prepared core $\text{NaGdF}_4:40\% \text{Yb}^{3+}/10\% \text{Tm}^{3+}$, average size: 15.7 ± 1.9 nm in diameter. (e) TEM image of the as-prepared core-shell $\text{NaGdF}_4:40\% \text{Yb}^{3+}/10\% \text{Tm}^{3+} @ \text{NaGdF}_4:15\% \text{Tb}^{3+}$ UCNPs, average size: 28.0 ± 2.4 nm in diameter. (f) The SAED pattern of the $\text{NaGdF}_4:40\% \text{Yb}^{3+}/10\% \text{Tm}^{3+} @ \text{NaGdF}_4:15\% \text{Tb}^{3+}$, matching a hexagonal NaGdF_4 lattice (JCPDS file number 27-0699). The scale bars in TEM images are 50 nm.

Figure R4 (Fig. S15 in the revised Supplementary Information) **Characterization of HCl-treated ligand free UCNPs.** (a) Fourier transform infrared spectroscopy (FTIR) spectra of (i) OA-UCNPs and (ii) Ligand free-UCNPs. (b) TEM image of the ligand free-UCNPs (Scale bar: 20 nm). (c) The size distribution of the ligand free-UCNPs measured

by DLS. (d) Emission spectra and brightness comparison between the OA-UCNPs and the ligand free-UCNPs. FTIR spectra confirm the successful removal of OA ligands on the surfaces of ligand-free nanoparticles. The bands at 1552 cm^{-1} and 1460 cm^{-1} OA-UCNPs were associated with the asymmetric (ν_{as}) and symmetric (ν_s) stretching vibration of $-\text{COOH}$ groups, and disappeared in the spectrum of ligand free-UCNPs. The average diameter (ca. 26 nm) obtained from the DLS measurement is larger than that determined by TEM measurement ($18.5\pm 1.8\text{ nm}$), because DLS measured the hydrodynamic diameters of nanoparticles. As shown in (d), no significant change in the emission spectrum and brightness of the ligand-free UCNPs in comparison with the OA-UCNPs, indicating that the removal of oleate ligand did not significantly affect the optical properties of the nanoparticles.

Figure R5 (Fig. S19 in the revised Supplementary Information) Characterization of COOH-functionalized hydrophilic PAA-UCNPs. (a) Fourier transform infrared spectroscopy (FTIR) spectra of (i) OA-UCNPs and (ii) PAA-UCNPs. (b) TEM image of PAA-UCNPs (Scale bar: 5 nm). (c) Size distribution of PAA-UCNPs measured by DLS. (d) Emission spectra and brightness comparison between the OA-UCNPs and the PAA-UCNPs. FTIR spectra confirm the successful ligand exchange on the surface of nanoparticles. OA-UCNPs exhibit a broadband at about 3433 cm^{-1} and a weak band at 1738 cm^{-1} , associated with the $-\text{COOH}$ stretching vibration, which suggests the presence of trace amount of oleic acid on the surfaces of nanoparticles. After reaction with poly (acrylic acid) at a reactively high temperature, the oleic acids were replaced, and these bands are significantly enhanced and shifted to 3389 cm^{-1} and 1647 cm^{-1} , respectively. In addition, the peak at 1103 cm^{-1} , attributed to the C-O stretching mode, also increased and was shifted to 1095 cm^{-1} . These features indicate that a large amount of $-\text{COOH}$ groups present on the surface of nanoparticles. Furthermore, the peak at 721 cm^{-1} , associated with the in-plane rocking vibration mode of $-(\text{CH}_2)_n-$ ($n > 4$), disappeared in the PAA-UCNPs. Based on these results, it can be inferred that the oleic acids on the surface of UCNPs have been successfully replaced with PAA ligands. The PAA-UCNPs exhibited an average hydrodynamic diameter of 24 nm, obtained by DLS measurement, which is larger than that observed from TEM measurement ($18.5\pm 1.8\text{ nm}$).

(6) Line 178 “...lateral imaging resolution reaches 66 nm...”. This number, which is also used in the abstract, is essentially meaningless. The cut through the image is very noisy, and just selecting the narrowest feature is not scientifically sound. There are sophisticated, Fourier-based algorithms to determine resolution. At least, one would expect statistics over a large number of cross-sections, including a statistical analysis. In fact, the FWHM of the neighboring UCNP is already much larger. Also, it appears that there is a broad halo of non-depleted low-resolution background intensity, which is suppressed in the image.

Response: We thank the reviewer very much for the comments on the imaging resolution. We agree that the resolution can be better determined by performing a statistical analysis over number of cross-sections, if standard luminescent samples, *e.g.*, monodispersed nanobeads with exactly the same physical dimensions, are being imaged. However, in our case, the nanoparticles, having a size distribution, were spread on a pre-treated glass slide by spin-coating, where nanoparticle clusters can also be formed. The size variation in nanoparticles and in nanoparticle clusters can give rise to significantly different FWHMs in different cross-sections in the image. Since the imaging resolution of an imaging system, characterizing the spatial resolving power, refers to the finest structure that the system is able to distinguish, we select the finest details to evaluate the resolution of the system.

In addition, we think it is a bit out of the scope of our study to determine the imaging resolution very precisely through using some sophisticated algorithms. Instead, we employed a routine method based on the FWHM analysis, which is well accepted by the super-resolution imaging community [8, 10]. In order to minimize the influence of the noise, we have performed fitting when extracting the FWHMs (Fig. R6), as shown in Figures 3c and 4e in the revised manuscript.

The apparent halo was due to background noise.

Change in the manuscript: In order to minimize the influence of the noise, we have performed Gaussian fitting when extracting the FWHMs.

Figure R6 (Fig. 3c in the revised manuscript) Line profiles of the images and resolution analysis, fitted with Gaussian functions

(7) The authors claim that the cell imaging studies reveal tracks of single nanoparticles. I have severe doubts that this is true. It was not characterized how these particles enter the cell. Most likely, they were endocytosed, which means that multiple particles reside in one vesicle. This is also suggested by the apparently greater size dispersion in the image. Again, there is only a single cross section shown. Without appropriate statistics, this is meaningless.

Response: We thank the reviewer very much for the comments on the particle tracking experiment. We agree that most likely the nanoparticles were endocytosed and thus multiple particles resided in one vesicle [49]. In order to provide a presentation on our most important advances, the results of the nanoparticle tracking study were moved to the supplementary information. In addition, we have reduced the claim from “single particle tracking” to “nanoparticle tracking”. Despite this, we would like to emphasize that the significance of the particle tracking study was the demonstration of the capacity of UCNPs for long-term and continuous bioimaging with super-resolution, due to the excellent photostability of these nanoparticles. Long-term and continuous imaging has been a challenge when using organic fluorescent probes.

Change in the manuscript: We have moved the results of nanoparticle tracking study to the supplementary information as Figure S16 and reduced the claims accordingly.

(8) The manuscript and especially the supplement could greatly benefit from careful editing. Examples of mistakes from the supplement: “ 2π -heli phase”, “photon multiple tube (PMT)”, “muliphoton laser scanning”, no units given in Table S1.

Response: We are sorry for the mistakes in the original manuscript and supplementary information. We have made careful editing and corrections throughout the manuscript and supplement.

Changes in the manuscript: “ 2π -helix phase”, “photomultiplier tube (PMT)”, “multiphoton laser scanning”, etc. Units are given in Table S1

(9) Line 19: “...which generally requires dangerously high light intensity...” What is “dangerously” supposed to mean? I find it completely inappropriate here. Moreover, the authors use 17.7 MW/cm^2 to reach 96% depletion, corresponding to 50 mW (at the sample?), if I understand their data correctly. Is that dangerous as well in the view of the authors? How did they calculate the power density?

Response: We have removed inappropriate statements in the revised manuscript.

In our study, the laser power was measured at the front aperture of the objective. The diameter of the depletion laser beam profile at the focal spot was measured by a CCD camera. The power density was calculated to get the power per area, taking into account the donut shape of the depletion laser beam. We repeated the measurement of the saturation intensity, giving rise to 50% emission off, and the saturation intensity is determined to be 849 kW/cm^2 . This is significantly smaller than that used in traditional STED (1–240 MW

cm²) using fluorescent dyes such as ATTO dyes, Alexa Fluor, semiconductor quantum dots and nanodiamonds with nitrogen vacancies as contrast agents [8, 21].

Change in the manuscript:

Sentence modified (lines 15, page 2): “..., which generally requires a **relatively** high light intensity to cause adequate depletion efficiency.”

(10) Line 45: SIM works in the realm of diffraction, as long as it is used in a linear way (REF 18). The sentence should be stated clearly. Also, MINFLUX should be mentioned as a new super-resolution approach.

Response: We thank the reviewer for pointing out our inaccurate description regarding the structured illumination microscopy technique. We agree that linear SIM works in the realm of diffraction. We have modified the text accordingly in the Introduction section in the revised manuscript. In addition, we have included MINFLUX as a representative super-resolution approach in the Introduction section and added some relevant reference [50].

Change in the manuscript:

Sentences modified (lines 37-47, page 3): “Exploitation of optical switching to control the transition of luminophores between two optically distinguishable states (on/off) is an exciting and popular way to circumvent the diffraction limit^{13, 14}. This approach has led to many far field super-resolution fluorescence microscopy techniques, such as stimulated emission depletion microscopy (STED)^{15, 16, 17}, single molecule localization nanoscopy (PALM/STORM, MINFLUX)^{18, 19, 20}, and saturated structured illumination microscopy^{21, 22}, which breaks the diffraction limit and allow for imaging at a spatial resolution of down to a few nanometers. **A major category** of these nanoscopy methods, **including pulse STED and PALM/STORM**, provides sub-diffraction resolution essentially by transiently switching the luminophores between the bright “on” and dark “off” states, causing those within the same diffraction range to emit successively, rather than simultaneously²³.”

Reference added:

Balzarotti, F. *et al.* Nanometer resolution imaging and tracking of fluorescent molecules with minimal photon fluxes. *Science*, doi:10.1126/science.aak9913 (2016).

(11) Lifetimes should be reported on semi-log plots (Fig. 3e, Fig. S7).

Response: Lifetime data have been presented on semi-log plots as suggested, as seen in Figure 2c and Figure S6 in the revised manuscript.

Reviewer 2:

General comments:

The current manuscript describes a strategy for high-efficiency emission depletion by introducing auxiliary matter-matter interaction inside the highly doped upconversion nanoparticles. The observed low power depletion (upconversion emission switch off at 455 nm) is attributed to the transfer electrons to side transition pathway to avoid the re-excitation of the emitting state by the 810 nm depletion laser. As a result, the authors applied this new discovered material property for low-power super resolution STED microscopy applications and demonstrated an improved resolution of 66 nm to image single NaYF₄:Yb,Tm nanoparticles.

Developing more efficient photo-switchable luminescent probes for low-power super resolution microscopy is highly important as it is widely known that the current STED microscopy approaches using organic dyes and other luminescent nanoparticles commonly suffer from the high power depletion lasers to achieve sub-diffraction-limit super resolution imaging. Currently, in order to switch each individual pixel on and off for super-resolution imaging, a high-power bulky laser is needed in all the commercial STED system. This ends up with very expensive equipment, typically over \$1 million. And with such a high-powered laser shining on a fragile biological sample, the sample essentially becomes 'cooked'. Significantly reducing the power requirement removes the need for bulky and expensive lasers and makes super resolution microscopy approach much more biocompatible.

The main credit of this work includes observing the optical switch off of 450 nm upconversion emissions using a low power 810 nm depletion laser, and to introduce the emerging luminescent materials, the highly doped upconversion nanoparticles, to solve the key bottleneck issue in super resolution, which is a highly novel approach.

Interestingly, we have a similar work (Amplified stimulated emission in upconversion nanoparticles for super-resolution nanoscopy; doi:10.1038/nature21366) to be published online by Nature on 22nd February 2017 (submitted to Nature on 23rd August 2016; accepted on 4th January 2017). It is fair to regard this work by He and co-workers as an independent study. Three additional experiments by He and his co-workers are highly complementary to our work: 1. Demonstration of multicolor super resolution imaging by design and synthesis of Tb based upconversion nanoparticles using the energy migration effect (reported by Xiaogang Liu and co-workers in 2011); 2. Whole spectrum (deep UV and near infrared to infrared) measurement to monitor the upconversion and depletion process; 3. Demonstration of single nanoparticle tracking (super resolution) in living cells.

Before other minor and grammar suggestions, the authors need to thoroughly consider our main points to improve the quality of this work before it should be eventually accepted by Nature Communications.

Response: We appreciate the reviewer for his/her positive evaluation and recommendation on our work. We also acknowledge the reviewer for letting us notice their interesting work with a similar topic and his/her objective judgement that our work is an independent study from theirs.

Specific comments:

(1) The mechanism explained by the authors seems incorrect. The observed stimulated depletion should happen from the $^3\text{H}_4$ level. It is really the net stimulated depletion surpasses absorption, after the population inversion is established in the highly doped upconversion nanoparticles, and when 810 nm depletion laser is introduced, matching the transition from $^3\text{H}_4$ to $^3\text{H}_6$ (see the detailed mechanism analysis from our work).

Response: We thank the reviewer very much for the comments on the mechanism of the optical depletion of 455 nm upconversion luminescence. After carefully studying the experimental results and mechanism reported in the Nature paper by the reviewer and coworkers (Ref. 10) and comparing them with ours shown in the present study, we realize that different mechanisms probably prevails in these two studies, based on the discrepancies in some key experimental results in these two independent works obtained from different nanoparticles synthesized in different labs, as summarized below:

- a. In our study, an intensity increase of the 700 nm red emission ($^3\text{F}_3 \rightarrow ^3\text{H}_6$) was observed in UCNPs including those with high (10%) and low (0.5%) Tm^{3+} -doping concentrations when co-irradiated by an 810 nm CW laser beam (Figure R7), which suggests the occurrence of the stimulated emission process $^1\text{D}_2 \rightarrow ^3\text{F}_2$ that directly populates the $^3\text{F}_2$ state, which would quickly decay to the $^3\text{F}_3$ state.
- b. In our study, the lifetime of the 455 nm emission was decreased with the addition of the depletion laser at 810 nm in all UCNPs with different Tm^{3+} concentrations, referring to Figure 2c and Figure S6 in the revised manuscript.
- c. We observed a significant enhancement of the 455 nm emission in low Tm^{3+} -doping samples (Figure R7(a)) when adding the 810 nm laser.
- d. In our study, a pronounced enhancement of the emission for the 1470 nm band ($^3\text{H}_4 \rightarrow ^3\text{F}_4$) was observed in both high and low Tm^{3+} -doping UCNPs (Fig. R8), indicating the population increase of the $^3\text{H}_4$ state after the addition of the 810 nm laser beam.

Figure R7 (Figs. 1b and 1e in the revised manuscript) Upconversion emission spectra of 0.5% Tm³⁺ (a) and 10% Tm³⁺ (b) Tm-UCNPs.

Figure R8 The intensity change of 1470 nm and 1800 nm emission bands of NaYF₄:18%Yb³⁺/10%Tm³⁺ (a, Fig. 2d in the revised manuscript) and NaYF₄:18%Yb³⁺/0.5%Tm³⁺ (b, Fig. S9 in the revised Supplementary Information) UCNPs under 975 nm excitation with/without 810 nm irradiation.

In the present study, all spectroscopic data, including the emission enhancement/inhibition for different emission bands, concentration dependence, and lifetime data, presented in Figures 1 and 2 in the revised manuscript, strongly support a formulation of the mechanism for the observed optical depletion of 455 nm upconversion luminescence caused by the 810 nm laser beam (Fig. R9):

- The addition of the depletion laser beam at 810 nm has dual action on the 455 nm upconversion luminescence generated by the 975 nm excitation laser: **a depletion effect** by de-exciting the ¹D₂ state via a stimulated emission process (¹D₂ → ³F₂), and **an enhancement effect (or synergistic excitation)** by populating the intermediate ³H₄ state followed by re-populating the ¹D₂ state with the assistance of the 975 nm beam. The matching of the 810 nm laser light with the transition of ¹D₂ → ³F₂ is supported by

many previous reports [51-54], which will be discussed in detail in our response to Comment 2 in the following. It should be noted that this transition was often overlooked in the literature and its contribution was assigned to other transitions, such as ${}^3\text{H}_4 \rightarrow {}^3\text{H}_6$ and ${}^1\text{G}_4 \rightarrow {}^3\text{H}_5$, due to the spectral overlap among these three transitions [51-53, 55].

- The net change of the 455 nm upconversion luminescence is determined by the **competition between the depletion effect and the enhancement effect**.
- With high Tm^{3+} concentrations, **the highly efficient CR1 process** (${}^3\text{H}_4 + {}^3\text{H}_6 \rightarrow {}^3\text{F}_4 + {}^3\text{F}_4$) [56-58] depicted in Figure R9 greatly suppresses the synergistic excitation effect and simultaneously amplify the depletion effect of the 810 nm beam by transferring the electrons at ${}^3\text{H}_4$ state to the ${}^3\text{F}_4$ state, where an IR energy dissipating channel is introduced by emitting ~ 1800 nm radiation. This two-fold effect of the CR1 process make the re-population rate of the ${}^1\text{D}_2$ state by both lasers to be weaker than the de-excitation rate by the depletion laser, leading to inhibition of the 455 nm luminescence.
- With low Tm^{3+} concentrations, the absence or inefficiency of the CR1 process makes the re-population rate of the ${}^1\text{D}_2$ state by both lasers dominate over the de-excitation rate by the depletion laser, leading to enhancement of the 455 nm luminescence.

Figure R9 (Figs. 2a-2b in the revised manuscript) Proposed optical emission depletion mechanism for the 455 nm upconversion band of Tm^{3+} of $\text{NaYF}_4:18\% \text{Yb}^{3+}/10\% \text{Tm}^{3+}$ nanoparticles.

Changes in the manuscript:

Accordingly, we have modified the title of our manuscript and our claims and description about the mechanism in the revised version (Pages 7-10 in the revised manuscript).

New title: Achieving high-efficiency emission depletion nanoscopy by employing interionic interaction in upconversion nanoparticles

(2) We think fundamentally the authors are confused by the energy levels. The 1D_2 level of Tm^{3+} is the excited state accumulating four 980 nm photons sensitized by Yb^{3+} , which level is responsible for the 455 nm, 480 nm, 660 nm and 740 nm emissions. There is no transition bandgap for 810 nm absorption from this level.

Response: We apologize for a typo in the original manuscript that assigned the population of the 1D_2 state to a six-photon process. We hold the view that the 1D_2 state is populated through a five-photon process under single 975 nm excitation, which is well supported in the literature [54, 59-62]. In this process, one energy transfer step from Yb^{3+} ion to Tm^{3+} ion populates the 3H_5 level of Tm^{3+} from 3H_6 . The 3H_5 decays to the 3F_4 level. A second energy transfer step raises the Tm^{3+} ion from 3F_4 to 3F_2 that quickly decays to 3H_4 . Subsequently, a third transfer step raises the Tm^{3+} ion from 3H_4 to 1G_4 . A fourth energy transfer step from Yb^{3+} ion to Tm^{3+} ion may also take place to populate the Tm^{3+} ion from 1G_4 to 1D_2 , which is however usually less efficient due to the relative large energy mismatch ($\lambda(^1G_4 \rightarrow ^1D_2)$: around 1500 nm) [52]. An alternative, more efficient way to populate 1D_2 is through cross relaxation between two adjacent Tm^{3+} ions with two main routes: $^3H_4 + ^1G_4 \rightarrow ^3F_4 + ^1D_2$ and $^3H_4 + ^1G_4 \rightarrow ^1D_2 + ^3F_4$, exhibiting a five-photon process [61-63].

We agree with the reviewer that the 1D_2 level is responsible for the 455 nm ($^1D_2 \rightarrow ^3F_4$), 660 nm ($^1D_2 \rightarrow ^3H_4$) and 740 nm emissions ($^1D_2 \rightarrow ^3F_3$). However, we think it is the 1G_4 state that is responsible for the emissions around 480 nm ($^1G_4 \rightarrow ^3H_6$), consistent with previous reports [48, 64].

We agree that 810 nm light does not match with the bandgap of $^1D_2 \rightarrow ^3F_3$. However, the matching of the $^1D_2 \rightarrow ^3F_2$ transition with light near 800 nm is supported by many previous reports [51-54]. According to the theoretical calculations and experimental measurements under ultralow temperature (~ 10 K), the $^1D_2 \rightarrow ^3F_2$ transition is centered around 800 nm and could cover quite a few nanometers [65-68]. Furthermore, the FWHM of $^1D_2 \rightarrow ^3F_2$ transition spectrum would be broadened under ambient temperature (~ 300 K) and transient high temperature (under illumination of tightly focused light) [69]. In addition, in our experiments the depletion laser is centered at 810 nm and the FWHM is 4 nm, making the matching with the $^1D_2 \rightarrow ^3F_2$ transition accessible.

It should be noted that the $^1D_2 \rightarrow ^3F_2$ transition is very easy to be overlooked and was indeed often overlooked in the literature probably due to the following reasons: (a) A common misconception exists that the 3F_2 and 3F_3 states can be treated as a single state (usually denoted as $^3F_{2,3}$) while ignoring their difference because of the fact that the ions at the 3F_2 state quickly decays to the 3F_3 state nonradiatively; (b) The transition $^1D_2 \rightarrow ^3F_2$ spectrally overlap with other transitions, including $^3H_4 \rightarrow ^3H_6$ and $^1G_4 \rightarrow ^3H_5$, [51-53] making it barely distinguishable from others and thus causing negligence. As discussed in our response to Comment 1, the occurrence of the stimulated emission process $^1D_2 \rightarrow ^3F_2$ is substantially supported by the following results:

a. The 700 nm emission, originating from the 3F_3 state, to which the ions at the 3F_2 state quickly decay, was

obviously enhanced with the addition of the depletion laser at 810 nm, compared to single 975 nm excitation, in our UCNPs with both low and high Tm^{3+} concentrations.

b. The lifetime of the 455 nm emission was pronouncedly decreased with the addition of the depletion laser at 810 nm in all our UCNPs with different Tm^{3+} concentrations.

c. In our UCNPs with low (0.5%) Tm^{3+} concentrations, where the 475 nm emission (three-photon process) was enhanced (by ~240%) with the addition of the depletion laser at 810 nm, the 455 nm emission (five-photon process) was enhanced (by ~55%) much less significantly (Fig. R7(a)). In principle, in UCNPs higher-order multiphoton emission (higher-order power dependence) would be enhanced more significantly than lower-order multiphoton emission, if stimulated emission did not happen to the higher emitting state ($^1\text{D}_2$).

Last but not least, it should be noted that in STED system the depletion laser wavelength is generally intended to be located in the long-wavelength tail of the emission spectrum of the fluorophores (or to be deviated from the emission peak) in order to minimize the fluorescence background caused by the depletion laser itself and guarantee high depletion efficiency [8]. As shown in Fig. R10, the emission intensity ($I_{455\text{ nm}}$) under single 810-nm excitation is almost one order magnitude smaller than that under single 795-nm excitation. The selection of the depletion laser wavelength in our study is in line with this general requirement on the depletion laser.

Figure R10. Upconversion emission spectra of 10% Tm-UCNPs under excitation of 795-nm laser and 810-nm laser with the same power density, respectively.

Changes in the manuscript:

Sentences added (lines 129-130, page 7): “The matching of the 810 nm laser light with the transition of $^1\text{D}_2 \rightarrow ^3\text{F}_2$ is supported by many previous reports^{37, 39, 40, 41}.”

References added:

Zhang H, Jia T, Shang X, Zhang S, Sun Z, Qiu J. Mechanisms of the blue emission of NaYF₄:Tm³⁺ nanoparticles excited by an 800 nm continuous wave laser. *Phys. Chem. Chem. Phys.* **18**, 25905-25914 (2016).

Morrison C. A., Leavitt R. P. Chapter 46 Spectroscopic properties of triply ionized. *Handbook on the Physics & Chemistry of Rare Earths* **5**, 461-692 (1982).

Gruber J. B., Leavitt R. P., Morrison C. A. Absorption spectrum, energy levels, and crystal - field parameters of Tm³⁺:LaCl₃. *J. Chem. Phys.* **74**, 2705-2709 (1981).

Simpson D. A., *et al.* Visible and near infra-red up-conversion in Tm³⁺/Yb³⁺ co-doped silica fibers under 980 nm excitation. *Opt. Express* **16**, 13781-13799 (2008).

(3) See line 124, page 6, they attributed ¹D₂ transitions to six-photon upconversion process according to a literature. We checked the literature and found that 5-photon process was involved in Y₂O₃: 0.2 mol.% Tm³⁺ and 3 mol.% Yb³⁺ in the reference. That's totally different host and doping concentration cases. Actually, the ¹D₂ population in conventional NaYF₄ nanoparticles (18-20% Yb³⁺; <2 mol% Tm³⁺) always attribute to 4-photon process. To verify the 6 photon process, claimed by the authors, we suggest to measure the power-intensity curve to confirm the transition process.

Response: We apologize for a typo in the original manuscript that assigned the population of the ¹D₂ state to a six-photon process. Actually, we hold the view that the ¹D₂ state is populated through a five-photon process under single 975 nm excitation, which is supported by previous reports [54, 59-62]. In this process, one energy transfer step from Yb³⁺ ion to Tm³⁺ ion populates the ³H₅ level of Tm³⁺ from ³H₆. The ³H₅ decays rapidly to the ³F₄ level. A second energy transfer step raises the Tm³⁺ ion from ³F₄ to ³F₂ that quickly decays to ³H₄. Subsequently, a third transfer step raises the Tm³⁺ ion from ³H₄ to ¹G₄. Eventually, cross relaxation between adjacent Tm³⁺ ions, with two possible routes: ³H₄ + ¹G₄ → ³F₄ + ¹D₂ and ³H₄ + ¹G₄ → ¹D₂ + ³F₄, populate the ¹D₂ state, exhibiting a five-photon process. This five-photon upconversion pathway to populate the ¹D₂ state has been reported not only in Yb³⁺-Tm³⁺-codoped oxides (*e.g.* Y₂O₃) [63] but also in NaYF₄ hosts [54, 61, 62]. An alternative way to populate ¹D₂ is through a fourth energy transfer step from an excited Yb³⁺ ion to the Tm³⁺ ion at the ¹G₄ state, which is however usually less efficient due to the relative large energy mismatch ($\lambda(^1G_4 \rightarrow ^1D_2)$: around 1500 nm) [52]. The five-photon process for the population of the ¹D₂ state is supported by the measured power dependence of the 455 nm emission under single 975 nm excitation, as shown in Fig. R11.

Figure R11 Power dependence of the 455 nm emission under single 975 nm excitation

Change in the manuscript:

Sentence modified to (lines 126-128, page 6-7): “As illustrated in Fig. 2a, the 1D_2 emitting state, is populated by the 975-nm excitation beam through a five-photon upconversion process^{36,37,38},...”

References added:

Chen, G. Y., Somesfalean, G., Zhang, Z. G., Sun, Q. & Wang, F. P. Ultraviolet upconversion fluorescence in rare-earth-ion-doped Y_2O_3 induced by infrared diode laser excitation. *Opt. Lett.* **32**, 87-89 (2007).

Chen, X. & Song, Z. Study on six-photon and five-photon ultraviolet upconversion luminescence. *J. Opt. Soc. Am. B* **24**, 965-971 (2007).

Zhang, H. *et al.*, Mechanisms of the blue emission of $NaYF_4:Tm^{3+}$ nanoparticles excited by an 800 nm continuous wave laser. *Phys. Chem. Chem. Phys.* **18**, 25905-25914 (2016).

Wang, G. *et al.* Intense ultraviolet upconversion luminescence from hexagonal $NaYF_4:Yb^{3+}/Tm^{3+}$ microcrystals. *Opt. Express* **16**, 11907-11914 (2008).

Zhang, H., Li, Y., Lin, Y., Huang, Y. & Duan, X. Composition tuning the upconversion emission in $NaYF_4:Yb/Tm$ hexaplate nanocrystals. *Nanoscale* **3**, 963-966, (2011).

(4) To further exclude the possibility of stimulated depletion (emission), and the side-way emission depletion mechanism described by the authors, the lifetime measurement (Fig. S7) still shows the slow decay lifetime (>50 us) when the blue emission at 80% depletion power is applied. It is very obvious that there is no stimulated emission depletion at 455 nm, otherwise, the lifetime should be dramatically reduced to sub microsecond range.

Response: We thank the reviewer very much for his/her comments on the decay lifetime data. As noted by Reviewer #3, there was a bias issue in our previous time-resolved measurements. In our previous lifetime measurements, the laser beam at 975 nm was modulated by a mechanical chopper at a frequency of 1 kHz with a duty cycle of 50%, yielding a 500 μ s pulse duration and a 500 μ s off time. We have identified that the finite diameter of the laser beam mechanically chopped by the chopper led to a relatively slow instrument response function (IRF), causing a non-negligible bias on the lifetime measurement.

We have significantly optimized the lifetime experiments by further minimizing the beam diameter and utilizing a modulation frequency as high as possible to obtain a significantly faster IRF. Furthermore, we quantified the IRF of the system by measuring the fluorescence lifetime of rhodamine dye (with lifetime in order of ~ns), and then deconvolved the decay curves of upconversion luminescence with the obtained IRF to eliminate the influence of the system bias. We have updated the lifetime data with the system bias corrected throughout the manuscript and in the supplementary information, including Figure 2c and Figure S6, as shown in Fig. R12.

In our experiments, the lifetime decreased obviously, but did not dramatically reduced to sub microsecond range, which can be ascribed to the upconversion excitation approach, where a 975 nm laser was used to generate the 455 nm luminescence and was modulated to acquire the emission decay curves. Due to the stepwise pumping nature, the obtained decay curves usually are not merely determined by the lifetime of the emitting state of interest and often reflect the lifetimes of the intermediate states instead [70], which are significantly longer than that of the 1D_2 state and not affected by the depletion laser.

Change in the manuscript:

We have updated the lifetime measurement, data with the system bias corrected throughout the manuscript and in the supplementary information, including Figure 2c and Figure S6.

Sentences added (Lines 138-143, page 7): “The apparently less dramatic lifetime change in our study can be ascribed to the upconversion excitation approach, where a 975 nm laser was used to generate the 455 nm luminescence and was modulated to acquire the emission decay curves. Due to the stepwise pumping nature, the obtained decay curves are not solely determined by the lifetime of the emitting state and often reflect the transition properties of the intermediate states instead⁴².”

Figure R12 (Fig. S6 in the revised Supplementary Information) **The luminescence decay curves of the 455-nm emission of UCNP with different Tm^{3+} doping concentrations (0.5% Tm^{3+} , 2% Tm^{3+} , 5% Tm^{3+} , 7% Tm^{3+} , 15% Tm^{3+} , 20% Tm^{3+}) under 975-nm&810-nm simultaneous excitation.** The 975 nm laser beam was mechanically modulated using a chopper, while the 810 nm laser beam irradiated the sample continuously. The power density of the 810 nm depletion laser beam at the sample was adjusted to 0 MW/cm², 0.3 MW/cm², 1.48 MW/cm², 2.95 MW/cm², 5.9 MW/cm², 11.8 MW/cm², and 15.7 MW/cm² in sequence, and an emission decay curve was recorded at each power density. The lifetime of the 455nm luminescence decreased when increasing the power of the 810-nm depletion laser in all UCNP with different Tm^{3+} concentrations, indicating the occurrence of the stimulated emission process $^1D_2 \xrightarrow{810\text{ nm}} ^3F_2$.

(5) The super resolution imaging of nanoparticles on cells is not a convincing demonstration, as it does not show any specific imaging of sub-cellular structures. The typical demonstration, acceptable by super resolution community, should be to show the resolution improvement in imaging the cytoskeleton of a cell, made up of microtubules and actin filaments. The current demonstration of imaging the nanoparticles on the cell does not deliver additional value than the imaging of single nanoparticles.

Response: We thank the reviewer very much for the suggestion on the biological imaging experiments. As requested, we have improved the nanoparticle synthesis, surface modification, and antibody bioconjugation, and have achieved high quality immunolabelling of the cytoskeleton protein desmin using secondary antibody conjugated UCNPs (HeLa cancer cell; primary antibody Anti-desmin antibody (rabbit), secondary antibody: goat anti-rabbit IgG antibody), imaged using the upconversion nanoscopy technique developed in this study (Figure R13). The line profile analysis for a local area indicates that the lateral imaging resolution is increased to 82 nm, less than $\lambda/10$ ($\lambda_{ex}=975$ nm) (Figure R13). Cytoskeleton protein labeling with inorganic nanoparticles is generally challenging and had never been achieved using UCNPs previously. To the best of our knowledge, this is the first demonstration of immunolabelling of fine subcellular structures using UCNPs, as well as the first demonstration of super-resolution bio-imaging using UCNPs.

Change in the manuscript:

We have made a lot of effort in the super-resolution bio-imaging using UCNPs and finally achieved and updated the biological super-resolution imaging data in Figure 4 in the revised manuscript.

Figure R13 (Fig. 4 in the revised manuscript) **Immunofluorescence labeling of cellular cytoskeleton protein desmin with antibody conjugated UCNPs and super-resolution imaging.** (a) The multiphoton imaging under 975 nm excitation of some cytoskeleton structures and desmin proteins in HeLa cancer cells incubated with anti-desmin primary antibody and immunostained with UCNPs (~11.8 nm in diameter) bioconjugated with goat Anti-rabbit IgG secondary antibody. (b) The same region with (a) imaged in the super-resolution mode (975 nm excitation and the 810 nm STED laser beam), Scale bar: 2 μm . (c-n) Magnified areas selected from a, b (marked by white dotted squares) and line profile analyses; Images in c, f, i and l are taken from the white dotted squares in a; Images in d, g, j and m are taken from the white dotted squares in b; (e, h, k, n) Line profiles analyses of several areas indicated by arrow heads in c and d, f and g, i and j, and l and m, respectively.

Sentences added (lines 235-250, page 11): “Finally, we imaged the cytoskeleton of HeLa cancer cells with UCNP-mediated nanoscopy. High Tm^{3+} -doping UCNPs ($\text{NaYF}_4:18\% \text{Yb}^{3+}, 10\% \text{Tm}^{3+}$) with an average diameter of $11.8(\pm 2.2)$ nm were synthesized (Fig. S5). After surface modification with poly(acrylic acid) (PAA) (Fig. S19), these UCNPs were conjugated with the secondary antibody (goat anti-rabbit IgG antibody) (Fig. S20). Fixed HeLa cancer cells were incubated with anti-desmin monoclonal antibody (cytoskeleton marker)⁴⁶, and then stained with the secondary antibody IgG conjugated UCNPs. Such immunostained HeLa cells were imaged in the super-resolution mode under 975-nm and 810-nm co-irradiation, and with a reference under irradiation of just the 975-nm laser. As shown in Fig. 4, subcellular structures were clearly imaged, and a significant improvement in the spatial resolution was achieved with the assistance of the 810-nm depletion laser beam (Figs. 4a-4n). To evaluate the resolution of STED imaging, line profiles of several areas were extracted and fitted with Gaussian functions. The results reveal that the lateral imaging resolution has been improved to ~82 nm, less than $\lambda/10$ ($\lambda_{\text{ex}} = 975$ nm) (Fig. 4c-4e). To the best of our knowledge, this is the first demonstration of successful immunolabelling of fine subcellular structures as well as the first demonstration of super-resolution cell imaging using UCNPs.”

(6) Low frame rate (every 5 second) for single particle tracking experiment needs a bit more work. Additionally, how do we know it is the single nanoparticle instead of the particle cluster? As it is often the case that single monodispersed nanoparticles become aggregates in the biological physiological conditions due to their sticky surface. Also, according to the authors “there is no essential decrease in the intensity”, but the intensity of the particles was increased from 0.3 to 0.8, why?

Response: We thank the reviewer very much for the comments on the particle tracking experiment. We agree that it was probably particle cluster that was being imaged, as the nanoparticles were often endocytosed into cells and thus multiple particles resided in one vesicle [49]. We have reduced the claim from “single particle tracking” to “nanoparticle tracking”. In addition, in order to provide a presentation on our most important advances, the results of the nanoparticle tracking study were moved to the supplementary information. Despite

of this, we would like to emphasize that the significance of the particle tracking study was the demonstration of the capacity of UCNPs for long-term and continuous bioimaging with super-resolution, due to the excellent photostability of these nanoparticles. Long-term and continuous imaging is in need and has long been a challenge when using organic fluorescent probes.

The UCNPs have excellent photostability, as reported in other studies and confirmed by our measurements shown in Figure 3 in the revised manuscript. The variation of intensity in the particle tracking experiment was mainly due to the motion of nanoparticles induced defocusing in the imaging process.

Change in the manuscript: We have modified and moved the figure of nanoparticle tracking study to the revised supplementary information (Fig. S16) and reduced the claims accordingly.

(7) The long pixel dwell time (6-10 ms) will lead to relatively long image acquisition times compared to traditional STED (50 -200 μ s). So how to avoid the cell damage (photo toxicity) in this case? Additional data may be needed here to prove the cell viability after the long time tracking.

Response: We are sorry for not clarifying the dwell time in scanning imaging in the original manuscript. In our system, galvanometer mirrors were used and pixel dwell times of 100 μ s and 20 μ s were employed in the laser scanning super-resolution imaging and nanoparticle tracking experiments, respectively. The dwell times were comparable to that used in traditional STED imaging (50-200 μ s) in order to minimize the photo toxicity, and thus no problem in terms of the cell damage (photo toxicity) in this case.

Change in the manuscript:

Sentence added (lines 231-234, page 11): “The pixel dwell time (100 μ s) employed in the laser scanning super-resolution imaging is comparable to that used in traditional STED imaging (50-200 μ s), and the image acquisition speed can be increased with STED-parallelization strategies⁴⁵.”

(8) In the main text, the transitions of two emissions at 700 nm and 1064 nm should be indicated when the single beam 810 nm is applied.

Response: As suggested by the reviewer, the upconversion pathways for the 700 nm and 1064 nm emission under single 810 nm excitation have been indicated in the revised manuscript. A corresponding diagram of energy level (Fig. R14, or Fig. S7 in the revised supplementary information) and relevant references have been added [54, 71].

Change in the manuscript:

The following figure has been added in the revised supplementary information.

Figure R14 (Fig. S7 in the revised Supplementary Information) Upconversion pathways of the 700 nm and 1064 nm emissions under single 810 nm excitation

References added:

Zhang, H. *et al.* Mechanisms of the blue emission of NaYF₄:Tm³⁺ nanoparticles excited by an 800 nm continuous wave laser. *Phys. Chem. Chem. Phys.* **18**, 25905-25914 (2016).

Loiko, P. & Pollnau, M. Stochastic Model of Energy-Transfer Processes Among Rare-Earth Ions: Example of Al₂O₃:Tm³⁺. *J. Phys. Chem. C* **120**, 26480-26489 (2016).

(9) The emission peaks in Fig. 3 should be labelled accordingly with transition notes to facilitate readers. Error bars are needed in figure 2e and in figure 3c.

Response: We thank the reviewer very much for the valuable suggestions. More experiments have been performed and Fig. 3d, Fig. 2e and Fig. 3c (in the original manuscript) have been modified accordingly and renumbered as Figure 2d, Figure 1f and Figure 1g, respectively, in the revised manuscript.

(10) The samples have different particle sizes, e.g. 22 nm for NaYF₄:18%Yb³⁺, 20%Tm³⁺, 18 nm for NaYF₄:18%Yb³⁺, 10%Tm³⁺. How to exclude the size effect that affecting the depletion efficiency?

Response: We have synthesized two more UCNP samples with significant different average diameters (11.8 nm and 49.5 nm, as shown in Figures R1 and R15, respectively). We measured the dependence of the depletion efficiency on the power of the 810 nm beam of these samples and compared with that of the old one (~18 nm). The results reveal that there is no significant difference in the depletion efficiency of these different sized nanoparticles, as shown in Figure R15, indicating indistinctive size effect. This also agrees with the findings reported in Figure 5d in Ref. 10.

Change in the manuscript:

Sentence added (lines 116-120, page 6): “For different sized UCNPs with the same composition ($\text{NaYF}_4:18\% \text{Yb}^{3+}, 10\% \text{Tm}^{3+}$), all show efficient depletion of the 455 nm upconversion luminescence ($\lambda_{\text{exc}}=975 \text{ nm}$) by the 810 nm laser beam, with slightly different saturation excitation intensities (Fig. S5).”

Figure R15 (Fig. S5 in the revised Supplementary Information) **The effect of nanoparticle size on the depletion efficiency.** (a) The depletion efficiencies of three different sized UCNP samples at different powers of the 810 nm depletion laser. (b) TEM image of the high Tm^{3+} -doping ($\text{NaGdF}_4:18\% \text{Yb}^{3+}/10\% \text{Tm}^{3+}$) UCNPs with an average diameter of $11.8 \pm 2.2 \text{ nm}$, small nanoparticles employed in the immunolabelling experiments. (c) TEM image of the high Tm^{3+} -doping UCNPs ($\text{NaYF}_4:18\% \text{Yb}^{3+}/10\% \text{Tm}^{3+}$) with an average diameter of $49.5 \pm 1.6 \text{ nm}$. The 455 nm emission of all the samples can be efficiently depleted by the 810 nm laser with slight difference in the depletion efficiency, showing insignificant size effect. The TEM image of the measured $\text{NaYF}_4:18\% \text{Yb}^{3+}/10\% \text{Tm}^{3+}$ with an average diameter of $18.0 \pm 1.8 \text{ nm}$ was shown in Figure 1a of the main text.

(11) The whole spectra including the NIR part should be given in the supplementary Fig. S5.

Response: To present our work more logically, we have made adjustment to the content of the original manuscript. The content of Fig. S5 in the original manuscript has been moved to the main text as Figure 1e, and the corresponding NIR part for 0.5% Tm^{3+} UCNPs has been added to supplementary information as a supplementary figure (Fig. S9), as shown in Fig. R16.

Change in the manuscript:

Sentence modified (Lines 168-171, page 8): “Our experimental results have shown that with the increase of Tm^{3+} doping concentration (from 0.5% to 10%), the ratio I_{1470}/I_{1800} shows a very significant decrease (from about 5:1 to 1:32), indicating an enhancement factor of ~ 160 for I_{1800} with respect to I_{1470} (Fig. 2e, Fig. S9).”

Figure R16 (Fig. S9 in the revised Supplementary Information) NIR emission spectra of low Tm^{3+} -doping UCNPs ($\text{NaYF}_4:18\% \text{Yb}^{3+}/0.5\% \text{Tm}^{3+}$) under different excitation conditions. The power density is 700 kW/cm^2 for the 975-nm CW laser beam, and 17.7 MW/cm^2 for the 810-nm CW laser beam. Note that the intensities of 1400 nm and 1800 nm emissions were both increased with the addition of the 810 nm laser.

(12) Can the decay lifetimes of 1800-nm emission at different depletion efficiencies be measured? This will provide more powerful evidence than that of 455 nm.

Response: Limited by instrumentation, we cannot measure the decay lifetime of the 1800 nm emission band originating from the transition ${}^3\text{F}_4 \rightarrow {}^3\text{H}_6$. In addition, we have reservations on the significance of the lifetime data of 1800 nm emission due to the following considerations:

(a) Two critical processes that are responsible for the optical depletion of the 455 nm luminescence are a stimulated emission process (${}^1\text{D}_2 \xrightarrow{810 \text{ nm}} {}^3\text{F}_2$) and a cross relaxation between Tm^{3+} ions (${}^3\text{H}_4 + {}^3\text{H}_6 \rightarrow {}^3\text{F}_4 + {}^3\text{F}_4$), referring to our response to Comment 1.

(b) The lifetime data of 1800 nm emission cannot provide evidence for the stimulated emission process ${}^1\text{D}_2 \xrightarrow{810 \text{ nm}} {}^3\text{F}_2$, since the ${}^3\text{F}_4$ state is not involved in this process.

(c) The ${}^3\text{F}_4$ state is involved in the cross relaxation ${}^3\text{H}_4 + {}^3\text{H}_6 \rightarrow {}^3\text{F}_4 + {}^3\text{F}_4$. The spectroscopic data presented in Figure 2c and Figure 2d have adequately supported the importance of this cross relaxation process.

(d) Finally, we speculate that the decay lifetime of the 1800 nm emission would not change with addition of 810 nm laser light, since the ${}^3\text{F}_4$ state is not involved in any stimulated emission processes according to not only our theoretical analysis but also that in Ref. 10.

(13) Supplementary figure S11: “with our experimental data (Fig. 2(b))” should be “with our experimental data (Fig. 2(e))”?

Response: We thank the reviewer for pointing out this typo in the original manuscript. This typo has been corrected in the revised manuscript.

Reviewer 3:

General comments:

This paper entitled “Achieving high-efficiency emission depletion nanoscopy by inhibiting re-excitation to the emitting-state of upconversion nanoparticles” by Zhan et al. describes a very exciting approach to achieve super-resolution microscopy. Through the cross-relaxation in the upconversion nanoparticles, the saturation intensity can be decreased by 2-3 orders of magnitude, leading to super-resolution with low power. This interesting finding has led to the demonstration of 66-nm super-resolution. The authors have further demonstrated that the UCNPs can be used to stain the subcellular organelle, for super-resolution and single particle tracking. Overall, I think the novelty and breath of the manuscript meets well the publication requirements. However, the depth of this work should be further improved, with substantial further experiments, to justify its publication in Nature Communications.

Below are my comments/suggestions:

Response: We thank the reviewer very much for his/her positive comments and suggestions.

Specific comments:

(1) According to the depletion equation, $\eta = \frac{1}{1+I_{STED}/I_s}$. The $I_s=1.78 \text{ MW cm}^{-2}$, and the STED intensity $=17.7 \text{ MW cm}^{-2}$, which is 10x the I_s . Based on this, it is unlikely that the intensity can be depleted to 2%, and similarly, the resolution of 66 nm cannot be obtained, with such I_s , as the resolution of STED obeys a square root law: $d_{STED} = d_c \frac{1}{\sqrt{1+\alpha I_{STED}/I_s}}$. And more importantly, here the $\alpha < 1$, and I_{STED} is only one quarter that of Gaussian focal spot, as it is modulated to a donut. Therefore, I encourage the authors to double check the measurement of the saturation intensity.

Response: We thank the reviewer very much for reminding us the discrepancy between our experimental data and the theoretical equation for the emission depletion reported previously. We would like to point out that the depletion equation (i.e., $\eta = \frac{1}{1+I_{STED}/I_s}$) and the resolution equation for STED imaging ($d_{STED} = d_c \frac{1}{\sqrt{1+\alpha I_{STED}/I_s}}$), derived previously by Hell et al. [39, 72] are probably not applicable to the lanthanide upconversion luminescence in our study. These equations were obtained with assumptions based on a simplified Jablonski diagram of typical downconversion fluorescence, involving photophysical processes of light absorption, relaxation, spontaneous emission and stimulated emission, and are really dependent on the specific luminescence mechanism. The explicit form of the depletion equation is even dependent on the excitation/depletion approach, yielding different equations for pulse- and CW-STED [72]. These equations lose their validity in the present case of optical depletion of lanthanide upconversion phosphorescence,

involving complicated energy transfer pathways between lanthanide ions besides photophysical processes, which are not considered in the derivation of these equations. The inapplicability of these equations to optical depletion of lanthanide upconversion emission is also confirmed by our previous work [73] and a very recent work [10], reporting that optical depletion efficiency data of upconversion luminescence could not be well fitted with the depletion equation $\eta = \frac{1}{1+I_{STED}/I_s}$ and the resolution could not be well fitted with the equation $d_{STED} = d_c \frac{1}{\sqrt{1+\alpha I_{STED}/I_s}}$. However, we still adopt the parameter -- the saturation intensity, at which 50% emission off is achieved, to describe the emission depletion.

Regarding the statement of “2% residual luminescence” in the original manuscript, we will clarify in our response to Comment 2 below.

Following the reviewer’s suggestion, we have repeated the measurement of the saturation intensity and made necessary corrections. The saturation intensity (I_s) is 849 kW/cm², and when the depletion efficiency reaches 96%, the power intensity is 17.7 MW/cm².

Changes made in the manuscript:

Sentence modified (lines 116-123, page 6.): “The degree of luminescence inhibition was measured for various values of I_{810} (Fig. 1g). About 4% residual luminescence can be detected for $I_{810} = 17.7 \text{ MW/cm}^2$, confirming the effectiveness of the luminescence inhibition process. The saturation excitation intensity I_{sat} , inducing 50% emission off, is estimated to be 849 kW/cm² (power ~2.4 mW). The 810 nm laser itself can only generate very weak upconversion emission (Fig. 1b). The background luminescence intensity generated by single 810 nm excitation (17.7 MW/cm²) is around 2% of that generated by single 975 nm excitation (700 kW/cm²) (Fig. 1b).”

(2) In page 7, the two statements “Less than 2% residual luminescence can be achieved for $I_{810} = 17.7 \text{ MW/cm}^2$ ” and “when co-irradiated by an 810 nm CW laser beam (17.7 MW/cm²), the most intense 455 nm blue emission was greatly quenched, with a depletion efficiency approaching 96%” is inconsistent.

Response: We are sorry for our original expression that may cause misunderstanding. By “Less than 2% residual luminescence can be achieved for $I_{810} = 17.7 \text{ MW/cm}^2$ ”, we meant that a weak background luminescence could be detected under single 810-nm excitation (17.7 MW/cm²) and the relative intensity is 2 % compared to that under single 975-nm excitation (700 kW/cm²). We have modified this statement in the revised manuscript.

Changes in the manuscript:

Sentence modified (lines 116-123, page 6.): “The degree of luminescence inhibition was measured for various values of I_{810} (Fig. 1g). About 4% residual luminescence can be detected for $I_{810} = 17.7 \text{ MW/cm}^2$, confirming the effectiveness of the luminescence inhibition process. The saturation excitation intensity I_{sat} , inducing 50% emission off, is estimated to be 849 kW/cm^2 (power $\sim 2.4 \text{ mW}$). The 810 nm laser itself can only generate very weak upconversion emission (Fig. 1b). The background luminescence intensity generated by single 810 nm excitation (17.7 MW/cm^2) is around 2% of that generated by single 975 nm excitation (700 kW/cm^2) (Fig. 1b).”

(3) It is very interesting to see that when the Tm^{3+} concentration is greater than 10% in the 18% Yb doping nanoparticle, the depletion efficiency decreases with the doping concentration.

a) Any explanation to this?

b) What is the relationship between the doping concentration and the depletion efficiency? Is it related with the cross-relaxation?

Response: This phenomenon is correlated with our mechanism on the optical depletion. All our spectroscopic data strongly support a formulation of the mechanism for the observed optical depletion of 455 nm upconversion luminescence caused by the 810 nm laser beam (Figure R17):

- The addition of the depletion laser beam at 810 nm has dual action on the 455 nm upconversion luminescence generated by the 975 nm excitation laser: **a depletion effect** by de-exciting the $^1\text{D}_2$ state via a stimulated emission process, and **an enhancement effect (or synergistic excitation)** by populating the intermediate $^3\text{H}_4$ state followed by re-populating the $^1\text{D}_2$ state with the assistance of the 975 nm beam.
- The net change of the 455 nm upconversion luminescence is determined by the **competition** between the depletion effect and the enhancement effect.
- With high Tm^{3+} concentrations (high CR rate for $^3\text{H}_4 + ^3\text{H}_6 \rightarrow ^3\text{F}_4 + ^3\text{F}_4$, transferring the electrons at $^3\text{H}_4$ state to the $^3\text{F}_4$ state, the CR1 in Figure R17), the re-population rate of the $^1\text{D}_2$ state by both lasers is much weaker than the de-excitation rate by only the depletion laser (810 nm), leading to inhibition of the 455 nm luminescence.
- With low Tm^{3+} concentrations (low CR rate for $^3\text{H}_4 + ^3\text{H}_6 \rightarrow ^3\text{F}_4 + ^3\text{F}_4$), the re-population rate of the $^1\text{D}_2$ state by both lasers is stronger than the de-excitation rate by only the depletion laser (810 nm), leading to enhancement of the 455 nm luminescence.

This mechanism is well supported by all spectroscopic data that we have obtained, including the emission enhancement/inhibition for different emission bands, concentration dependence, and lifetime data.

In such a framework, any process that potentially strengthen the enhancement effect (or synergistic excitation) of the 810 nm beam will compromise the depletion efficiency. Regarding the decrease of the depletion efficiency induced by the 810 nm beam with the Tm^{3+} doping concentration above 10%, a possible reason could be that the otherwise inefficient cross relaxations between Tm^{3+} ions become prominent at adequately high Tm^{3+} concentration, *e.g.*, $^1\text{G}_4 + ^3\text{F}_4 \rightarrow ^1\text{D}_2 + ^3\text{H}_6$ and $^3\text{F}_{2,3} + ^3\text{H}_4 \rightarrow ^1\text{D}_2 + ^3\text{H}_6$, as discussed in detail in a previous report [54], which facilitate the synergistic excitation effect of the 810 nm laser beam.

Changes in the manuscript:

Sentence added (Lines 179-184, page 9): “The decrease of the depletion efficiency induced by the 810 nm beam with the Tm^{3+} doping concentration above 10% (Fig. 1f) could be due to the activation of the otherwise inefficient cross relaxations between Tm^{3+} ions at adequately high Tm^{3+} concentration, *e.g.*, $^1\text{G}_4 + ^3\text{F}_4 \rightarrow ^1\text{D}_2 + ^3\text{H}_6$ and $^3\text{F}_{2,3} + ^3\text{H}_4 \rightarrow ^1\text{D}_2 + ^3\text{H}_6$, as reported in a previous report³⁷, which enhance the synergistic excitation effect of the 810 nm laser beam.”

Figure R17 (Figs. 2a-2b in the revised manuscript) Proposed optical emission depletion mechanism for the 455 nm upconversion band of Tm^{3+} of $\text{NaYF}_4:18\% \text{Yb}^{3+}/10\% \text{Tm}^{3+}$ nanoparticles.

(4) The mechanism of stimulated depletion on $^1\text{D}_2$ level is not well supported. The stimulated emission is a very fast process which has lifetime of $\sim\text{ps}$ level. If the stimulated emission is from the highest energy level, one should see very dramatic lifetime change. While in Fig. 2e, the lifetime is from 45 μs to 32 μs .

- a) In the methods section, the authors only stated that, the lifetime is measured with a 1 kHz chopper, with window of 1 ms. How long is the duration of the pulsed excitation pulse? The effect of the excitation pulse should be carefully measured to avoid any bias to the lifetime measurement.

Response: We thank the reviewer very much for his/her valuable comments, reminding us the bias issue in the lifetime measurement. In our previous lifetime measurements, the laser beam at 975 nm was modulated by a mechanical chopper at a frequency of 1 kHz with a duty cycle of 50%, yielding a 500 μ s pulse duration and a 500 μ s off time. We realized the finite diameter of the laser beam mechanically chopped by the chopper led to a relatively slow instrument response function (IRF), causing a non-negligible bias on the lifetime measurement. As requested, we have optimized the lifetime measurement by further minimizing the beam diameter and utilizing a modulation frequency as high as possible to obtain a significantly faster IRF. Furthermore, we quantified the IRF of the system by measuring the lifetime of rhodamine fluorescence (with lifetime in order of ~ns), and then deconvolved the decay curves of upconversion luminescence with this IRF to eliminate the influence of the system bias. We have updated the lifetime data with the system bias corrected throughout the manuscript and in the supplementary information, including Figure 2c and Figure S6.

The apparently less dramatic lifetime change in our study can be ascribed to the upconversion excitation approach, where a 975 nm laser was used to generate the 455 nm luminescence and was modulated to acquire the emission decay curves. Due to the stepwise pumping nature, the obtained decay curves are not solely determined by the lifetime of the emitting state and often reflect the transition properties of the intermediate states instead [70]. For instance, Gamelin *et al.* investigated the decay behavior of a simplified three-level ensemble upconversion system composed of identical ions through numerical simulations based on a rate equation model, and found that the decay of the transient population of the upper state in energy transfer upconversion process lasted substantially longer than the natural decay lifetime of the upper state and had a rate constant twice that of the intermediate under low-power excitation conditions [70].

Change in the manuscript:

We have updated the lifetime data with the system bias corrected throughout the manuscript and in the supplementary information, including Figure 2c and Figure S6.

Sentences added (Lines 138-143, page 7): “The apparently less dramatic lifetime change in our study can be ascribed to the upconversion excitation approach, where a 975 nm laser was used to generate the 455 nm luminescence and was modulated to acquire the emission decay curves. Due to the stepwise pumping nature, the obtained decay curves are not solely determined by the lifetime of the emitting state and often reflect the transition properties of the intermediate states instead⁴².”

(5) The Fig. 2e should be more carefully studied. Currently, it only tells us that when increasing the 808 nm intensity, for one specific doping concentration, the lifetime decreases. I suggest the authors to give more thorough study on this, by using UCNPs with different doping concentration, to study the effect of stimulated

emission to the lifetime. This can also help to understanding the relationship between the doping concentration and the spontaneous emission/stimulated emission.

Response: We thank the reviewer very much for his/her comments and suggestions. As requested, we have thoroughly investigated the effect of stimulated emission to the apparent decay lifetime of 455 nm emission for UCNPs with different Tm^{3+} doping concentrations. Our results show that the apparent lifetime is decreased in all samples, but to different extent with different Tm^{3+} doping concentrations, as shown in Figure R18. The lifetime decrease in samples with low Tm^{3+} concentrations also indicates that in our study the stimulated emission happened to the $^1\text{D}_2$ state rather than the $^3\text{H}_4$ state. The dependence of the relative change of the measured lifetime on the Tm^{3+} concentrations supports our speculation in our responses to previous Comments 3 and 4, disclosing that the lifetime obtained with an upconversion excitation approach relies on the concrete energy transfer pathways involving the intermediate energy states, which is highly doping concentration dependent.

Change in the manuscript:

We have updated the lifetime data with the system bias corrected throughout the manuscript and in the supplementary information, including Figure 2c and Figure S6.

Sentences added (Lines 138-143, page 7): “The apparently less dramatic lifetime change in our study can be ascribed to the upconversion excitation approach, where a 975 nm laser was used to generate the 455 nm luminescence and was modulated to acquire the emission decay curves. Due to the stepwise pumping nature, the obtained decay curves are not solely determined by the lifetime of the emitting state and often reflect the transition properties of the intermediate states instead⁴².”

Figure R18 (Fig. S6 in the revised Supplementary Information) The luminescence decay curves of the 455-nm emission of UCNP with different Tm^{3+} doping concentrations (0.5% Tm^{3+} , 2% Tm^{3+} , 5% Tm^{3+} , 7% Tm^{3+} , 15% Tm^{3+} , 20% Tm^{3+}) under 975-nm&810-nm simultaneous excitation. The 975 nm laser beam was mechanically modulated using a chopper, while the 810 nm laser beam irradiated the sample continuously. The power density of the 810 nm depletion laser beam at the sample was adjusted to 0 MW/cm², 0.3 MW/cm², 1.48 MW/cm², 2.95 MW/cm², 5.9 MW/cm², 11.8 MW/cm², and 15.7 MW/cm² in sequence, and an emission decay curve was recorded at each power density. The lifetime of the 455nm luminescence decreased when increasing the power of the 810-nm depletion laser in all UCNP with different Tm^{3+} concentrations, indicating the occurrence of the stimulated emission process $^1D_2 \xrightarrow{810\ nm} ^3F_2$.

(6) In Fig. S11 a simulated dependency of the fluorescence intensity, and 810 nm laser power has been given. Please correlate this figure with Fig. 2e to show how well the theory matches with the experiment.

Response: We thank the reviewer for his/her comment and advice. Our model and numerical simulation are intended to qualitatively evaluate the optical depletion effect induced by the 810 nm laser, using the parameter values from the literatures. We think the concentration dependence of the emission depletion is the major thing that we need to interpret by turning to simulations, which can help qualitatively identify the key mechanisms that account for the emission depletion. As requested, we have also implemented simulation for low Tm^{3+} -doping samples and have included the simulated results in Figure S12 in the supplementary information, as shown in Figure R19. The contrast between the low and high Tm^{3+} -doping samples confirms the importance of the CR1 process (${}^3\text{H}_4 + {}^3\text{H}_6 \rightarrow {}^3\text{F}_4 + {}^3\text{F}_4$) in the optical depletion of 455 nm upconversion luminescence, in accordance with experimental results.

Change in the manuscript:

We have included the simulated results of both low and high doping concentrations in Figure S12 in the revised supplementary information.

Sentence modified (Lines 184-187, page 9): “The critical role of the Tm^{3+} concentration dependent CR1 process is also supported by the results of our numerical simulations (Figs. S11 and S12).”

Figure R19 (Fig. S12 in the revised Supplementary Information) Simulated dependence of the intensity change of the 455-nm luminescence on the power of the 810 nm depletion beam in (a) low and (b) high Tm^{3+} -doping UCNPs.

(7) This work should demonstrate a real biological super-resolution imaging, by specifically labeling the UCNPs on one of the subcellular organelle. The current Fig. 5a and 5b are just showing that the UCNPs can enter the cell non-specifically, and super-resolution imaging can see a smaller UCNP spots, but no biological information can be revealed.

Response: We thank the reviewer very much for the valuable comment and suggestion. As requested, we have made a lot of efforts and significant progress in subcellular labeling and imaging experiments. Cytoskeleton protein labeling with inorganic nanoparticles is generally challenging and has never been achieved using UCNPs. We improved the nanoparticle synthesis, surface modification, and antibody bioconjugation, and have successfully achieved high quality immunolabelling of the cytoskeleton protein desmin using secondary antibody conjugated UCNPs (HeLa cancer cell; primary antibody Anti-desmin antibody (rabbit), secondary antibody: goat anti-rabbit IgG antibody), as shown in Figure R20. To the best of our knowledge, this is the first demonstration of immunolabelling fine subcellular structure using UCNPs.

Change in the manuscript:

We have made a lot of effort in the super-resolution bio-imaging using UCNPs and finally achieved and updated the biological super-resolution imaging data in Figure 4 in the revised experiment.

Figure R20 (Fig. 4 in the revised manuscript) Immunofluorescence labeling of cellular cytoskeleton protein desmin with UCNPs and super-resolution. (a) The multiphoton imaging under 975 nm excitation of some cytoskeleton structures and desmin proteins in HeLa cancer cells incubated with anti-desmin primary antibody and immunostained with UCNPs (~11.8 nm in diameter) bioconjugated with goat Anti-rabbit IgG secondary antibody. (b) The same region with (a) imaged in the super-resolution mode (975 nm excitation and the 810 nm STED laser beam), Scale bar: 2 μm. (c-n) Magnified areas selected from a, b (marked by white dotted squares) and line profile analyses; Images in c, f, i and l are taken from the white dotted squares in a; Images in d, g, j and m are taken from the white dotted squares in b; (e, h, k, n) Line profiles analyses of several areas indicated by arrow heads in c and d, f and g, i and j, and l and m, respectively.

Sentences added (lines 235-250, page 11): “Finally, we imaged the cytoskeleton of HeLa cancer cells with UCNP-mediated nanoscopy. High Tm³⁺-doping UCNPs (NaYF₄:18%Yb³⁺,10%Tm³⁺) with an average diameter of 11.8(±2.2) nm were synthesized (Fig. S5). After surface modification with poly(acrylic acid) (PAA) (Fig. S19), these UCNPs were conjugated with the secondary antibody (goat anti-rabbit IgG antibody) (Fig. S20). Fixed HeLa cancer cells were incubated with anti-desmin monoclonal antibody (cytoskeleton marker)⁴⁶, and then stained with the secondary antibody IgG conjugated UCNPs. Such immunostained HeLa cells were imaged in the super-resolution mode under 975-nm and 810-nm co-irradiation, and with a reference under irradiation of just the 975-nm laser. As shown in Fig. 4, subcellular structures were clearly imaged, and a significant improvement in the spatial resolution was achieved with the assistance of the 810-nm depletion laser beam (Figs. 4a-4n). To evaluate the resolution of STED imaging, line profiles of several areas were extracted and fitted with Gaussian functions. The results reveal that the lateral imaging resolution has been improved to ~82 nm, less than $\lambda/10$ ($\lambda_{ex} = 975$ nm) (Fig. 4c-4e). To the best of our knowledge, this is the first demonstration of successful immunolabelling of fine subcellular structures as well as the first demonstration of super-resolution cell imaging using UCNPs.”

(8) Likewise, the current Fig. 5i is only traces of the UCNPs, not necessarily any of the cellular molecules. It is only when the UCNP label is proved with other single molecule techniques, that one can claim “single particle tracking for cancer cell”. Specificity in the UCNP labeling should be demonstrated with *e.g.* correlative study of labeling the organelle with organic dye. Ideally, if this experiment can be performed in super-resolution, it will broaden the application of super-resolution in cellular study significantly.

Response: We thank the reviewer very much for the comments on the particle tracking experiment. We realize that it was probably nanoparticle cluster that was being imaged, as the nanoparticles were often endocytosed into cells and thus multiple particles resided in one vesicle [49]. We have reduced the claim from “single particle tracking” to “nanoparticle tracking”. In order to provide a presentation on our most important advances, the results of the nanoparticle tracking study were moved to the supplementary information. Despite of this, we would like to emphasize that the significance of the particle tracking study was the demonstration of the capacity of UCNPs for long-term and continuous bioimaging with super-resolution, due to the excellent photostability of these nanoparticles. Long-term and continuous imaging has been a challenge when using organic fluorescent probes.

Change in the manuscript:

We have moved the results of nanoparticle tracking study to the supplementary information as Fig. S16 in the revised manuscript and reduced the claims in the revised supplementary information.

(9) It is unclear to me of the yellow line in Fig. 2d. Why is it curved, like a manually drawing?

Response: We are sorry for this confusing content in the original manuscript. Our image analysis toolbox provides a function for intensity profile analysis for arbitrarily drawn lines. The yellow line in Fig. 2d in the original manuscript was indeed a manual drawing, crossing the centers of a few bright spots in the image, and the pixel intensity along this line was extracted and presented. To avoid any confusion, we have removed this manual drawing and the corresponding line profile analysis in the revised manuscript.

Change in the manuscript:

We have removed the manual drawing and corresponding line profile analysis, as shown in Figure 1d in the revised manuscript, in order to avoid any confusion.

(10) In my opinion Fig. 1 should be moved to SI, since the mechanism in this work doesn't belong to any of the mechanisms illustrated. I encourage the authors to improve the diagram for better understanding.

Response: We thank the reviewer for the comment and suggestions. We have moved Figure 1 in the original manuscript to the supplementary information as Fig. S1, after necessary modification to make it more relevant, precise and understandable, as shown in Figure R21.

Change in the manuscript:

The following schematic diagram (Figure 1 in the original manuscript) has been moved to the revised supplementary information as Fig. S1 after modification.

Figure R21 (Fig. S1 in the revised Supplementary Information) **Schematics of different optical depletion mechanisms.** (a) Traditional stimulated emission depletion (STED) mechanism; (b) Excited state absorption (ESA) induced emission depletion; (c) Ground state depletion (GSD) mechanism; (d) Matter-matter interaction assisted STED mechanism in Yb/Tm-codoped UCNP system. To date, several optical switching mechanisms of luminescence have been established and exploited in optical super-resolution imaging. With intense light-matter interaction caused by the additional depletion laser, the population of the emitting state was depleted to other dissipating states, through stimulated emission depletion (a), excited state absorption (b), or ground state depletion

(c). In these mechanisms, relatively high intensity of the depletion laser is required to guarantee considerable emission depletion efficiency, which inevitably increases the risk of light toxicity to the biological tissue under investigation or causes severe photobleaching to the luminescence probes in use. The proposed matter-matter interaction assisted STED mechanism in Yb/Tm-codoped UCNPs system employs interionic cross relaxation to suppress the synergistic excitation effect and amplify the stimulated emission depletion effect of the depletion laser, reducing the requirement on depletion laser intensity (d).

(11) The authors have claimed that the 1800 nm is significantly enhanced with 975&810 nm simultaneous illumination. However, in Fig. 3 the 700 nm, 1064 nm, and 1800 nm (black line) are shown just like the sum of the contribution of separate 975 nm (red) and 810 nm (green). Please explain it.

Response: The enhancement of the 700/1064/1800 nm emission with 975&810 nm simultaneous irradiation is relative to the case of single 975 nm excitation. The apparently less significant enhancement of the 700/1064/1800 nm band than expected can be explained by the excitation saturation effect of upconversion luminescence [74, 75]:

a. Under single 975 nm (denoted by case A) or 810 nm excitation (denoted by case B), an equilibrium between the population densities of different energy states was eventually achieved, starting from the original equilibrium with no laser excitation, where most ions stay at the ground state. Because of the adequately high excitation intensities used in our study, the excitation condition cannot be regarded as weak excitation and the ground state has been heavily evacuated, which indicates the occurrence of saturation of upconversion luminescence, predicting a smaller slope efficiency factor than n for an n -photon upconversion luminescence [74, 75]. Under such conditions, further increases in the excitation intensity will generate a far less significant increase of the emission intensity.

b. In the case of 975&810 nm co-irradiation (first the 975 nm irradiation followed by the addition of the 810 nm beam, denoted by case C), if splitting the contributions of 975 nm and 810 nm beams in the generation of the 700/1064/1800 nm emission, the additional intensity of the 700/1064/1800 nm band caused by the 810 nm beam will be smaller than that generated by the single 810 nm excitation in Case B, since the ground state has been efficiently evacuated by the 975 nm beam and saturation phenomenon for the 700/1064/1800 nm band has happened. Thus, if only considering the direction excitation effect of the 810 nm beam and ignoring other effects (*e.g.*, the stimulated emission mechanism discovered in this study), the intensity of the 700/1064/1800 nm emission in case C would be weaker than the sum of the intensities in case A and case B, which would be in conflict with our experimental results.

Detailed analysis reveals that the intensity of the 700/1064/1800 nm emission band under co-irradiation is slightly larger than the sum of the contributions of separate 975 nm (case A) and 810 nm excitation (case B),

as shown in Figure R22, which indicates the presence of other pathways populating the 3F_4 state, in line with our proposed mechanism, which is a stimulated emission process ($^1D_2 \xrightarrow{810\text{ nm}} ^3F_2$) followed by a cross-relaxation process ($^3H_4 + ^3H_6 \rightarrow ^3F_4 + ^3F_4$).

In addition, regarding the intensity change of the 1800 nm band, since the population density at the 1D_2 state (the 7th excited state, excited through a five-photon process) is generally much smaller than that at the much lower-lying 3F_4 state (the 1st excited state), the intensity increase caused by the electron transfer from the 1D_2 state indirectly to 3F_4 state is expected to be insignificant.

Figure R22 The intensity changes of 700 nm, 1064 nm, 1470 nm and 1800 nm emission bands of NaYF₄:18%Yb³⁺/10%Tm³⁺ UCNPs under single 975 nm, single 810 nm and both laser excitations (975 nm & 810 nm). The sums of single excitation contribution (975 nm + 810 nm) were plotted.

(12) From Fig. 5, it is interesting to see that often when the intensity drops, the velocity increases. Further statistical analysis should be performed on this.

Response: As the emission from the UCNPs is extremely photostable, confirmed by the photostability study in our work (Fig. 3c in the revised manuscript), the intensity variation of UCNPs in the nanoparticle tracking study was mainly caused by the position change of the nanoparticles relative to the focus of the excitation beam. Intensity drops indicate that the particles are moving away from the beam focus, correlated with the velocity increase of the particles. We have moved the results of nanoparticle tracking study to the supplementary information, referring to our response Comment 8.

Change in the manuscript:

We have moved the results of nanoparticle tracking study to the supplementary information as Fig. S16 in the revised manuscript and reduced the claims in the revised supplementary information.

Reviewer 4:

General comments:

I read this manuscript with considerable interest, given the implied strong claim on new mechanism for the depletion in STED-like microscopy in the abstract. The idea is to lock electrons at lower energy states (or to let the ion switch between these states, not allowing switching to 455 nm emitting state) to prohibit re-population of the emitting state. To my knowledge, this idea with the use of lanthanide-based UCNPs in STED-like microscopy is new. The idea can indeed potentially impact STED-microscopy by reducing requirement on STED-laser power, in additions advantaging in no photobleaching, and dual- and potentially multi-color STED imaging using single excitation and STED beams. The paper might be of interest not only for super-resolution microscopy field, but also for lanthanide/up-conversion materials.

However, I also noticed several essential drawbacks which preclude the publication unless eliminated.

Response: We thank the reviewer very much for his/her positive comments and suggestions.

1. Main claim was not properly supported by experimental data and analysis.
2. The claims are not appropriately discussed in the context of previous literature.

I will discuss them in detail.

Specific comments:

(1) In the simplest two-level scheme, the depletion efficiency depends on the spontaneous and the stimulated emission (SE) lifetimes of the emitting state. The latter is determined by the SE cross-section and the STED-laser power/focal area. The authors continuously claim that they locked the Tm^{3+} in between lower energy states, thus not allowing Tm^{3+} electrons to return back to the emitting state, basically making the competition between spontaneous and stimulated emission not/less important. This, however, was not properly supported neither by experiment nor by analysis.

Response: We thank the reviewer very much for his/her valuable comments which have led us to a deeper understanding on the upconversion optical depletion mechanism. All our spectroscopic data strongly support a formulation of the mechanism for the observed optical depletion of 455 nm upconversion luminescence caused by the 810 nm laser beam (Figure R23):

- The addition of the depletion laser beam at 810 nm has dual action on the 455 nm upconversion luminescence generated by the 975 nm excitation laser: **a depletion effect** by de-exciting the 1D_2 state via a stimulated emission process ($^1D_2 \rightarrow ^3F_2$), and **an enhancement effect (or synergistic excitation)** by populating the intermediate 3H_4 state followed by re-populating the 1D_2 state with the assistance of the 975 nm beam.

- The net change of the 455 nm upconversion luminescence is determined by the **competition between the depletion effect and the enhancement effect**.
- With high Tm^{3+} concentrations, the highly efficient CR1 process (${}^3\text{H}_4 + {}^3\text{H}_6 \rightarrow {}^3\text{F}_4 + {}^3\text{F}_4$) [76, 77] depicted in Figure R23 suppresses the synergistic excitation effect and simultaneously amplifies the depletion effect of the 810 nm beam by transferring the electrons at ${}^3\text{H}_4$ state to the ${}^3\text{F}_4$ state where an IR energy dissipating channel is introduced by emitting ~ 1800 nm radiation. This two-fold effect of the CR1 process make the re-population rate of the ${}^1\text{D}_2$ state by both lasers to be much weaker than the de-excitation rate by only the depletion laser, leading to inhibition of the 455 nm luminescence.
- With low Tm^{3+} concentrations, the absence or inefficiency of the CR1 process makes the re-population rate of the ${}^1\text{D}_2$ state by both lasers to be dominant over the de-excitation rate of the depletion laser, leading to enhancement of the 455 nm luminescence.

This mechanism is well supported by all spectroscopic data that we have obtained, including the emission enhancement/inhibition for different emission bands, concentration dependence, and lifetime data, presented in Figures 1 and 2 in the revised manuscript.

It should be noted that our claim in the original manuscript, that locking Tm^{3+} ions in between lower energy states, despite being not precise (and now has been revised to be more precise), is not in conflict with this revised formulation of the mechanism. Considering the critical role of the CR1 process in suppressing the synergistic excitation effect and simultaneously amplifying the depletion effect of the 810 nm beam by transferring the electrons at ${}^3\text{H}_4$ state to the ${}^3\text{F}_4$ state, the ${}^3\text{F}_4$ state indeed acts a sink or trap which dissipates the excitation energy added by the 810 nm beam. In order to describe the mechanism more precisely and in more details, we have modified our claims and made significant revision to the manuscript in the revised version. We have also modified the title of our manuscript accordingly.

Change in the manuscript:

We have modified the title of our manuscript and our claims accordingly in the revised version.

New title: Achieving high-efficiency emission depletion nanoscopy by employing interionic interaction in upconversion nanoparticles.

Sentences modified (lines 150-164, Page 7): “Along with the optical depletion through stimulated emission effect of the 810 nm laser beam, a synergistic excitation enhancement effect by the same laser beam is also active due to the presence of a ground state absorption process matching 810 nm, i.e., ${}^3\text{H}_6 \xrightarrow{810 \text{ nm}} {}^3\text{H}_4$, which efficiently populates a necessary intermediate state (i.e., ${}^3\text{H}_4$) of the upconversion process (Fig. S7). The increase of the population density at the ${}^3\text{H}_4$ state, well supported by the observed enhanced emission at around 1470 nm originating from the transition ${}^3\text{H}_4 \rightarrow {}^3\text{F}_4$ (Fig. 2d, Fig. S8), is in favor of upconversion

emissions originating from higher states above the 3H_4 state (including 1G_4 and 1D_2 states) under the co-irradiation of the 975 nm excitation beam, although single 810 nm excitation can only generate very weak upconversion luminescence (Fig. 1b, Fig. S7). This emission enhancement effect competes with the depletion effect, compromising the optical depletion efficiency induced by the 810 nm beam (Fig. 2b). A key cross relaxation (CR1) process between Tm^{3+} ions could suppress the aforementioned emission enhancement effect of 810 nm beam and facilitate the optical depletion, i.e., $^3H_4 + ^3H_6 \rightarrow ^3F_4 + ^3F_4$, through transferring electrons at the 3H_4 state to the 3F_4 state (Figs. 2a and 2b).”

Figure R23 (Figs. 2a-2b in the revised manuscript) Proposed optical emission depletion mechanism for the 455 nm upconversion band of Tm^{3+} of $NaYF_4:18\% Yb^{3+}/10\% Tm^{3+}$ nanoparticles.

(2) Line 126-127. “Subsequently, the ions at the $^3F_{2,3}$ states quickly decay to the 3H_4 state nonradiatively.” How quickly? What is the non-radiative decay lifetime? If it is quick, why Fig. 3d demonstrates strong 700 nm and 1064 nm emission bands due to radiative transitions from $^3F_{2,3}$? What about cross-section for 810-ESA back to 1D_2 ?

Response: We thank the reviewer for the comments. It is well accepted and supported by numerous reports that the ions at the $^3F_{2,3}$ states can quickly decay to the 3H_4 state nonradiatively through a multiphonon relaxation process [48, 78-81]. The multiphonon nonradiative transition rate for this process is dependent on the phonon energy of the crystal lattice, typically in order of $10^5 s^{-1}$ in sodium yttrium fluoride crystals according to Ivanova et al. [79], corresponding to a nonradiative lifetime in order of 10 μs . It should be noted that here “quickly” is a relative term and thus does not mean an infinitely small decay lifetime. The 3F_3 state can still hold a certain amount of population, which is indicated by the emission bands at around 700 nm and 1064 nm [48, 54].

We apologize for the unclear presentation of the spectral data in Figure 3d in the original manuscript, which has caused some misunderstanding that the 700 nm and 1064 nm emission bands were strong. Actually, the

spectra presented in Figure 3d in the original manuscript were individually normalized in each spectral range, including 360-400 nm, 570-615 nm, 680-715 nm, 980-1140 nm and 1300-2100 nm, separated by cutoff marks, in order to maximize the relative intensity changes of different emission bands with and without the 810 nm irradiation. Thus, intensity comparison across these spectral ranges is not meaningful. In fact, the 700 nm emission band is much weaker than the 455 nm emission band. We have included the 700 nm emission in Figure 1b in the revised manuscript to provide an appropriate picture of its relative intensity to other visible emission bands. In addition, we have added clarification for the spectra normalization strategy in the figure legend of Figure 2 in the revised manuscript.

We did not find the value for the cross-section of the $^3F_2 \rightarrow ^1D_2$ ESA process induced by the 810 nm light. However, in general, the cross-section for the ESA process is comparable to that of the corresponding SE process even if they are not identical. In this case, the net effect of the 810 nm beam is determined by the population densities at the 1D_2 and 3F_2 states. In our experiments, the population density at the 3F_2 state under single 975 nm excitation was smaller than that at the 1D_2 state, supported by the relatively weak emission at around 700 nm, thus yielding a net SE effect of the 810 nm beam. This is in accordance with our theoretical explanation.

Change in the manuscript: We have expanded the spectral range in Figure 1b in the revised manuscript to include the emission band at around 700 nm. We have added clarification for the spectra normalization strategy in the figure legend of Figure 2 in the revised manuscript.

(3) Lines 128-130. “The following cross-relaxation process between Tm^{3+} ions contributes to the concentration of excitation energy into the lower 3F_4 state and eventually dissipates it in the form of infrared emission, efficiently padlocking the repopulation of the 1D_2 state under excitation of the 975 nm beam.” I do not buy it unless I see the lifetimes of all the above-mentioned processes.

Response: We thank the reviewer very much for his/her valuable comments, which have led us to a deeper understanding of the mechanism. We have modified the formulation of the mechanism and made revision to the manuscript accordingly. For more details, please refer to our response to previous Comment 1.

(4) Simulating 10 rate equations with >20 unknown parameters including the cross-sections and the lifetimes, is not convincing.

Response: We thank the reviewer very much for his/her comments. Our model and numerical simulation are not intended to quantitatively evaluate the absolute optical depletion efficiency induced by the 810 nm laser, due to the difficulty of experimentally obtaining the parameter values needed (thus they have to be estimated based on the literature). Similar to other reports [10, 82], by simulations, we aimed to qualitatively identify key mechanisms that account for the observed optical emission depletion, by correlating with major

experimental facts. Particularly, in view of the concentration dependence of the emission depletion, we have implemented simulations for low and high Tm^{3+} -doping samples and included the simulated results in Figure S12 in the supplementary information. The contrast between the low (0.5%) and high (10%) Tm^{3+} -doping samples confirms the importance of the concentration-dependent CR1 process ($^3\text{H}_4 + ^3\text{H}_6 \rightarrow ^3\text{F}_4 + ^3\text{F}_4$) in the optical depletion of 455 nm upconversion luminescence (Fig. R24), in accordance with experimental results.

Change in the manuscript: We have included the simulated results of both low and high doping concentrations in Figure S12 in the supplementary information.

Sentence modified (Lines 184-185, page 9): “The critical role of the Tm^{3+} concentration dependent CR1 process is also supported by the results of our numerical simulations (Figs. S11 and S12).”

Figure R24 (Fig. S12 in the revised Supplementary Information) The simulated dependence of the depletion efficiency of the 455-nm luminescence on the power of the 810 nm depletion beam in (a) low and (b) high Tm^{3+} -doping UCNPs.

(5) It can well be that with low Tm^{3+} concentration (low CR rate), re-population rate of the $^1\text{D}_2$ state by both lasers is stronger than de-excitation by only depletion laser, and that is why you have increase of luminescence at low Tm^{3+} concentrations. And with presence of CR, the depletion efficiency comes to its theoretical maximum based purely on the $^1\text{D}_2$ radiative lifetime and the SE cross section. This might actually be not far from the truth based on my next comment.

Response: We thank the reviewer for his/her very valuable comments, which have inspired us to get a deeper understanding of the mechanism. We have modified the formulation of the mechanism and made revision to the manuscript accordingly. For more details, please refer to our response to previous Comment 1.

(6) Lines 119-120. “The saturation excitation intensity I_{sat} (50% emission off) is estimated to be 1.78 MW/cm² (power ~5 mW), which is orders of magnitude lower than that in common STED cases.” This is not correct. The claim should be removed. P_{sat} typically amounts for 10 mW in standard STED microscopy

(Willig et al, STED microscopy with continuous wave beams, Nat Meth, 2007). Moreover, for Tb³⁺ and Eu³⁺ 4f-4f transitions P_{sat} is 10-20 mW as well (Alekhin et al, STED properties of Ce³⁺, Tb³⁺, and Eu³⁺ doped inorganic scintillators, Opt express, 2017).

It can well be that SE cross section of the Tm³⁺ ¹D₂ - ³F_{2,3} transitions and the radiative lifetime of the Tm³⁺ ¹D₂ - ³F₄ transition will alone result in $P_{\text{sat}} = 5$ mW, allowing 96% depletion at 50 mW. All the speculation about locking the electrons at the lower Tm³⁺ levels will then be invalid. Unless the authors have solid proof for the opposite, the claims should not be present in the publication.

Response: As suggested, we have removed the misleading claim regarding the comparison of the saturation power used in our study with those in common STED cases in the revised manuscript. We have reformulated the mechanism and removed the claims about locking the electrons at the lower Tm³⁺ levels. Please refer to our response to Comment 1.

We have repeated the measurement of the saturation intensity and made necessary corrections in the revised manuscript. The saturation intensity (I_s) is 849 kW/cm², and when the depletion efficiency reaches 95%, the power intensity is 17.7 MW/cm².

Change in the manuscript: The claim “which is orders of magnitude lower than that in common STED cases” has been removed in the revised manuscript. We have reformulated the mechanism and removed the claims about locking the electrons at the lower Tm³⁺ levels.

Sentence modified (lines 116-120, page 6.): “The degree of luminescence inhibition was measured for various values of I_{810} (Fig. 1g). About 4% residual luminescence can be detected for $I_{810} = 17.7$ MW/cm², confirming the effectiveness of the luminescence inhibition process. The saturation excitation intensity I_{sat} , inducing 50% emission off, is estimated to be 849 kW/cm² (power ~2.4 mW).”

References added:

Alekhin, M. S. *et al.* STED properties of Ce³⁺, Tb³⁺, and Eu³⁺ doped inorganic scintillators. *Opt. Express* **25**, 1251-1261, (2017).

Alekhin, M. S.; Patton, G.; Dujardin, C.; Douissard, P. A.; Lebugle, M.; Novotny, L.; Stampanoni, M., Stimulated scintillation emission depletion X-ray imaging. *Opt. Express* **25**, 654-669 (2017).

(7) Lines 64-65. “we propose a novel low-energy state trap assisted mechanism for emission depletion (Fig. 1d)”. The previous work of the authors (Wu et al, “Optical depletion mechanism of upconverting luminescence and its potential for multi-photon STED-like microscopy”, Opt. express, 2015), where they investigated similar mechanisms of similar class of UCNPs, is not cited. This questions a bit the novelty of the current research.

Response: We thank the reviewer for the comments. We would like to clarify that our previous work (Wu et al., Opt. Express 2015, 23(25), 32401-32412) was conducted in a completely different upconversion system,

NaYF₄:Yb³⁺/Er³⁺, where Er³⁺ is the emitting ion which gives all spectroscopic features, and reported an optical depletion mechanism based on **excited state absorption**, which is intrinsically different from the mechanism we proposed in the present study. In addition, the optical depletion of the green emission (excited at 795 nm) induced by the depletion laser at 1140 nm in our previous work, although substantial (~30% emission off), needed to be improved for ease of subsequent applications. We did not manage to implement super-resolution microscopy in the previous work.

In order to provide a better overview for the history of the development of optical depletion approach of upconversion nanomaterials, we have cited our previous work in the revised manuscript and modified the Introduction section accordingly.

Change in the manuscript:

Introduction modified (lines 44-63, page 3): “Lanthanide UCNPs have been developed as an important group of luminescent biomarkers during the last decade. UCNPs convert low-intensity near-infrared (NIR) excitation to shorter-wavelength NIR, visible and ultraviolet (UV) emissions. Such unique luminescence properties of UCNPs enable superior bioimaging without many of the constraints associated with conventional optical biomarkers, including photobleaching, photoblinking, tissue autofluorescence, limited imaging depth and high light toxicity^{24, 25, 26}. However, efficient optical modulation of UCNPs has not been well established, which hinders their use in advanced luminescence imaging techniques, such as STED microscopy. In the few studies on optical modulation of UCNPs, highly efficient optical depletion of upconversion luminescence has rarely been reported in major groups of UCNPs (NaYF₄:Yb³⁺, Er³⁺/Tm³⁺/Ho³⁺) that provide bright luminescence upon NIR excitation, except a demonstration of optical inhibition of upconversion luminescence in low efficient YAG:Pr³⁺ nanoparticles²⁷, emitting toxic UV light upon visible excitation with limited applications in biomedical areas. The research efforts exploring co-irradiation by additional wavelength(s) have usually resulted in luminescence enhancement rather than silencing due to the presence of complicated photon recycling pathways among the lanthanide dopants under multi-wavelength excitation^{28, 29}. In our previous study on another upconversion system (NaYF₄:Yb³⁺,Er³⁺), the first optical depletion of Er³⁺ emission was shown by inducing excited state absorption, intrinsically different from the mechanism we proposed in the present study, but the efficiency needed improvement for applications³⁰.”

Reference added:

Wu, R. *et al.*, Optical depletion mechanism of upconverting luminescence and its potential for multi-photon STED-like microscopy. *Opt. Express* **23**(25), 32401-32412, (2015).

(8) Line 124. “A six-photon upconversion process into ¹D₂ level” In Fig 3a, only a four-photon process is illustrated.

Response: We apologize for this typo in the original manuscript. Actually, we hold the view that the 1D_2 state is populated through a five-photon process under single 975 nm excitation, which is well supported by many reports [54, 59-61]. In this process, the 3H_4 and 1G_4 states are first populated through a two- and three-photon process, respectively. Subsequent cross relaxations between the ions at the 3H_4 and 1G_4 states, including $^3H_4 + ^1G_4 \rightarrow ^3F_4 + ^1D_2$ and $^3H_4 + ^1G_4 \rightarrow ^1D_2 + ^3F_4$, populate the 1D_2 state, exhibiting a five-photon process. The five-photon upconversion process into the 1D_2 state is well supported by the excitation power dependence of the 455 nm emission under single 975 nm excitation, as shown in Figure R25. The upconversion mechanism has been modified accordingly in Figure 2a in the revised manuscript.

Figure R25 The excitation power dependence of the 455 nm emission under single 975 nm excitation

Change in the manuscript:

Sentence modified to (lines 126-128, page 6-7): “As illustrated in Fig. 2a, the 1D_2 emitting state, is populated by the 975-nm excitation beam through a five-photon upconversion process^{36,37,38},”

Figure 2a is modified in the revised manuscript.

References added:

Chen, G. Y., Somesfalean, G., Zhang, Z. G., Sun, Q. & Wang, F. P. Ultraviolet upconversion fluorescence in rare-earth-ion-doped Y_2O_3 induced by infrared diode laser excitation. *Opt. Lett.* **32**, 87-89 (2007).

Chen, X. & Song, Z. Study on six-photon and five-photon ultraviolet upconversion luminescence. *J. Opt. Soc. Am. B* **24**, 965-971 (2007).

Zhang, H. *et al.*, Mechanisms of the blue emission of $NaYF_4:Tm^{3+}$ nanoparticles excited by an 800 nm continuous wave laser. *Phys. Chem. Chem. Phys.* **18**, 25905-25914 (2016).

(9) Lines 178-179. “A factor of 8 improvement to 66 nm”. It means that without the depletion beam the resolution was 530 nm. More or less the same is demonstrated in Fig. 4. Theoretically, $R=\lambda/2NA$ is 160 nm for 455 nm emission and 1.35NA objective used in the experiment. How comes then 530 nm resolution?

Response: We apologize for the improper resolution comparison in the original manuscript. In Fig. 4c in the original manuscript (Fig. 3c in the revised manuscript), two neighboring nanoparticles were imaged. For a fair comparison, the transverse line profile (~ 340 nm FWHM) instead of the longitudinal one (~530 nm FWHM) should be extracted and analyzed to evaluate/compare the point spread functions of two imaging modes (super-resolution and non-super-resolution).

In addition, in laser scanning microcopy the resolution is determined by the point spread function of the excitation (975 nm) beam instead of the wave nature of the emission wavelength (455 nm), approximated by the equation $R=0.61\lambda/NA$. In the present study employing multi-photon imaging of UCNPs, the derivative equation would be $R=0.61\lambda/\sqrt{N}NA$, with N denoting the N -photon excitation process. In our case, the excitation wavelength was 975 nm and the NA was 1.35. Considering saturated excitation (700 kW/cm² in the imaging), the slope of power dependence probably changes from theoretical value 5 to 2-4. Taking a slope of 3.5 for example, multiphoton imaging with, i.e., $d=0.61\lambda/\sqrt{3.5}NA$ would give a theoretical resolution of ~236 nm. The experimentally achieved ~340 nm resolution in our study is reasonable because of imperfect imaging and underutilized NA of objective.

Change in the manuscript:

Claim removed: “A factor of 8 improvement to 66 nm” was removed in the revised manuscript.

(10) Lines 46-49. Trying to explain the main super-resolution methods in one sentence confuses more than explains. The statement in line 48 contradicts the principles of cw-STED.

Response: We thank the reviewer for the comments. We have modified the discussion on super-resolution methods to make it more precise.

Change in the manuscript:

Sentences modified (lines 33-43, page 3): “Exploitation of optical switching to control the transition of luminophores between two optically distinguishable states (on/off) is an exciting and popular way to circumvent the diffraction limit^{13, 14}. This approach has led to many far field super-resolution fluorescence microscopy techniques, such as stimulated emission depletion microscopy (STED)^{15, 16, 17}, single molecule localization nanoscopy (PALM/STORM, MINFLUX)^{18, 19, 20}, and saturated structured illumination microscopy^{21, 22}, which breaks the diffraction limit and allow for imaging at a spatial resolution of down to a few nanometers. A major category of these nanoscopy methods, including pulse STED and PALM/STORM, provides sub-diffraction resolution essentially by transiently switching the luminophores between the bright

“on” and dark “off” states, causing those within the same diffraction range to emit successively, rather than simultaneously²³.”

(11) Lines 58-63. “...of great significance to create efficient optical depletion pathways”. You mentioned light toxicity, photobleaching, but what about STED-parallelization (“Nanoscopy with more than 100,000 'doughnuts’”, *Nat Meth*, Chmyrov, 2013). Faster imaging is also a very good motivation.

Response: We thank the reviewer for the suggestion. We have added discussion and reference about STED-parallelization in the revised manuscript.

Change in the manuscript:

Sentence added (lines 231-234, page 11): “The pixel dwell time (100 μ s) employed in the laser scanning super-resolution imaging is comparable to that used in traditional STED imaging (50-200 μ s), and the image acquisition speed can be increased with STED-parallelization strategies⁴⁵.”

Reference added:

Chmyrov, A. *et al.* Nanoscopy with more than 100,000 'doughnuts'. *Nat. Meth.* **10**, 737-740, (2013).

(12) It would be much easier to follow the articles if the following is implemented:

- Numbering all the relevant transitions and referring to those numbers in the text.
- Perhaps making 1-2 large electron-energy diagrams instead of more smaller ones.
- Lines 50-75. Make text and illustration more consistent with each other. Emitting/dissipating states with S1/S0 in illustration.

Response: We thank the reviewer for the comments and advice. We have numbered all the relevant transitions and cross relaxations and referred to those numbers in the text. As well, we have modified the energy diagram for the depletion process (Fig. 2a in the revised manuscript).

We have merged electron-energy diagrams as much as possible in the revised manuscript, as shown in Fig. R26 (Fig. 2a in the revised manuscript) and Fig. S13 in the revised manuscript. However, we keep others in the present form, because they represent different conditions.

Taking the reviewer’s advice, we have moved Figure 1 and the associated text in the original manuscript to the supplementary information as Fig. S1, after necessary modification to make it more precise and understandable.

Figure R26 (Fig. 2a in the revised manuscript) Proposed optical emission depletion mechanism of the 455 nm upconversion band of Tm^{3+} of $\text{NaYF}_4:18\% \text{Yb}^{3+}/10\% \text{Tm}^{3+}$ nanoparticles.

General comments:

All-in-all, the presented idea and the experimental results are of considerable interest and worth for publishing. But either the claims should be reduced, or they should be supported by stronger data/analysis to be published.

Response: We thank the reviewer for his/her recommendation and comments. We appreciate his/her specific comments again, which particularly have led us to a deeper understanding on the mechanism. We have modified the formulation of the mechanism and claims accordingly in the revised manuscript.

In addition, we have made a lot of effort in the super-resolution bio-imaging using UCNPs and finally achieved and updated the biological super-resolution imaging data in Figure 4 in the revised manuscript. We have improved the nanoparticle synthesis, surface modification, and antibody bioconjugation, and have successfully achieved high quality immunolabelling of the cytoskeleton protein desmin using secondary antibody conjugated UCNPs (HeLa cancer cell; primary antibody Anti-desmin antibody (rabbit), secondary antibody: goat anti-rabbit IgG antibody), imaged using the upconversion nanoscopy technique developed in this study (Figure R27). The line profile analysis for a local area indicates that the lateral imaging resolution is increased to 82 nm, less than $\lambda/10$ ($\lambda_{\text{ex}}=975$ nm) (Figure R27). Cytoskeleton protein labeling with inorganic nanoparticles is generally challenging and had never been achieved using UCNPs previously. To the best of our knowledge, this is the first demonstration of immunolabelling of fine subcellular structures using UCNPs, as well as the first demonstration of super-resolution bio-imaging using UCNPs.

Figure 27 (Fig. 4 in the revised manuscript) **Immunofluorescence labeling of cellular cytoskeleton protein desmin with antibody conjugated UCNP and super-resolution imaging.** (a) The multiphoton imaging under 975 nm excitation of some cytoskeleton structures and desmin proteins in HeLa cancer cells incubated with anti-desmin primary antibody and immunostained with UCNP (~11.8 nm in diameter) bioconjugated with goat Anti-rabbit IgG secondary antibody. (b) The same region with (a) imaged in the super-resolution mode (975 nm excitation and the 810 nm STED laser beam), Scale bar: 2 μm . (c-n) Magnified areas selected from a, b (marked by white dotted squares) and line profile analyses; Images in c, f, i and l are taken from the white dotted squares in a; Images in d, g, j and m are taken from the white dotted squares in b; (e, h, k, n) Line profiles analyses of several areas indicated by arrow heads in c and d, f and g, i and j, and l and m, respectively.

We hope that the manuscript now fulfills the criteria of importance and interest to be published in Nature Communications.

On behalf of all authors,

Sailing He

Professor, Fellow of IEEE, SPIE, OSA and EA Academy

School of Electrical Engineering, Royal Institute of Technology (KTH)

S-100 44 Stockholm, SWEDEN

<http://www.ee.kth.se/>

Tel: +46-8-7908465

E-mail: sailing@kth.se

Chief scientist

JRCEP [Joint Research Center of Photonics of KTH (Sweden), Lund University (Sweden) and Zhejiang University(ZJU) (China)]

References:

- [1] B. Zhou, B. Shi, D. Jin, X. Liu, *Nat. Nano.*, 10 (2015) 924-936.
- [2] S. Wu, G. Han, D.J. Milliron, S. Aloni, V. Altoe, D.V. Talapin, B.E. Cohen, P.J. Schuck, *Proc. Natl. Acad. Sci. U. S. A.*, 106 (2009) 10917-10921.
- [3] C. Wang, A. Fukazawa, M. Taki, Y. Sato, T. Higashiyama, S. Yamaguchi, *Angewandte Chemie*, 127 (2015) 15428-15432.
- [4] J.-i. Hotta, E. Fron, P. Dedecker, K.P. Janssen, C. Li, K. Müllen, B. Harke, J. Bückers, S.W. Hell, J. Hofkens, *Journal of the American Chemical Society*, 132 (2010) 5021-5023.
- [5] R. Strack, *Nat Meth*, 13 (2016) 196-197.
- [6] J.G. Danzl, S.C. Sidenstein, C. Gregor, N.T. Urban, P. Ilgen, S. Jakobs, S.W. Hell, *Nature Photonics*, 10 (2016) 122-128.
- [7] Y. Wu, X. Wu, L. Toro, E. Stefani, *Biophysical Journal*, 108 (2014) 477a.
- [8] J. Hanne, H.J. Falk, F. Gorkitz, P. Hoyer, J. Engelhardt, S.J. Sahl, S.W. Hell, *Nat Commun*, 6 (2015).
- [9] Y. Il Park, J.H. Kim, K.T. Lee, K.S. Jeon, H. Bin Na, J.H. Yu, H.M. Kim, N. Lee, S.H. Choi, S.I. Baik, H. Kim, S.P. Park, B.J. Park, Y.W. Kim, S.H. Lee, S.Y. Yoon, I.C. Song, W.K. Moon, Y.D. Suh, T. Hyeon, *Adv. Mater.*, 21 (2009) 4467-4471.
- [10] Y. Liu, Y. Lu, X. Yang, X. Zheng, S. Wen, F. Wang, X. Vidal, J. Zhao, D. Liu, Z. Zhou, C. Ma, J. Zhou, J.A. Piper, P. Xi, D. Jin, *Nature*, 543 (2017) 229-233.
- [11] J. Tønnesen, F. Nadrigny, Katrin I. Willig, R. Wedlich-Söldner, U.V. Nägerl, *Biophysical Journal*, 101 (2011) 2545-2552.
- [12] G. Moneron, S.W. Hell, *Optics express*, 17 (2009) 14567-14573.
- [13] N.G. Horton, K. Wang, D. Kibat, C.G. Clark, F.W. Wise, C.B. Schaffer, C. Xu, *Nature photonics*, 7 (2013) 205-209.
- [14] D.J. Gargas, E.M. Chan, A.D. Ostrowski, S. Aloni, M.V.P. Altoe, E.S. Barnard, B. Sanii, J.J. Urban, D.J. Milliron, B.E. Cohen, P.J. Schuck, *Nat. Nano.*, 9 (2014) 300-305.
- [15] Q. Liu, Y. Sun, T. Yang, W. Feng, C. Li, F. Li, *J. Am. Chem. Soc.*, 133 (2011) 17122-17125.
- [16] W. Zheng, S.Y. Zhou, Z. Chen, P. Hu, Y.S. Liu, D.T. Tu, H.M. Zhu, R.F. Li, M.D. Huang, X.Y. Chen, *Angew. Chem. Int. Ed.*, 52 (2013) 6671-6676.
- [17] A.D. Ostrowski, E.M. Chan, D.J. Gargas, E.M. Katz, G. Han, P.J. Schuck, D.J. Milliron, B.E. Cohen, *ACS Nano*, 6 (2012) 2686-2692.

- [18] J. Liu, G. Chen, S. Hao, C. Yang, *Nanoscale*, 9 (2016) 91-98.
- [19] Y. Dong, C. Shannon, *Analytical chemistry*, 72 (2000) 2371-2376.
- [20] R.A. Sperling, W.J. Parak, *Philosophical Transactions of the Royal Society A: Mathematical, Physical and Engineering Sciences*, 368 (2010) 1333-1383.
- [21] X. Yang, Y.-K. Tzeng, Z. Zhu, Z. Huang, X. Chen, Y. Liu, H.-C. Chang, L. Huang, W.-D. Li, P. Xi, *RSC Advances*, 4 (2014) 11305-11310.
- [22] Y. Sonnefraud, H.G. Sinclair, Y. Sivan, M.R. Foreman, C.W. Dunsby, M.A. Neil, P.M. French, S.A. Maier, *Nano letters*, 14 (2014) 4449-4453.
- [23] Y. Sivan, Y. Sonnefraud, S.p. Kéna-Cohen, J.B. Pendry, S.A. Maier, *ACS nano*, 6 (2012) 5291-5296.
- [24] E. Cortés, P.A. Huidobro, H.G. Sinclair, S. Guldbrand, W.J. Peveler, T. Davies, S. Parrinello, F. Görlitz, C. Dunsby, M. Neil, *ACS nano*, 10 (2016) 10454-10461.
- [25] R.A. Sperling, W.J. Parak, *Philosophical Transactions of the Royal Society A: Mathematical, Physical and Engineering Sciences*, 368 (2010) 1333-1383.
- [26] Q. Zhan, J. Qian, H. Liang, G. Somesfalean, D. Wang, S. He, Z. Zhang, S. Andersson-Engels, *ACS Nano*, 5 (2011) 3744-3757.
- [27] J.C. Boyer, M.P. Manseau, J.I. Murray, F.C.J.M. van Veggel, *Langmuir*, 26 (2010) 1157-1164.
- [28] S.A. Hilderbrand, F. Shao, C. Salthouse, U. Mahmood, R. Weissleder, *Chemical Communications*, (2009) 4188-4190.
- [29] H.P. Zhou, C.H. Xu, W. Sun, C.H. Yan, *Adv. Funct. Mater.*, 19 (2009) 3892-3900.
- [30] R. Naccache, F. Vetrone, V. Mahalingam, L.A. Cuccia, J.A. Capobianco, *Chem. Mater.*, 21 (2009) 717-723.
- [31] H.S. Qian, Y. Zhang, *Langmuir*, 24 (2008) 12123-12125.
- [32] H.S. Qian, H.C. Guo, P.C.L. Ho, R. Mahendran, Y. Zhang, *Small*, 5 (2009) 2285-2290.
- [33] O. Ehlert, R. Thomann, M. Darbandi, T. Nann, *ACS Nano*, 2 (2008) 120-124.
- [34] J. Lee, T.S. Lee, J. Ryu, S. Hong, M. Kang, K. Im, J.H. Kang, S.M. Lim, S. Park, R. Song, *J. Nucl. Med.*, 54 (2013) 96-103.
- [35] X.F. Yu, Z.B. Sun, M. Li, Y. Xiang, Q.Q. Wang, F.F. Tang, Y.L. Wu, Z.J. Cao, W.X. Li, *Biomaterials*, 31 (2010) 8724-8731.
- [36] Z. Chen, H. Chen, H. Hu, M. Yu, F. Li, Q. Zhang, Z. Zhou, T. Yi, C. Huang, *J. Am. Chem. Soc.*, 130 (2008) 3023-3029.
- [37] L.L. Li, P.W. Wu, K. Hwang, Y. Lu, *J. Am. Chem. Soc.*, 135 (2013) 2411-2414.
- [38] L.-L. Li, Y. Lu, *J. Am. Chem. Soc.*, 137 (2015) 5272-5275.
- [39] B. Harke, J. Keller, C.K. Ullal, V. Westphal, A. Schönle, S.W. Hell, *Optics Express*, 16 (2008) 4154-4162.
- [40] M.G.L. Gustafsson, *Journal of Microscopy-Oxford*, 198 (2000) 82-87.

- [41] MudryE, BelkebirK, GirardJ, SavatierJ, E. Le Moal, NicolettiC, AllainM, SentenacA, *Nature Photonics*, 6 (2012) 312-315.
- [42] C.T. Xu, P. Svenmarker, H. Liu, X. Wu, M.E. Messing, L.R. Wallenberg, S. Andersson-Engels, *ACS Nano*, 6 (2012) 4788-4795.
- [43] H. Liu, C.T. Xu, G. Dumlupinar, O.B. Jensen, P.E. Andersen, S. Andersson-Engels, *Nanoscale*, 5 (2013) 10034-10040.
- [44] L.M. Maestro, E.M. Rodriguez, F. Vetrone, R. Naccache, H.L. Ramirez, D. Jaque, J.A. Capobianco, J.G. Solé, *Opt. Express*, 18 (2010) 23544-23553.
- [45] C.T. Xu, N. Svensson, J. Axelsson, P. Svenmarker, G. Somesfalean, G. Chen, H. Liang, H. Liu, Z. Zhang, S. Andersson-Engels, *Appl. Phys. Lett.*, 93 (2008) 171103-171103.
- [46] M. Nyk, R. Kumar, T.Y. Ohulchanskyy, E.J. Bergey, P.N. Prasad, *Nano Lett.*, 8 (2008) 3834-3838.
- [47] I. Coto Hernández, C. Peres, F. Cella Zanacchi, M. d'Amora, S. Christodoulou, P. Bianchini, A. Diaspro, G. Vicidomini, *Journal of Biophotonics*, 7 (2014) 376-380.
- [48] J. Zhao, D. Jin, E.P. Schartner, Y. Lu, Y. Liu, A.V. Zvyagin, L. Zhang, J.M. Dawes, P. Xi, J.A. Piper, E.M. Goldys, T.M. Monro, *Nat. Nano.*, 8 (2013) 729-734.
- [49] T.G. Iversen, T. Skotland, K. Sandvig, *Nano Today*, 6 (2011) 176-185.
- [50] F. Balzarotti, Y. Eilers, K.C. Gwosch, A.H. Gynnå, V. Westphal, F.D. Stefani, J. Elf, S.W. Hell, *Science*, (2016).
- [51] C.A. Morrison, R.P. Leavitt, *Handbook on the Physics & Chemistry of Rare Earths*, 5 (1982) 461-692.
- [52] J.B. Gruber, R.P. Leavitt, C.A. Morrison, *Journal of Chemical Physics*, 74 (1981) 2705-2709.
- [53] D.A. Simpson, W.E.K. Gibbs, S.F. Collins, W. Blanc, B. Dussardier, G. Monnom, P. Peterka, G.W. Baxter, *Optics Express*, 16 (2008) 13781-13799.
- [54] H. Zhang, T. Jia, X. Shang, S. Zhang, Z. Sun, J. Qiu, *Phys. Chem. Chem. Phys.*, 18 (2016) 25905-25914.
- [55] R. O'Connor, R. Mahiou, D. Martinant, M.T. Fournier, *Journal of Alloys and Compounds*, 225 (1995) 107-110.
- [56] R.C. Stoneman, L. Esterowitz, *IEEE Journal of Selected Topics in Quantum Electronics*, 1 (1995) 78-81.
- [57] X. Wen, G. Tang, Q. Yang, X. Chen, Q. Qian, Q. Zhang, Z. Yang, *Scientific Reports*, 6 (2016) 20344.
- [58] R. Stoneman, L. Esterowitz, *Optics letters*, 15 (1990) 486-488.
- [59] G.Y. Chen, G. Somesfalean, Z.G. Zhang, Q. Sun, F.P. Wang, *Optics Letters*, 32 (2007) 87-89.
- [60] X. Chen, Z. Song, *Journal of the Optical Society of America B*, 24 (2007) 965-971.
- [61] G. Wang, W. Qin, L. Wang, G. Wei, P. Zhu, R. Kim, *Opt. Express*, 16 (2008) 11907-11914.
- [62] H. Zhang, Y. Li, Y. Lin, Y. Huang, X. Duan, *Nanoscale*, 3 (2011) 963-966.
- [63] G.Y. Chen, G. Somesfalean, Z.G. Zhang, Q. Sun, F.P. Wang, *Opt. Lett.*, 32 (2007) 87-89.

- [64] J. Zhou, G. Chen, E. Wu, G. Bi, B. Wu, Y. Teng, S. Zhou, J. Qiu, *Nano Lett.*, 13 (2013) 2241-2246.
- [65] G.H. Dieke, R.A. Satten, *American Journal of Physics*, 38 (1970) 525-525.
- [66] J.B. Gruber, W.F. Krupke, J.M. Poindexter, *Journal of Chemical Physics*, 41 (1964) 3363-3377.
- [67] T.R. Faulkner, *Molecular Physics*, 38 (1979) 1767-1780.
- [68] V.A. Antonov, P.A. Arsenev, K.E. Bienert, A.V. Potemkin, *Physica Status Solidi*, 19 (2010) 289-299.
- [69] S. Svanberg, *Atomic and Molecular Spectroscopy-Basic Aspects and Practical Applications*, Springer-Verlag Berlin Heidelberg GmbH 2004.
- [70] D. Gamelin, H. Gudel, *Upconversion Processes in Transition Metal and Rare Earth Metal Systems*, in: H. Yersin (Ed.) *Transition Metal and Rare Earth Compounds*, Springer Berlin/Heidelberg 2001, pp. 1-56.
- [71] P. Loiko, M. Pollnau, *J. Phys. Chem. C*, 120 (2016) 26480-26489.
- [72] M. Leutenegger, C. Eggeling, S.W. Hell, *Opt. Express*, 18 (2010) 26417-26429.
- [73] R. Wu, Q. Zhan, H. Liu, X. Wen, B. Wang, S. He, *Opt. Express*, 23 (2015) 32401-32412.
- [74] J.F. Suyver, A. Aebischer, S. Garcíia-Revilla, P. Gerner, H.U. Güdel, *Phys. Rev. B*, 71 (2005) 125123-125123.
- [75] M. Pollnau, D.R. Gamelin, S.R. Lüthi, H.U. Güdel, M.P. Hehlen, *Phys. Rev. B*, 61 (2000) 3337-3346.
- [76] A.V. Smith, J.J. Smith, *Opt. Express*, 24 (2016) 975-992.
- [77] X. Wen, G. Tang, Q. Yang, X. Chen, Q. Qian, Q. Zhang, Z. Yang, *Scientific Reports*, 6 (2016) 20344-20344.
- [78] F. Wang, R. Deng, J. Wang, Q. Wang, Y. Han, H. Zhu, X. Chen, X. Liu, *Nat. Mater.*, 10 (2011) 968-973.
- [79] S.E. Ivanova, A.M. Tkachuk, A. Mirzaeva, F. Pelle, *Optics and Spectroscopy*, 105 (2008) 228-241.
- [80] H. Zhang, T. Jia, X. Shang, S. Zhang, Z. Sun, J. Qiu, *Physical Chemistry Chemical Physics Pccp*, 18 (2016).
- [81] G. Wang, W. Qin, L. Wang, G. Wei, P. Zhu, R. Kim, *Optics express*, 16 (2008) 11907-11914.
- [82] R. Deng, F. Qin, R. Chen, W. Huang, M. Hong, X. Liu, *Nat. Nano.*, 10 (2015) 237-242.

Reviewers' comments:

Reviewer #2 (Remarks to the Author):

In this revised version, the authors have significantly improved the quality of this work by the super resolution imaging of cellular cytoskeleton protein structures, however, the fundamental in describing the depletion mechanism was wrong. In order to help this work to become eventually publishable in the prestigious journal, Nature Communications, my team has spent some significant amount of time on carefully checking many details of this revision. Major revision is still required. see our detailed 5 page comments attached.

Reviewer #3 (Remarks to the Author):

In the revision, the authors have addressed my previous comments adequately. More impressively, they have managed to add the cell skeleton image of UCNPs, by synthesizing much smaller UCNPs (11.8 nm). This has given a clear answer to the long standing question of whether UCNPs can be used as an alternative dye for cellular imaging. With the significant improvement of the revision, I am pleased to accept the manuscript at its current form.

Reviewer #4 (Remarks to the Author):

My recommendations and critics were mostly dealt by the authors in the revised version. The reading was substantially improved, although still some ambiguities are present. I am also very grateful for several clarifications and explanations the authors made regarding their work. I still insist on making changes in the manuscript. Main de-excitation mechanism is still not clearly described. Some of my comments are new, based on new decay time measurements, and based on comparison with the work of Reviewer 2 (Nature 2017 doi:10.1038/nature21366), which I was not aware of during my first review process.

1. In the revised version, authors removed incorrect claim that Isat of their system is orders of magnitude lower than that in common STED cases.

First, I was probably unclear here, but I meant to remove incorrect claim "orders of magnitude lower", but not the comparison itself. I suggest to leave the comparison of $\text{NaYF}_4:\text{Yb}^{3+}, \text{Tm}^{3+}$ Isat with Isat of previously reported systems (dyes, proteins, lanthanides, vacancies), and mention Isat values from multiple references. In the work of Reviewer 2, Isat is 4.5 smaller. This also should be cited in your work. By the way, such discrepancy (and also others in experimental results) is probably caused not by different NPs grown under different conditions (Fig.S5 points at minimal quantum confinement effect), but rather because of the use of different depletion lasers (810 vs 808 nm) with different emission bandwidths, which might easily mismatch narrow Tm^{3+} absorption-emission bands.

Second, another misleading claim remained in the manuscript in Lines 197-199: "The power densities for the two NIR beams were several orders of magnitude lower than those used in typical STED imaging (in order of GW/cm^2)⁴³". In the case of the reference 43, the applied power was indeed huge, but also the resolution achieved was <25 nm. For 66 nm resolution (achieved by the authors), lower STED-laser powers was used. I insist to remove the statement "several order of magnitude", be more precise in comparison, and compare with multiple reports.

After all, one of the main abstract claims, lines 17-19 "we show efficient emission depletion ... which significantly lowers the laser intensity requirements of optical depletion" relies on these claims.

2. Lines 49-50: the work of Reviewer 2 needs to be cited here.

3. Claims in the abstract Lines 17-19, "with assistance of interionic cross relaxation, which significantly lowers the laser intensity requirements"

and in the introduction, lines 66-72, "We establish an efficient optical depletion approach...by enhancing the inter-Tm³⁺ interaction."

are apparently leading the reader into the wrong direction.

Based on authors explanation of the experimental results on pages 7-9, I came to the conclusion that the solely depleting action of the 810nm laser – is de-excitation of the 1D₂ state via stimulated emission to the 3F₂ state. From their response to my comment(1) "With high Tm³⁺ concentrations, the highly efficient CR1 process suppresses the synergistic excitation effect and simultaneously amplifies the depletion effect of the 810 nm beam by transferring the electrons at 3H₄ state to the 3F₄", it is clear that CR1 does not de-excite 1D₂ state, but de-excites 3H₄ state, facilitating faster de-population of the latter. Main point is, either with only 975 nm or with both 975+810 nm beams applied, the CR1 process will have the same effect on the de-/re-population rates of the energy states involved. The 1D₂ will be re-populated at the same or higher rate in the presence of 810 nm beam. If the only process de-exciting the 1D₂ state is the stimulated emission to the 3F₂ state, then there should not be any claims that the depletion efficiency in NaYF₄:Yb³⁺,Tm³⁺ is better compared to conventional STED (light-matter interaction), due to matter-matter interaction. The depletion is more efficient then solely due to better combination of Tm³⁺ 1D₂ - 3F₄ SE cross-section and radiative emission lifetime compared to those of STED-proteins and dyes. Then it is advisable to put some literature survey on those values for Tm³⁺ transitions. The claims in lines 17-19 and 66-72 need to be re-formulated accordingly.

If there are more 1D₂ de-excitation mechanisms involved, they need to be clearly stated and explained. Basically this is the same comment as I originally did.

4. New decay time measurements of Fig. S6 and Table S1 reveal interesting details. I agree with other reviewers that the decay times under the additional 810 nm irradiation should be substantially shorter (say ~10 us for 95% SE depletion, assuming ~200us radiative lifetime of the 1D₂ state). Step-wise pumping nature might affect the measured decay time, but the drop of the decay time from 200us to 8 us with increase in Tm concentration points on a strong cross-relaxation process, the one that is described in the work of Reviewer 2 as CR3. If cumulative decay time due to radiative emission and cross-relaxation is $\tau = 200\text{us}/8\text{us}$ for 0.5%/20% Tm, the decay time due to those + 810 nm depletion is $150\text{us}/2.3\text{us}$ for 0.5%/20% Tm, then following the $1/\tau = 1/\tau_1 + 1/\tau_2$ equation, the decay time due to the 810-depletion alone drastically drops from 630 us to 3 us for 0.5%/20% Tm, whereas it should be independent on Tm concentration. I doubt that step-wise pumping nature can explain this 200-fold variation.

5. Regarding the response of the authors to my comment (4) "Our model and numerical simulation are not intended to quantitatively evaluate the absolute optical depletion efficiency induced by the 810 nm laser, due to the difficulty of experimentally obtaining the parameter values needed (thus they have to be estimated based on the literature)."

I agree that it is difficult to measure those constants, even literature values are often vary quite a lot. Nevertheless, I recommend tabulating all the constants the authors used in these simulations, and where possible, compare those constants with literature values.

6. Line 179 (Fig.2b). Is it a misspell, did you mean Fig. 2e,f?

In this revised version, the authors have significantly improved the quality of this work by the super resolution imaging of cellular cytoskeleton protein structures, however, the fundamental in describing the depletion mechanism was wrong. In order to help this work to become eventually publishable in the prestigious journal, Nature Communications, my team has spent some significant amount of time on carefully checking many details of this revision. Major revision is still required.

Three main credits of this work should eventually justify its publication into Nature Communications:

1. Demonstration of the second color from Tb doped core-shell upconversion nanoparticles using energy migration effect, this suggests the **potential** for multi-color super resolution using one pair of excitation (975 nm) and depletion (810 nm) beams;
2. Broad spectrum (deep UV and near infrared to infrared) measurement;
3. Super resolution imaging of cellular cytoskeleton protein structures with high speed (100 us dwelling time)

The major issue of the stimulated emission depletion at $1D_2$ level remains incorrect:

In response to our early comments on the mechanism (and referees #3 and 4's comments): *"We agree that 810 nm light does not match with the bandgap of $1D_2 \rightarrow 3F_3$. However, the matching of the $1D_2 \rightarrow 3F_2$ transition with light near 800 nm is supported by many previous reports [51-54]. According to the theoretical calculations and experimental measurements under ultralow temperature (~ 10 K), the $1D_2 \rightarrow 3F_2$ transition is centered around 800 nm and could cover quite a few nanometers [65-68]. Furthermore, the FWHM of $1D_2 \rightarrow 3F_2$ transition spectrum would be broadened under ambient temperature (~ 300 K) and transient high temperature (under illumination of tightly focused light) [69]. In addition, in our experiments the depletion laser is centered at 810 nm and the FWHM is 4 nm, making the matching with the $1D_2 \rightarrow 3F_2$ transition accessible. It should be noted that the $1D_2 \rightarrow 3F_2$ transition is very easy to be overlooked and was indeed often overlooked in the literature probably due to the following reasons: (a) A common misconception exists that the $3F_2$ and $3F_3$ states can be treated as a single state (usually denoted as $3F_{2,3}$) while ignoring their difference because of the fact that the ions at the $3F_2$ state quickly decays to the $3F_3$ state nonradiatively; (b) The transition $1D_2 \rightarrow 3F_2$ spectrally overlap with other transitions, including $3H_4 \rightarrow 3H_6$ and $1G_4 \rightarrow 3H_5$, [51-53] making it barely distinguishable from others and thus causing negligence."*

We have carefully checked their cited references, and found that the authors misinterpreted the details in the literatures. Most of these references are found irrelevant.

- 1) Ref [65] "G.H. Dieke, R.A. Satten, American Journal of Physics, 38 (1970) 525-525". We couldn't find this paper. The correct one should be "G.H. Dieke, R.A. Satten, American Journal of Physics, 38 (1970) 399". This is a book review, it is hard to get supporting information related to the Tm^{3+} transitions.
- 2) According to ref [66], the energy of $Tm^{3+}:1D_2$ level equals to 27522.9 cm^{-1} , $3F_2$ 14854.9 cm^{-1} in Y_2O_3 . From this data, we can calculate the energy gap of $1D_2 \rightarrow 3F_2$ transition to be 12668 cm^{-1} , equaling to 789.4 nm. It is 20 nm gap between 790 nm and 810 nm, not a few nanometers, impossible to be caused by temperature broadening.
- 3) Ref [67] is about $CsNaLnCl_6$ system that has the cubic elpasolite structure at room temperature (space group $Fm\bar{3}m$) with the Ln^{3+} ion situated at a site of six coordinate octahedral (O_h) symmetry. But in the current work, $NaYF_4$ crystals have hexagonal structure, where Ln^{3+} ion situated at a site of nine coordinate C_{3h} or C_s symmetry (see ref. Angew. Chem. Int. Ed. 2013, 52, 1128-1133). Totally different crystal field cases.
- 4) Ref [68] "V.A. Antonov, P.A. Arsenev, K.E. Bienert, A.V. Potemkin, Physica Status Solidi, 19 (2010) 289-299". The correct reference should be "V.A. Antonov, P.A. Arsenev, K.E. Bienert, A.V. Potemkin, Physica Status Solidi, 19 (1973) 289-299". This work is about the splitting levels of rare earth ions in $YAlO_3$ crystals, where Y^{3+} sites have C_{1h} symmetry. However, the rare earth ions in crystal β - $NaYF_4$ have

totally different point group symmetry, which is responsible for different splitting energy levels. So there's relatively little relationship to the current work.

- 5) Ref [52], LaCl_3 has very similar symmetry to $\beta\text{-NaYF}_4$. From this paper, energy gap of ${}^1D_2 \rightarrow {}^3F_2$ transition could be calculated as 12629 cm^{-1} , equaling to 791.8 nm. This value still does not match with 810 nm.
- 6) Ref [53] is about optical fiber; here the overlap should be caused by spectral inhomogeneous broadening in noncrystalline glass.
- 7) Ref [54] has a clear conclusion that 'the ESA process (${}^3F_2 \rightarrow {}^1D_2$) is insignificant for the 456 nm emission of the $\text{NaYF}_4:1\% \text{ Tm}^{3+}$ UCNPs under 800 nm laser excitation.' How do this support the claim 'The matching of the 810 nm laser light with the transition of ${}^1D_2 \rightarrow {}^3F_2$ '?

To sum up, most of these references are inaccurately cited, and some even shows opposite conclusion to the author's claims (e.g. Phys. Chem. Chem. Phys. 18, 25905-25914, 2016).

No information about ${}^1D_2 \rightarrow {}^3F_2$... Opt. Express 16, 13781-13799 (2008)

- 8) Ref [70], again, it is a book chapter, 1-55 pages, so hard for us to find relevant information! It seems only the page 15 is closer "The condition specifying that the intermediate lifetime is substantially longer than that of the upper state is met in most cases, but not in all. This is because higher-excited states by definition have more radiative decay pathways and emit at shorter wavelengths, and both factors contribute to increasing their total radiative decay rate constants (see Eqs.2 and 4). Upper states are also generally more susceptible to nonradiative cross-relaxation deactivation through channels not available to lower-energy states. Nevertheless, upconversion processes involving short-lived intermediates with longer-lived upper states are occasionally encountered. Many of the correlations described above are only applicable when k_1 is significantly smaller than k_2 . As k_2 decreases and becomes comparable to or smaller than k_1 , the information content of such experiments declines significantly. The analogous curves calculated for short-pulse excitation using a ratio of $k_2 = k_1/40$ show the following: while the pure GSA/ETU and GSA/ESA curves are still distinguishable by a delayed maximum in the former, the case of 40% ESA:60% ETU is essentially indistinguishable from that of the pure ESA. In the limit of $k_2 \ll k_1$, the upconversion rise time is correlated with the lifetime of the intermediate level, and the decay time with that of the upper level. The situation is even less informative for the square-wave experiment, where a negligible deviation is discernible between all three curves, occurring at the shortest time only.", but ... unfortunately nothing here supports their argument!

In fact, in our work (Nature Nano. 2013, 8, 729-734) discovering the highly doped upconversion nanocrystals ($\beta\text{-NaYF}_4: \text{Yb, Tm}$), we have carefully measured the power-dependent spectra of UCNPs at different Tm concentrations. I am attaching below the 1% Tm power dependent spectra Figure 2b and its spectrum peaks Figure 2c. It is obvious that 740 nm and 780 nm emissions belong to the ${}^1D_2 \rightarrow {}^3F_{2,3}$ transitions, very consistent to refs [52] and [66]. Therefore it is impossible for 810 nm depletion laser to stimulate ${}^1D_2 \rightarrow {}^3F_{2,3}$ transitions.

It is almost clear that the 810 nm overlaps the $3H_4 \rightarrow 3H_6$ transition for the stimulated emission, consistent to our recent work, Nature 543, 229-233 (2017).

It is interesting that the authors observed the emission increases at 700 nm and 1470 nm bands at the presence of 810 nm laser in high Tm dopant concentration NPs. This is in fact consistent to our mechanism: the cross relaxation in highly doped UCNPs creates photon avalanche-like effect and quickly populates the intermediate states, e.g. $3F_{2,3}$ and $3H_4$ and lower levels. The stimulated emission at $3H_4$ short circuits the upconversion pathway but accelerate the photon accumulations at all the two photon levels and lower levels.

As we concerns (consistent to Referee #3) in our previous comments, if the stimulated emission does happen at $1D_2$ level, then the lifetime should be at least ns to ps levels. I noticed the authors use chopper to generate pulsed illumination and depletion, and too slow to measure ns level lifetime. In fact, we did a very careful measurement and time domain characterizations using electronically current pulses to generate rapid switch on and off of both 980 and 808 nm laser diodes, see our recent Nature work, the extended data Figure 5, below:

From this measurement, we see the 455 nm emission at the onset of 808 nm laser still has fairly long decay lifetime (more than 10 us), which rules out the stimulated depletion at 1D2 level.

To improve the quality of this manuscript, other suggestions below should be considered:

1. “interionic interaction” is not commonly used in lanthanides. I do suggest the title as “Achieving high-efficiency emission depletion nanoscopy by employing cross relaxation in highly-doped upconversion nanoparticles”.
2. In abstract, two over claimed achievements: “multicolor” should be “two color”
“... the first super-resolution imaging of immunostained cytoskeleton structures of cancer cells” should be “the first super-resolution imaging of cytoskeleton structures of fixed cells”. This is because the staining protocol is non-specific because of the use of second antibody conjugated UCNPs to stain the primary antibody stained cytoskeleton structure. The ideal labeling, convincing to cell biology methods, should be to use the primary antibody conjugated UCNPs to directly recognize the cytoskeleton of live cells. I understand this could be another level of challenge in the bioconjugation chemistry. The current demonstration is sufficient.
3. “the matter-matter interaction between lanthanide ions provides an auxiliary mechanism for the optical depletion pathway of upconversion luminescence (Fig. S1)” becomes vague. It is really about concentration-dependent cross relaxation between lanthanide ions, and suggest to remove Fig S1.
4. The authors should treat the data, figures and discussion texts reported in the supplementary information sections at the same level of scientific accuracy as the main text. For example, moving the UCNPs tracking data into SI Fig 16 does not really help. We did a calculation quite carefully here, the high speed scanning (20 us dwelling time) makes the signal extremely weak to be detected from a single UCNP. Therefore it is almost certain that the observation are the clusters of many UCNPs. Then the value is lost here. Since the super res image of cytoskeleton is impressive enough, no point to show this tracking data (a few previous papers have reported the tracking experiments, again the same problem not being able to show the singles).
5. The spectra in Figure R10 show clear emission peaks corresponding to the transitions from Er^{3+} . Is this caused by contamination during particle synthesis? How do you rule out the upconversion emission generation upon 810 or 795 nm excitation in assistance with the intermediate level of Er^{3+} ?
6. Figure R11 should provide the sample information, e.g., doping concentration.
7. Quite a lot of previous works have proved that the population of 1D2 level in Yb-Tm system was related to 4 photon process. I leave this discussion open at this stage.

The authors should carefully check the manuscript due to some inconsistent data:

Main text:

The data shown in Figure 1e is different with the related description in the main text. Page 11 “ $\text{NaYF}_4:18\%\text{Yb}^{3+},10\%\text{Tm}^{3+}$ ” should be “ $\text{NaGdF}_4:18\%\text{Yb}^{3+},10\%\text{Tm}^{3+}$ ”.

Supporting information

On the page 4, in the “Preparation of hydrophilic Tm^{3+} -UCNPs with diameters around 20 nm” part, there are two methods used for the hydrophilic nanoparticles preparation. Which one has the diameter around 20 nm? Maybe the $\text{NaYF}_4:18\%\text{Yb},0.5\%\text{Tm}$ one (19.5nm) is close to this? But it does not seem like this sample. Also, in the followed description, one method is used for the 20 nm sample, while another method is for the 11.8 nm sample. The authors need to check it carefully.

In the Figure S5 (b), it is better to use 11.8 instead of 11.75.

In the figure S15, where is the 18.5 ± 1.8 nm from? Whether it is the TEM size of the OA free nanoparticles? Why it is a little bigger than that with OA (18.0 ± 1.8 nm). Also, the 18.0 ± 1.8 nm in the figure S19 should be “ 11.8 ± 2.2 nm”.

In the figure S15 “The bands at 1552 cm^{-1} and 1460 cm^{-1} of OA-UCNPs were associated with the asymmetric (ν_{as}) and symmetric (ν_s) stretching vibration of -COOH groups”, while in the figure S19 “OA-UCNPs exhibit a broadband at about 3433 cm^{-1} and a weak band at 1738 cm^{-1} , associated with the -COOH stretching vibration”. Inconsistent.

It is interesting to see that the intensity of the nanoparticles without OA, with OA or PAA have nearly same spectrum. How the emission spectrum are tested in figure S15 and S19? Is this single nanoparticle emission? What’s the laser power used in the test?

Other incorrect reference should be double checked through the manuscript:

For example, one page 5 “similar to previous reports in high Tm^{3+} -doping UCNPs under excitation of comparable intensity^{34, 35}”. There is nothing about high Tm doping as well as comparable intensity in Ref 34 (1%Tm, 15 W/cm^2). Also Ref 5 and 24 are the same work.

Manuscript No. NCOMMS-17-01804A

Title: Achieving high-efficiency emission depletion nanoscopy by employing **cross relaxation** in upconversion nanoparticles

Response Letter to Reviewers

Dear Reviewers,

Many thanks for your recommendation and positive evaluation on our manuscript as well as your further careful consideration and valuable comments on our revised manuscript, which again have helped us improve the manuscript. In the following, we provide a point-by-point response to the comments, together with the corresponding changes in the manuscript. As below, the reviewers' comments are written in **black** and our responses to them in **blue**. The important amendments or changes to the manuscript are given after the response in **red**.

Reviewers' comments:

Reviewer #2 (Remarks to the Author):

In this revised version, the authors have significantly improved the quality of this work by the super resolution imaging of cellular cytoskeleton protein structures, however, the fundamental in describing the depletion mechanism was wrong. In order to help this work to become eventually publishable in the prestigious journal, Nature Communications, my team has spent some significant amount of time on carefully checking many details of this revision. Major revision is still required.

Three main credits of this work should eventually justify its publication into Nature Communications:

1. Demonstration of the second color from Tb doped core-shell upconversion nanoparticles using energy migration effect, this suggests the potential for multi-color super resolution using one pair of excitation (975 nm) and depletion (810 nm) beams;
2. Broad spectrum (deep UV and near infrared to infrared) measurement;
3. Super resolution imaging of cellular cytoskeleton protein structures with high speed (100 μ s dwelling time)

Response: We appreciate the reviewer for his/her positive evaluation and recommendation on our revised manuscript and his/her team's efforts to improve the quality of our manuscript.

1. The major issue of the stimulated emission depletion at 1D_2 level remains incorrect: In response to our early comments on the mechanism (and referees #3 and 4's comments): "*We*

agree that 810 nm light does not match with the bandgap of $^1D_2 \rightarrow ^3F_3$. However, the matching of the $^1D_2 \rightarrow ^3F_2$ transition with light near 800 nm is supported by many previous reports [51-54]. According to the theoretical calculations and experimental measurements under ultralow temperature (~ 10 K), the $^1D_2 \rightarrow ^3F_2$ transition is centered around 800 nm and could cover quite a few nanometers [65-68]. Furthermore, the FWHM of $^1D_2 \rightarrow ^3F_2$ transition spectrum would be broadened under ambient temperature (~ 300 K) and transient high temperature (under illumination of tightly focused light) [69]. In addition, in our experiments the depletion laser is centered at 810 nm and the FWHM is 4 nm, making the matching with the $^1D_2 \rightarrow ^3F_2$ transition accessible. It should be noted that the $^1D_2 \rightarrow ^3F_2$ transition is very easy to be overlooked and was indeed often overlooked in the literature probably due to the following reasons: (a) A common misconception exists that the 3F_2 and 3F_3 states can be treated as a single state (usually denoted as $^3F_{2,3}$) while ignoring their difference because of the fact that the ions at the 3F_2 state quickly decays to the 3F_3 state nonradiatively; (b) The transition $^1D_2 \rightarrow ^3F_2$ spectrally overlap with other transitions, including $^3H_4 \rightarrow ^3H_6$ and $^1G_4 \rightarrow ^3H_5$, [51-53] making it barely distinguishable from others and thus causing negligence.”

We have carefully checked their cited references, and found that the authors misinterpreted the details in the literatures. Most of these references are found irrelevant.

Response: We acknowledge the reviewer for comment on our response to the question regarding the wavelength matching of 810 nm laser with the transition of $^1D_2 \rightarrow ^3F_2$. In a previous comment: “...The 1D_2 level of Tm^{3+} is the excited state accumulating four 980 nm photons sensitized by Yb^{3+} , which level is responsible for the 455 nm, 480 nm, 660 nm and **740 nm** emissions. There is no transition bandgap for 810 nm absorption from this level”, the reviewer thought that the $^1D_2 \rightarrow ^3F_2$ transition can only match wavelengths as short as **740 nm**. By above response, we tried to provide supporting information for our claim that this transition matches significantly longer wavelengths and can further match an 810 nm laser with a few nanometer bandwidth. We cited a few publications including review and research articles, covering different host materials not limited to $NaYF_4$ with hexagonal structure of C_{3h} or C_s symmetry. **It is well known that the electrons of lanthanides in the 4f shell have localized states and exhibit weak coupling to ligand electrons and lattice vibrations due to shielding by 5s and 5p electrons and thus the energy levels of the $4f^N$ multiplets do not vary significantly, approximately within 1% for different matrices [1]. By providing these references, we would like to argue that the $^1D_2 \rightarrow ^3F_2$ transition could match 810 nm in various matrices.**

We are sorry for inadequate clarifications in our previous Response Letter. Here, we would like to explain point-to-point why these publications were cited. In addition, specific to Tm^{3+} in $NaYF_4$, we would like to provide more supporting materials for the matching of 810 nm laser

with the $^1D_2 \rightarrow ^3F_2$ transition.

1) Ref [65] “G.H. Dieke, R.A. Satten, American Journal of Physics, 38 (1970) 525-525”. We couldn’t find this paper. The correct one should be “G.H. Dieke, R.A. Satten, American Journal of Physics, 38 (1970) 399”. This is a book review, it is hard to get supporting information related to the Tm^{3+} transitions.

Response: We are very sorry for the wrong citation information about the book that we intended to cite, “Spectra and energy levels of rare earth ions in crystals”, Interscience Publishers, 1968, authored by Gerhard Heinrich Dieke [2]. Various chapters of this book discuss thoroughly spectra and levels of free lanthanide ions, the crystal field, crystal symmetry and the structure of the spectrum, intensities, selection rules, and comparison with empirical data. Particularly, energy levels for Tm^{3+} in different host materials (Y_2O_3 and YCl_3) were summarized, from which the energy gap between the 1D_2 and 3F_2 can be estimated.

2) According to ref [66], the energy of $Tm^{3+}:^1D_2$ level equals to 27522.9 cm^{-1} , 3F_2 14854.9 cm^{-1} in Y_2O_3 . From this data, we can calculate the energy gap of $^1D_2 \rightarrow ^3F_2$ transition to be 12668 cm^{-1} , equaling to 789.4 nm . It is 20 nm gap between 790 nm and 810 nm , not a few nanometers, impossible to be caused by temperature broadening.

Response: We acknowledge the reviewer for the comment. The energy level positions for the 1D_2 (27522.9 cm^{-1}) and 3F_2 (14854.9 cm^{-1}) states of Tm^{3+} that the reviewer specified here were taken from Table III in ref. [3] (ref. [66] in our previous Response Letter), where a comparison between experimental and calculated **energy centers of “gravity”** for $Y_2O_3: 5\% Tm^{3+}$ at 4.2 K was provided. However, when calculating the energy gap between two $4f^N$ multiplets, **energy splitting should be also considered**. In Table I in the same reference [3], the author reported observed energy levels of $5\% Tm^{3+}$ in Y_2O_3 at 4.2 K , where energy splittings were also provided for the 1D_2 and 3F_2 states, see below.

According to these data, the $^1D_2 \rightarrow ^3F_2$ transition can match a relative large wavelength range at 4.2 K from **777 nm (12870.8 cm^{-1} : 27933.1 cm^{-1} - 15062.3 cm^{-1}) to 811 nm (12369.4 cm^{-1} : 27691.0 cm^{-1} - 15361.6 cm^{-1}) (Table RR1)**. It should be noted that these calculations are based on the centers of sub energy levels with their linewidths neglected. Considering the sublevel linewidths, the long wavelength limit would go further beyond 811 nm . A hint on the significant sublevel linewidths can be obtained from Ref. [4], which reported that an sublevel splitting the linewidth could cover hundreds of wave numbers and thus significantly broaden the spectrum, e.g., from 786.7 nm ($28002 - 15290\text{ cm}^{-1}$) to 804 nm ($27825 - 15390\text{ cm}^{-1}$) for $^1D_2 \rightarrow ^3F_2$ according to Fig. RR1 (Fig. 2 in the Ref. [4]). In addition, considering temperature broadening effect this range should be further broader at room temperature [4, 5] (Fig. RR2). **Our depletion laser of 810 nm with a few nanometer bandwidth is well located in this range.**

Table RR1 Observed energy levels of 5%Tm³⁺ in Y₂O₃ at 4.2 K. (Taken from ref. [3], i.e., ref. [66] in our previous Response Letter). Table reproduced with permission from AIP Publishing.

Empirical label	Energy levels in vacuum (cm ⁻¹)	Method of observation ^a	Relative intensity ^c	^s L _J level	Empirical label	Energy levels in vacuum (cm ⁻¹)	Method of observation ^a	Relative intensity ^c	^s L _J level
Z ₁	0.00	Af		³ H ₆	B ₁ ^d	14 566.9	Af	9n	³ F ₃
Z ₂	30.7	Af			B ₂ ^d	14 571.6	Af	9n	
Z ₃	89.3	Af			B ₃	14 626.9	Af	9b	
Z ₄	219.0	Af			B ₄	14 657.7	Af	4vn	
Z ₅	230.3	Af			B ₅	14 662.1	Af	4vn	
Z ₆	340.0	f			B ₆	14 690.1	A	1n	
Z ₇ ^c	382.4	A			B ₇	14 725.8	A	6b	
Z ₈	435.7	f			C ₁	15 062.3	Af	2n	³ F ₂
Z ₉	488.4	f			C ₂	15 185.5	A	5b	
Z ₁₁ ^c	692.3	A			C ₃	15 240.6	A	6b	
Z ₁₂	788.5	f			C ₄	15 361.6	A	0vb	
Z ₁₃	796.9	f			D ₁	20 901.8	F		¹ G ₄
Y ₁	5 615.0	Ff	...	³ H ₄	D ₂	21 015.0	A	9n	
Y ₂	5 673.6	AfF	4vn		D ₃	21 060.5	A	1vn	
Y ₃	5 780.4	AfF	vn		D ₄	21 526.9	A	8b	
Y ₄	6 005.3	Af	5vn		D ₇	21 595.3	A	10b	
Y ₅	6 018.4	Af	2n		D ₈	21 618.3	A	3n	
Y ₇	6 114.1	Af	9n		D ₉	21 778.5	A	7b	
Y ₈	6 144.1	Af	3n		E ₁	27 691.0	Af	6vn	¹ D ₂
Y ₉	6 189.0	Af	4n		E ₂	27 726.5	Af	5vn	
X ₁	8 258.1	Af	2.5n	³ H ₆	E ₄	27 874.3	A	7vb	
X ₂	8 300.6	Af	5n		E ₅	27 933.1	A	3vb	
X ₃	8 330.6	Af	5n		F ₁	33 876.6	Af	6n	¹ I ₆
X ₄	8 464.8	A	7n		F ₂	33 884.3	Af	9n	
X ₅	8 475.1	A	10n		F ₆	34 368.0	A	1n	
X ₆	8 543.4	A	5b		F ₆	34 397.7	A	6n	
X ₇	8 569.0	A	4vb		F ₇	34 520.1	A	1vb	
X ₈	8 756.6	A	3vb		G ₁	35 267.7	A	1vb	³ P ₀
X ₁₀	8 916.6	A	3vb		H ₁	35 920.8	A	8vn	³ P ₁
A ₁	12 555.9	A	4vn	³ F ₄	H ₂	36 158.7	A	4n	
A ₂	12 634.5	Af	1vn		H ₃	36 231.3	A	3b	
A ₃	12 696.6	Af	8n		I ₁	37 556.9	A	9vn	³ P ₂
A ₄	12 812.9	Af	6n		I ₂	37 753.9	A	8vn	
A ₅	12 842.5	Af	0n		I ₃	37 847.2	A	7n	
A ₆	12 871.8	f			I ₅	38 194.3	A	4b	
A ₈	13 016.9	A	1n						
A ₉	13 047.7	A	0n						

^a Method of observation: A, absorption; f, terminal state for fluorescence; F, initial state of fluorescence.
^b Approximate intensity and linewidth. Intensities are relative, on a scale of 0-10. Width indications are vn, very narrow (less than 2 cm⁻¹); n, narrow (2-5 cm⁻¹); b, broad (5-10 cm⁻¹); vb, very broad (greater than 10 cm⁻¹).
^c Levels observed in the 195°K absorption spectrum of Tm₂O₃.
^d Energy positions for close doublet measured at 4.2°K in 1% Tm³⁺ in Y₂O₃.

Figure RR1 (Figs. 2d and 2f. in ref. [6]) Experimental (solid lines) and calculated (dashed lines) low-temperature absorption spectra of NYF:Tm³⁺ crystals. The calculated spectra are obtained by fitting the parameters of the Gaussian components so as to obtain the best approximation of the experimental spectrum by the sum of the components (dashed lines); temperature 10 K; thulium concentration 20%; transitions from the ³H₆ ground state to the (a) ¹D₂ and (b) ³F₃, ³F₂ and ³H₄ states. Figure reproduced with permission from Pleiades Publishing.

3) Ref [67] is about Cs₂NaLnCl₆ system that has the cubic elpasolite structure at room

temperature (space group Fm3m) with the Ln³⁺ ion situated at a site of six coordinate octahedral (O_h) symmetry. But in the current work, NaYF₄ crystals have hexagonal structure, where Ln³⁺ ion situated at a site of nine coordinate C_{3h} or C_s symmetry (see ref. Angew. Chem. Int. Ed. 2013, 52, 1128-1133). Totally different crystal field cases.

Response: In ref. [7] (ref. [67] in our previous Response Letter), the energy levels and wave functions of the f¹² open shell of Tm³⁺ in Cs₂NaTmCl₆ were calculated with a crystal field model defined in the intermediate coupling approximation. Their calculated results give that the transition between ¹D₂ (27700 cm⁻¹) and ³F₂ (15122 cm⁻¹) states matches a **center-of-gravity** wavelength of 795 nm. Their absorption spectra measurements **4.2-20 K and 77 K** of the transitions of ³H₆→³F₂ (Figure 3 in ref. [7]) and ³H₆→¹D₂ (Figure 5 in ref. [7]) reveal that the position of ³F₂ state is around 15390 cm⁻¹ and the sublevels of the ¹D₂ state can be as low as 27600 cm⁻¹, **indicating that the transition ¹D₂ → ³F₂ could match a wavelength of 819 nm (12210 cm⁻¹) in Cs₂NaTmCl₆ at 77 K**. Again, these calculations are based on the centers of sub energy levels with their linewidths neglected. Considering the linewidths, the red tail of the spectrum would go further beyond 819 nm. In addition, considering temperature broadening effect, this transition could match even longer wavelengths at room temperature [4, 5].

Our motivation for citing publications about Tm³⁺ in various matrices is given above.

4) Ref [68] “V.A. Antonov, P.A. Arsenev, K.E. Bienert, A.V. Potemkin, Physica Status Solidi, 19 (2010) 289-299”. The correct reference should be “V.A. Antonov, P.A. Arsenev, K.E. Bienert, A.V. Potemkin, Physica Status Solidi, 19 (1973) 289-299”. This work is about the splitting levels of rare earth ions in YAlO₃ crystals, where Y³⁺ sites have C_{1h} symmetry. However, the rare earth ions in crystal β-NaYF₄ have totally different point group symmetry, which is responsible for different splitting energy levels. So there’s relatively little relationship to the current work.

Response: We are sorry for the date information of this publication (ref. [68] in our previous Response Letter). In Table 4 of this reference [7], the energy levels of the ¹D₂ and ³F₂ states of Tm³⁺ in YAlO₃ at 77 K were reported, according to which the ¹D₂ → ³F₂ transition could match a wavelength of 807 nm. These calculations are based on the centers of sub energy levels with their linewidths neglected. Considering the linewidths, the long wavelength limit would go further beyond 807 nm. In addition, considering temperature broadening effect, this transition could match even longer wavelengths at room temperature [4, 5].

Our motivation for citing publications about Tm³⁺ in various matrices is given above.

5) Ref [52], LaCl₃ has very similar symmetry to β-NaYF₄. From this paper, energy gap of ¹D₂ → ³F₂ transition could be calculated as 12629 cm⁻¹, equaling to 791.8 nm. This value still does not match with 810 nm.

Response: We acknowledge the reviewer for the comment. Again, it should be noted the energy splitting and temperature broadening should be considered when calculating the energy gap for the transition. According to Table 2 in ref. [8] (ref. [52] in our previous Response Letter), the sub energy levels of 1D_2 state could be down to 27534 cm^{-1} and that of the 3F_2 state can be up to 15051 cm^{-1} at 77 K, matching a wavelength of 801 nm. It should be noted that these calculations are based on the centers of sub energy levels with their linewidths neglected. Considering the linewidths, the long wavelength limit would go further beyond 801 nm. In addition, considering temperature broadening effect, this transition could match even longer wavelengths at room temperature [4, 5]. Regarding the amplitude of the influence of temperature broadening, more discussion is given below where we discuss spectroscopic properties of Tm^{3+} in $NaYF_4$.

In addition, it should be also noted that all calculations about the energy gap between the 1D_2 and 3F_2 states using experimental results are based on **absorption spectra at low temperature**, which have already indicated that our laser at 810 nm ($\sim 4\text{ nm}$ FWHM) can match the $^1D_2 \rightarrow ^3F_2$ transition at room temperature. Considering the red shift of the **emission spectrum** ($^1D_2 \rightarrow ^3F_2$) relative to the **absorption spectrum** ($^3F_2 \rightarrow ^1D_2$), the matching can be consolidated. As an example, in Ref. [9], the emission and absorption spectra around 800 nm of thulium-doped fibers were reported, where the peak wavelength of the emission spectrum could be shifted to a longer wavelength by more than 20 nm.

Specific to Tm^{3+} in sodium yttrium fluoride, Ivanova et al. investigated the absorption spectra of $NYF:Tm^{3+}$ crystals at low (10 K) and room (300 K) temperatures [6]. **Indicated by the data reported in this article, the $^1D_2 \rightarrow ^3F_2$ transition could match a relatively broad range 758 nm - 860 nm at room temperature [6].** As presented in Figure RR1 (Figs. 2d and 2f. In ref. [6]), **at 10 K temperature**, the $^3F_2 \rightarrow ^1D_2$ transition (absorption spectra) could still match 12435 cm^{-1} ($27825\text{-}15390\text{ cm}^{-1}$), corresponding to **804 nm**, if considering the sub-level linewidths of both the 1D_2 and 3F_2 states.

A comparison between the absorption of an $NYF:Tm$ crystal at a temperature of 10 K and room temperature in Ref. [6] shows that the linewidths were easily broadened by tens of or even a hundred nanometers, depending on the wavelengths, as shown in Fig. RR2. When the temperature was increased to 300 K, significant spectral broadening prevailed, as shown in Fig. RR3. Considering the linewidths of both the 1D_2 and 3F_2 states, the absorption of the $^3F_2 \rightarrow ^1D_2$ transition was estimated to roughly match wavelengths in **a relatively broad range 758 nm - 860 nm**. As an example, if we roughly take the energy of $Tm^{3+}:^1D_2$ level at 361 nm (27700.8 cm^{-1}) (indicated by a red line in Fig. RR3), and the energy of $Tm^{3+}:^3F_2$ level at 650 nm (15384.6

cm^{-1}), this transition could match a wavelength of **812 nm**, close to the wavelength of our depletion laser (**810 nm, FWHM 4 nm**) [6].

Figure RR2 (Figs. 1b-1f in Ref. [6]) Absorption spectra of an NYF:Tm crystal at a temperature of **10 K** (solid lines) and **room temperature** (dashed lines). Figure reproduced with permission from Pleiades Publishing.

Figure RR3 (Figs. 4c and 4e in ref. [6]) **Absorption cross-section spectra** of a NYF:Tm³⁺ crystal at room temperature (300 K). The final levels of the transitions from the ³H₆ ground state to the (a) ¹D₂ and (b) ³F₃, ³F₂ and ³H₄ states. [**860 nm: (27276-15649) cm⁻¹**]. Figure reproduced with permission from Pleiades Publishing.

In addition, it should be also noted that the calculations above about the energy gap between 1D_2 and 3F_2 states using experimental results are based on absorption spectra. Considering the significant red shift of the emission spectrum relative to the absorption spectrum, as indicated by the data of other emission band ($^3H_6 - ^3F_4$) presented in Figs. 1 and 8 in Ref. [6, 9], the matching of the 810 nm laser with the $^1D_2 \rightarrow ^3F_2$ transition can be further consolidated.

Although in our work, NaYF₄ crystals have a hexagonal structure, the matching condition for the $^1D_2 \rightarrow ^3F_2$ transition with a laser of 810 nm (FWHM 4 nm) remains, since the energy levels of lanthanide dopants were found almost the same in cubic and hexagonal NaYF₄ structures [10, 11]. Wang et al. carried out photoluminescence studies of NaYF₄:Yb/Er (18/2 mol%) nanocrystals with varying dopant concentration of Gd³⁺, and found that with increasing dopant concentration of Gd³⁺, the structure of NaYF₄ was transformed from cubic to hexagonal, but the energy levels position and emission linewidths remained with non-noticeable change, as shown in Fig. RR4(a) [10]. Wang et al. reported UC luminescence spectra of NaYF₄:Yb/Tm (20/2 mol%) microcrystals with different phase structures (cubic and hexagonal) under 980-nm excitation, and found that no obvious linewidth change happened to the emission band and the shift of the Tm³⁺ emission peaks was very slight, as shown in Fig. RR4(b) [11].

Figure RR4 (A) (Figure S4 in Ref. [10]) Photoluminescence spectra of NaYF₄:Yb/Er (18/2 mol%) nanocrystals with varying dopant concentration of Gd³⁺. The emission spectra were normalized to Er³⁺ emission at 540 nm. With increasing Gd³⁺, pure hexagonal NaYF₄ were formed; (B) (Fig. 6 in Ref. [11]) UC luminescence spectra (**normalized to $^1D_2 \rightarrow ^3H_6$ transition**) of NaYF₄:Yb/Tm (20/2 mol%) microcrystals with different phase structures under 980-nm excitation: (a) cubic; (b) hexagonal. Figure reproduced with permission from Nature Publishing Group (ref. 10) and Elsevier (ref. 11).

6) Ref [53] is about optical fiber; here the overlap should be caused by spectral inhomogeneous broadening in noncrystalline glass.

Response: We hold the view that three different transitions, $^3H_4 \rightarrow ^3H_6$, $^1G_4 \rightarrow ^3H_5$, and $^1D_2 \rightarrow ^3F_2$, could overlap and contribute to the Tm³⁺ luminescence at around 800-810 nm, while traditionally this emission was often ascribed to the $^3H_4 \rightarrow ^3H_6$ transition. Ref. [12] (Ref. [53] in our previous Response Letter) was cited to show that the $^3H_4 \rightarrow ^3H_6$ and $^1G_4 \rightarrow ^3H_5$ transitions

could overlap with ${}^1D_2 \rightarrow {}^3F_2$, supporting our claim in our previous Response Letter, i.e., “...this transition (${}^1D_2 \rightarrow {}^3F_2$) was often overlooked in the literature and its contribution was assigned to other transitions, such as ${}^3H_4 \rightarrow {}^3H_6$ and ${}^1G_4 \rightarrow {}^3H_5$, due to the spectral overlap among these three transitions [51-53, 55]”. Although this research was carried out in optical fibers, it indicates that the ${}^3H_4 \rightarrow {}^3H_6$ and ${}^1G_4 \rightarrow {}^3H_5$ transitions could also overlap in other materials, as trivalent rare earth ions usually have relatively fixed energy level positions in different matrices, as explained above. The host material dependent spectral shift, if significant, should be orthokinetic for all the three transitions and not change the overlapping condition.

7) Ref [54] has a clear conclusion that ‘the ESA process (${}^3F_2 \rightarrow {}^1D_2$) is insignificant for the 456 nm emission of the NaYF₄:1% Tm³⁺ UCNPs under 800 nm laser excitation.’ How do this support the claim ‘The matching of the 810 nm laser light with the transition of ${}^1D_2 \rightarrow {}^3F_2$ ’?

Response: We acknowledge the reviewer for this comment. In Ref. [13] (Ref. [54] in our previous Response Letter), the authors explicitly claimed (paragraph 1, page 25911) “the enhanced 456 nm luminescence (Fig. 6a) of codoped UCNPs under simultaneous laser excitation of 800 nm and 980 nm revealed that ESA is a feasible pathway, and that the oscillator strength (Q) is strong,” supporting our claim about the possibility of the matching of 810 nm laser with the ${}^1D_2 \rightarrow {}^3F_2$ stimulated emission process (wavelength red-shifted compared to ESA ${}^3F_2 \rightarrow {}^1D_2$). As to the conclusion, “the ESA process is insignificant for the 456 nm emission of the NaYF₄:1% Tm³⁺ UCNPs **under 800 nm laser excitation**”, the authors meant the **contribution** of this ESA process to the 456 nm emission **under 800 nm laser excitation** is insignificant, but not arguing this ESA process is weak. The authors definitely ascribed the insignificant contribution of the ESA process to the small population of the ${}^3F_{2,3}$ state under single 800 nm excitation [13].

To sum up, most of these references are inaccurately cited, and some even shows opposite conclusion to the author’s claims (e.g. Phys. Chem. Chem. Phys. 18, 25905-25914, 2016).

Response: We have given point-to-point response, see above.

No information about ${}^1D_2 \rightarrow {}^3F_2$... Opt. Express 16, 13781-13799 (2008)

Response: Please refer to our response to the comment on the citation of Ref. [12] (Ref. [53] in our previous Response Letter).

Changes in the manuscript:

New reference added:

53. Ivanova SE, Tkachuk AM, Mirzaeva A, Pelle F. Spectroscopic study of thulium-activated double **sodium yttrium fluoride Na_{0.4}Y_{0.6}F(2.2):Tm³⁺** crystals: I. Intensity of spectra and luminescence kinetics. *Optics and Spectroscopy* **105**, 228-241 (2008).

Sentences added and modified in the revised manuscript (lines 123 to 133, pages 6-7):

“It should be noted that the $^1D_2 \rightarrow ^3F_2$ transition is very easy to be overlooked and was indeed often neglected in the literature probably due to the following reasons: (a) The 3F_2 and 3F_3 states are commonly treated as a single state (denoted as $^3F_{2,3}$) while ignoring their conspicuous energy difference since the ions at the 3F_2 state quickly decays to the 3F_3 state nonradiatively, and the transition of $^1D_2 \rightarrow ^3F_{2,3}$ was usually recognized to generate a single emission centered at 740-750 nm³³. Notably the $^1D_2 \rightarrow ^3F_2$ transition has an even bigger branching ratio than the $^1D_2 \rightarrow ^3F_3$ transition according to previous theoretical studies^{44, 45, 46, 47}; (b) The $^1D_2 \rightarrow ^3F_2$ transition spectrally overlaps with other transitions, including $^3H_4 \rightarrow ^3H_6$ and $^1G_4 \rightarrow ^3H_5$ ^{42, 48, 49}, making it barely distinguishable from others and thus causing negligence. The matching of the STED laser 810 nm light with the emission spectrum (red-shifting relative to the absorption spectrum) of the $^1D_2 \rightarrow ^3F_2$ transition is supported by many previous reports that indicate the absorption spectrum of $^3F_2 \rightarrow ^1D_2$ transition is centered around 800 nm and could cover quite a few nanometers^{42, 50, 51, 52, 53}.”

8) Ref [70], again, it is a book chapter, 1-55 pages, so hard for us to find relevant information! It seems only the page 15 is closer “The condition specifying that the intermediate lifetime is substantially longer than that of the upper state is met in most cases, but not in all. This is because higher-excited states by definition have more radiative decay pathways and emit at shorter wavelengths, and both factors contribute to increasing their total radiative decay rate constants (see Eqs. 2 and 4). Upper states are also generally more susceptible to nonradiative cross-relaxation deactivation through channels not available to lower-energy states. Nevertheless, upconversion processes involving short-lived intermediates with longer-lived upper states are occasionally encountered. Many of the correlations described above are only applicable when k_1 is significantly smaller than k_2 . As k_2 decreases and becomes comparable to or smaller than k_1 , the information content of such experiments declines significantly. The analogous curves calculated for short-pulse excitation using a ratio of $k_2 = k_1/40$ show the following: while the pure GSA/ETU and GSA/ESA curves are still distinguishable by a delayed maximum in the former, the case of 40% ESA: 60% ETU is essentially indistinguishable from that of the pure ESA. In the limit of $k_2 \ll k_1$, the upconversion rise time is correlated with the lifetime of the intermediate level, and the decay time with that of the upper level. The situation is even less informative for the square-wave experiment, where a negligible deviation is discernible between all three curves, occurring at the shortest time only”, but ... unfortunately nothing here supports their argument!

Response: In our previous Response Letter, Ref. [14] (Ref. [70] in our previous Response Letter) was cited to support our argument: “Due to the stepwise pumping nature, the obtained decay curves usually are not merely determined by the lifetime of the emitting state of interest and

often reflect the lifetimes of the intermediate states instead [70], which are significantly longer than that of the 1D_2 state...”. By this argument, we meant that an indirect pumping method via ETU process in lifetime measurement (as most researchers do, we used in the upconversion field to measure lifetimes of different energy levels) could bring systematic error, specifically the extracted lifetime of the emitting state is influenced by the lifetimes of intermediate states. Our argument is firmly based on the discovery in Ref. [14] as discussed below.

In paragraph 2, page 9 in Ref. [14], the authors pointed out that “In GSA/ETU, upconversion may proceed as long as the intermediate $^4I_{11/2}$ population is nonzero. The upconversion luminescence signal can thus last longer than predicted by the natural decay rate of the emitting state, and in this case is observed to decay at a rate approximately twice that of the intermediate state,...”. **This actually means that the measured lifetime of the emitting state by indirect excitation could be a half of that of the intermediate state in specific conditions ($k_2 \gg k_1$),** where $k_{1(2)}$ denotes the decay rate constant of the intermediate (upper) state. When the magnitude relationship between k_2 and k_1 changes, the conclusion will be different. As in the arguments that the reviewer cited above, “In the limit of $k_2 \ll k_1$, the upconversion rise time is correlated with the lifetime of the intermediate level, and **the decay time with that of the upper level**”. Obviously, in other regimes, the measured lifetime of the upper level will be influenced by the lifetime of the intermediate level to different extent when employing the indirect pumping approach. Although these conclusions are obtained based on a simplified three-level system describing the upconversion process of Er^{3+} singly doped systems, they can be translated to other complicated upconversion systems involving more types of ions and energy states. For instance, in the present study, the measured lifetime of the 1D_2 state, populated via a five-photon process, would be affected by the lifetimes of many long-lived intermediate states in a complicated manner.

Changes in the manuscript:

Sentences added/modified in the revised manuscript (lines 167-172, page 8):

“The less dramatic lifetime change in some samples than expected was probably ascribed to the multiple effects of the depletion laser (discussed below), as well as the stepwise upconversion pumping approach in the lifetime measurement, where the measured decay lifetimes are often not solely determined by the lifetime of the emitting state but also reflect the radiative properties of many long-lived intermediate states⁵⁷.”

2. In fact, in our work (Nature Nano. 2013, 8, 729-734) discovering the highly doped upconversion nanocrystals (β -NaYF₄:Yb,Tm), we have carefully measured the power-dependent spectra of UCNPs at different Tm concentrations. I am attaching below the 1% Tm power dependent spectra Figure 2b and its spectrum peaks Figure 2c. It is obvious that 740 nm

and 780 nm emissions belong to the $^1D_2 \rightarrow ^3F_{2,3}$ transitions, very consistent to refs [52] and [66]. Therefore it is impossible for 810 nm depletion laser to stimulate $^1D_2 \rightarrow ^3F_{2,3}$ transitions.

It is almost clear that the 810 nm overlaps the $^3H_4 \rightarrow ^3H_6$ transition for the stimulated emission, consistent to our recent work, Nature 543, 229-233 (2017).

Response: We acknowledge the reviewer for his/her inspiring comments on the assignment of the 740 nm, 780 nm and 810 nm emissions. We agree that the 740 nm emission can be ascribed to the transition from the 1D_2 state, specifically $^1D_2 \rightarrow ^3F_3$, and that the $^3H_4 \rightarrow ^3H_6$ transition makes the most contribution to the 810 nm emission. **However, we hold the view that the 780 nm emission originates from the transition of $^1G_4 \rightarrow ^3H_5$ rather than $^1D_2 \rightarrow ^3F_2$** , which is well supported by Fig. 1c in Ref. [15] (Nature Nano. 2013, 8, 729-734) authored by the reviewer, as shown below.

Figure RR5 (Fig. 1c in Ref. [15]) Upconversion spectra of a series of NaYF₄:Yb/Tm nanocrystals with varied Tm³⁺ concentrations under an excitation irradiance of 2.5 × 10⁶ W/cm², showing a steady increase in upconversion luminescence with increasing Tm³⁺ content from 0.2 mol% to 8 mol%.

In Fig. RR5, the comparison between the curves for 1 mol% Tm, 2 mol% Tm and 4 mol% Tm samples clearly shows that the emissions originating from the ¹D₂ state (455 nm from ¹D₂ → ³F₄, 515 nm from ¹D₂ → ³H₅, and 740 nm from ¹D₂ → ³F₃) **increases in step** with increasing the Tm³⁺ concentration, while the 780 nm emission increases significantly less than other bands. In 1 mol% Tm sample, the intensity at 780 nm is stronger than that at 740 nm, while in 2 mol% Tm and 4 mol% Tm samples, the 780 nm intensity is significantly weaker than that at 740 nm. According to the theoretical calculations in previous reports on Tm³⁺ in various matrices including hexagonal NaGdF₄, the ¹D₂ → ³F₂ transition has an even bigger branching ratio than the ¹D₂ → ³F₃ transition [16-19]. In combination, these studies exclude the possibility that the 780 nm originates from the ¹D₂ → ³F₂ transition, otherwise, the 780 nm intensity should be always stronger than that of 740 nm. Thus, the 780 nm emission should originate from a lower-order multi-photon upconversion process, which we believe to be the ¹G₄ → ³H₅ transition. In combination with our discussion about the energy gap between the ¹D₂ and ³F₂ states, we insist that the ¹D₂ → ³F₂ and ³H₄ → ³H₆ transitions overlap spectrally in a longer wavelength range.

In addition, it should be noted that the 740 nm emission (¹D₂ → ³F₃) in Fig. RR5 exhibits a relatively **broad bandwidth of 40-50 nm**, which can be ascribed to various broadening effects. It is reasonable to speculate that the ¹D₂ → ³F₂ transition could also cover a broad spectral range, making its matching with an 810 nm laser (FWHM 4 nm) easily achievable.

3. It is interesting that the authors observed the emission increases at 700 nm and 1470 nm bands at the presence of 810 nm laser in high Tm dopant concentration NPs. This is in fact consistent to our mechanism: the cross relaxation in highly doped UCNPs creates photon avalanche-like effect and quickly populates the intermediate states, *e.g.* ³F_{2,3} and ³H₄ and lower levels. The stimulated emission at ³H₄ short circuits the upconversion pathway but accelerate the photon accumulations at all the two photon levels and lower levels.

Response: We leave the discussion on the theoretical explanation of 700 nm intensity change open at this stage. However, the emission increase at 1470 nm band (³H₄ → ³F₄) observed in our nanosystem cannot be ascribed to the mechanism discovered in the Nature article authored by the reviewer [20]. According to the modelling and simulated results in Ref. [20] shown below, Fig. RR6 (Extended Data Figure 7 in Ref. [20]), the population of the ³H₄ state (n₃) significantly decreases after the addition of the 808 nm depletion laser, indicating emission decrease of the 1470 nm band (³H₄ → ³F₄), which is inconsistent with our observation. This discrepancy implies

that the mechanism reported in Ref. [20] is not active in our upconversion nanosystem.

Figure RR6 (Extended Data Figure 7b in Ref. [20]) Emitter populations as a function of time. The 980nm pumping is turned on at time = 0 s, while the 808 nm probing is turned on at time = 0.5 ms.

Figure reproduced with permission from Nature Publishing Group.

In samples with moderate Tm^{3+} concentration (e.g., $\text{NaYF}_4:18\% \text{Yb}^{3+}, 5\% \text{Tm}^{3+}$), we observed emission enhancement for the 475 nm band (${}^1\text{G}_4 \rightarrow {}^3\text{H}_6$) while the 455 nm emission was significantly depleted (Figure RR7). If the mechanism proposed in Ref. [20] is active in our situation, the 475 nm band (n_4) should be depleted more significantly than those originating from the ${}^3\text{H}_4$ state (n_3), as indicated by the simulated results in Fig. RR6 (Extended Data Figure 7b in Ref. [20]).

Figure RR7 The emission spectra of $\text{NaYF}_4:18\% \text{Yb}^{3+}, 5\% \text{Tm}^{3+}$ nanoparticles excited by 975-nm CW solely (700 kW cm^{-2}), 810-nm CW solely (17.7 MW cm^{-2}), 975-nm and 810-nm CW simultaneously. When carefully examining the mechanism reported in Ref. [20] for our situations, we carried out more 455 nm emission depletion studies by fixing the intensity of the 810 nm depletion laser to 14.2 MW/cm^2 while adjusting the intensity of the 975 nm excitation laser in a large

range from 175 kW cm⁻² to 17.5 MW cm⁻². It was found that the depletion efficiency of the 455 nm band decreases (increases) when increasing (decreasing) the intensity of the 975 nm laser, as shown in Fig. RR8. These results are in consistence with our mechanism and cases of conventional CW-STED, i.e., when increasing the excitation laser intensity the emission generally increases and it becomes harder to deplete it [21]. According to Ref. [21], the probability of spontaneous decay η_{CW} for conventional CW-STED writes as

$$\eta_{CW} = \frac{k_{ex} + k_s}{k_{ex} + (1 + \gamma)k_s},$$

with k_{ex} the excitation rate and k_s the spontaneous decay rate, and γ the effective saturation factor that accounts for the generally undesired excitation by the STED beam itself, indicating that a larger k_{ex} would cause a smaller depletion efficiency. However, the mechanism reported in Ref. [20] based on the population inversion (³H₄ relative to **the ground state** ³H₆) caused by the 975 nm excitation laser would predict a completely opposite trend, since larger (smaller) excitation intensity causes a more (less) prominent population inversion ($\Delta n = n(^3H_4) - n(^3H_6)$) between these two states, making the 455 nm emission depletion more (less) significant using identical depletion laser intensity (note: the net effect of stimulated emission is determined by the term of Power₈₀₈ * (n(³H₄) - n(³H₆)) according to Eq. (19) in Ref. [20], and n(³H₄) would equal n(³H₆) after the addition of the depletion laser (Fig. RR6)).

Figure RR8 Dependence of 455-nm depletion efficiency $((I_{455}^{975} - I_{455}^{975\&810})/I_{455}^{975})$ on the 975 nm excitation laser intensity (175 kW cm⁻² to 17.5 MW cm⁻²) with the depletion 810-nm laser intensity kept at 14.2 MW cm⁻². The measured sample was NaYF₄:18% Yb³⁺, 10% Tm³⁺.

However, inspired by the mechanism discovered by the reviewer in their nanosystem [20] and our recognition by far on the spectral overlapping between the transitions of ¹D₂ → ³F₂ and ³H₄ → ³H₆, we have realized that the stimulated emission process ³H₄ $\xrightarrow{810\text{ nm}}$ ³H₆, weakening the

synergistic enhancement effect of the 810 nm laser, should be also taken into account in the simulation, since the GSA and stimulated emission between ${}^3\text{H}_6$ and ${}^3\text{H}_4$ coexist. The net effect of the 810 nm laser on the population density of the ${}^3\text{H}_4$ state is determined by the magnitude relationship between $\sigma_{\text{se}}^{810} N({}^3\text{H}_4)$ and $\sigma_{\text{abs}}^{810} N({}^3\text{H}_6)$, where $\sigma_{\text{abs(se)}}^{810}$ denotes the absorption (stimulated emission) cross section of the transition of ${}^3\text{H}_6 \xrightarrow{810 \text{ nm}} {}^3\text{H}_4$ (${}^3\text{H}_4 \xrightarrow{810 \text{ nm}} {}^3\text{H}_6$), and $N({}^3\text{H}_4)$ and $N({}^3\text{H}_6)$ denote the population densities of the ${}^3\text{H}_4$ and ${}^3\text{H}_6$ states, respectively. **Our spectroscopic results of the 1470 nm band and emission depletion efficiency dependence on the 975 nm power points on that the ground state absorption process ${}^3\text{H}_6 \xrightarrow{810 \text{ nm}} {}^3\text{H}_4$ is dominant over the stimulated emission process (${}^3\text{H}_4 \xrightarrow{810 \text{ nm}} {}^3\text{H}_6$).**

Based on our responses to the previous comments, we believe that we cannot find any contradiction within our theory. Due to different situations in our study from that of reviewer's study, e.g., different lasing wavelengths (810 nm vs 808 nm) possibly with different FWHMs, nanoparticle crystal structure, composition and quality, it is reasonable that the population inversion (${}^3\text{H}_4$ relative to ${}^3\text{H}_6$) based emission depletion mechanism of the 455 nm band disclosed in Ref. [20] was not active in our study.

4. As we concern (consistent to Referee #3) in our previous comments, if the stimulated emission does happen at ${}^1\text{D}_2$ level, then the lifetime should be at least ns to ps levels. I noticed the authors use chopper to generate pulsed illumination and depletion, and too slow to measure ns level lifetime. In fact, we did a very careful measurement and time domain characterizations using electronically current pulses to generate rapid switch on and off of both 980 and 808 nm laser diodes, see our recent Nature work, the extended data Figure 5, below:

From this measurement, we see the 455 nm emission at the onset of 808 nm laser still has fairly long decay lifetime (more than 10 μs), which rules out the stimulated depletion at ${}^1\text{D}_2$ level.

Response: We acknowledge the reviewer for the comment on the lifetime change caused by stimulated emission process. We agree that usually in fluorescent dyes the lifetime of an excited state would be located in ns to ps levels when a stimulated emission process happens. This is not a steadfast rule, instead, it is mainly due to the intrinsic lifetime of the dyes, which is typically in ns range. In principle, the lifetime of stimulated emission is determined by the product of the SE cross section (σ_{STED}) and the photon flux (ρ_{STED}) of the depletion laser by $\tau_{\text{STED}} = 1/k_{\text{STED}} = 1/(\sigma_{\text{STED}} \times \rho_{\text{STED}})$. According to Ref. [22], STED reduces the lifetime of the excited state from $\tau_f = 1/k_{\text{fl}}$ to $\tau = 1/(k_{\text{fl}} + \sigma_{\text{STED}} \times \rho_{\text{STED}}) = 1/(k_{\text{fl}} + \sigma_{\text{STED}} \times I_{\text{STED}} \times \lambda_{\text{STED}}/(hc))$, where h is the Planck's constant, and c is the speed of light.

Taking $I_{\text{STED}} = 15.7 \text{ MW/cm}^2$ and $\sigma_{\text{STED}} = 10^{-21} - 10^{-20} \text{ cm}^2$ [9, 17, 23, 24], **it yields $\tau_{\text{STED}} \approx 1.5\text{--}6.7 \mu\text{s}$, which is quite a few orders of magnitude longer than stimulated emission lifetime in organic dyes (ns to ps)**. Thus, a measured lifetime in the order of $10 \mu\text{s}$ cannot exclude the occurrence of a stimulated emission process to an excited state of lanthanide ions.

Changes in the manuscript:

Sentences added in the revised manuscript (lines 163-172, page 8):

“Given that the emission cross sections of Tm^{3+} transition are typically in the range of $10^{-21}\text{--}10^{-20} \text{ cm}^2$ ^{45, 54, 55, 56}, current depletion laser intensity yields a stimulated emission lifetime from the $^1\text{D}_2$ state, τ_{STED} , in the order of a few μs (Supplementary Fig. 5), which are three orders of magnitude longer than stimulated emission lifetime in fluorescent organic dyes (ns to ps). The less dramatic lifetime change in some samples than expected was probably ascribed to the multiple effects of the depletion laser (discussed below), as well as the stepwise upconversion pumping approach in the lifetime measurement, where the measured decay lifetimes are often not solely determined by the lifetime of the emitting state but also reflect the radiative properties of many long-lived intermediate states ⁵⁷.”

Sentences added in the revised SI (lines 358-366, page 19):

“In principle, the lifetime of stimulated emission is determined by the product of the stimulated emission cross section (σ_{STED}) and the photon flux (ρ_{STED}) of the depletion laser by $\tau_{\text{STED}} = 1/k_{\text{STED}} = 1/(\sigma_{\text{STED}} \times \rho_{\text{STED}})$. According to Ref. [14], STED reduces the lifetime of the excited state from $\tau_f = 1/k_{\text{fl}}$ to $\tau = 1/(k_{\text{fl}} + \sigma_{\text{STED}} \times \rho_{\text{STED}}) = 1/(k_{\text{fl}} + \sigma_{\text{STED}} \times I_{\text{STED}} \times \lambda_{\text{STED}}/(hc))$, where h is the Planck's constant, and c is the speed of light. Taking $I_{\text{STED}} = 15.7 \text{ MW cm}^{-2}$ and $\sigma_{\text{STED}} = 10^{-21}\text{--}10^{-20} \text{ cm}^2$ [15-18], it yields $\tau_{\text{STED}} \approx 1.5\text{--}6.7 \mu\text{s}$, which is three orders of magnitude longer than the stimulated emission lifetime in organic dyes (ns to ps).”

5. To improve the quality of this manuscript, other suggestions below should be considered:

(1) “interionic interaction” is not commonly used in lanthanides. I do suggest the title as “Achieving high-efficiency emission depletion nanoscopy by employing cross relaxation in highly-doped upconversion nanoparticles”.

Response: We appreciate the reviewer’s advice. We have changed the title of our manuscript accordingly.

Change in the manuscript:

Title modified: “Achieving high-efficiency emission depletion nanoscopy by employing cross relaxation in upconversion nanoparticles”.

(2) In abstract, two over claimed achievements: “multicolor” should be “two color”. “... the first super-resolution imaging of immunostained cytoskeleton structures of cancer cells” should be “the first super-resolution imaging of cytoskeleton structures of fixed cells”. This is because the staining protocol is non-specific because of the use of second antibody conjugated UCNPs to stain the primary antibody stained cytoskeleton structure. The ideal labeling, convincing to cell biology methods, should be to use the primary antibody conjugated UCNPs to directly recognize the cytoskeleton of live cells. I understand this could be another level of challenge in the bioconjugation chemistry. The current demonstration is sufficient.

Response: We acknowledge the reviewer for his/her comments and encouraging attitude. “multicolor” has been changed to “two color” in the revised manuscript. The “the first super-resolution imaging of immunostained cytoskeleton structures of cancer cells” has been changed to “the first super-resolution imaging of immunostained cytoskeleton structures of fixed cells”. Our staining strategy and experimental protocol were very similar or the same with other commonly used staining protocols for immunofluorescence imaging [25-27], and the successful labelling of cytoskeleton structures of cancer cells should be ascribed to the function of immunological reaction. Immunostaining is an appropriate characterization of our result.

Changes in the manuscript:

Sentence modified (line 19, page 2): “We demonstrate two-color super-resolution imaging using upconversion nanoparticles...”.

Sentence modified (line 320, page 14): “...we have implemented two-color super-resolution imaging using a single excitation/depletion laser beam pair”.

Sentence modified (line 22, page 2): “We show the first super-resolution imaging of immunostained cytoskeleton structures of fixed cells”.

(3) “the matter-matter interaction between lanthanide ions provides an auxiliary mechanism for the optical depletion pathway of upconversion luminescence (Supplementary Fig. 1)” becomes vague. It is really about concentration-dependent cross relaxation between lanthanide ions, and suggest to remove Supplementary Fig. 1.

Response: We acknowledge the reviewer for the comment. As suggested, “cross relaxation” has been used to replace “matter-matter interaction” to make the description less ambiguous. However, we think the highly efficient CR1 process at high Tm^{3+} doping does provide an auxiliary mechanism to suppress the synergistic excitation effect and simultaneously enhances the SE depletion effect of the 810 nm beam by transferring the electrons at $^3\text{H}_4$ state to the $^3\text{F}_4$ state and thus significantly lowers the depletion laser intensity requirements. We have added more discussion to elucidate this point in the revised manuscript.

Figure S1 in the previous SI have been removed as suggested.

Change in the manuscript:

Figure S1 in the previous Supplementary Information has been removed.

Sentences added/modified in the revised manuscript (lines 210-225, page 10):

“Furthermore, it should be noted that the CR1 process can simultaneously enhance the STED effect of the 810 nm beam and thus lowers the depletion laser intensity requirements. As shown in Fig. 2b, in the case of low Tm^{3+} doping (without the CR1 process), the relaxing electrons from $^1\text{D}_2$ to $^3\text{F}_2$ via stimulated emission and then to $^3\text{H}_4$ would be readily re-pumped back to the $^1\text{D}_2$ state via the synergistic excitation of the two lasers. On the contrary, in the case of high Tm^{3+} doping the electrons at the $^1\text{D}_2$ state are first transferred to the $^3\text{H}_4$ state (via $^3\text{F}_2$) by the depletion laser, and then the CR1 process consecutively transfers the relaxing electrons from the $^3\text{H}_4$ state further to the $^3\text{F}_4$ state, where the CR1 process in fact introduce a second energy dissipating pathway $^3\text{F}_4 \rightarrow ^3\text{H}_6$. The cooperation of the STED effect of the 810 laser beam and the CR1 process makes the population center of “gravity” shift towards lower energy states and helps suppress the re-population rate of the emitting state under two laser irradiation, thus enhancing the stimulated emission induced depletion efficacy of the depletion laser. This downward shift of the population center of “gravity”, yielding a new equilibrium among the population densities of different energy states, can also explain the passive depletion of the $^1\text{G}_4$ state (475 nm emission) just below the $^1\text{D}_2$ state at high Tm^{3+} doping (Fig. 1b).”

(4) The authors should treat the data, figures and discussion texts reported in the supplementary information sections at the same level of scientific accuracy as the main text. For example, moving the UCNPs tracking data into SI Fig. 16 does not really help. We did a calculation quite carefully here, the high speed scanning (20 μs dwelling time) makes the signal extremely weak to be detected from a single UCNP. Therefore it is almost certain that the observation are the clusters of many UCNPs. Then the value is lost here. Since the super res image of cytoskeleton is impressive enough, no point to show this tracking data (a few previous papers have reported the tracking experiments, again the same problem not being able to show the singles).

Response: We appreciate the reviewer’s positive evaluation on our super-resolution imaging of

cytoskeleton. As suggested, we have removed the UCNP tracking data (Supplementary Fig. 16 in the previous SI) as well as the related nanoparticle characterization data (Supplementary Fig. 15 in the previous SI) of the HCl treated nanoparticles (prepared mainly for the nanoparticle tracking experiments) in the revised Supplementary Information.

Change in the manuscript:

The original Fig. S15 and Fig. S16 in the previous supplementary information have been removed.

Sentence modified in the manuscript (lines 254-256, page 12): “Such an excellent photostability is highly desired for STED nanoscopic imaging and the utilization of UCNPs would enable long time, continuous super-resolution imaging⁴⁴”.

(5) The spectra in Figure R10 show clear emission peaks corresponding to the transitions from Er^{3+} . Is this caused by contamination during particle synthesis? How do you rule out the upconversion emission generation upon 810 or 795 nm excitation in assistance with the intermediate level of Er^{3+} ?

Response: We thank the reviewer very much for his/her comments on this issue. In the original Figure R10, the spectra in the range of 520-560 nm seems to be from Er^{3+} . We carefully checked the experimental procedure of this part, and found that Er^{3+} doped UCNPs likely contaminated the Tm^{3+} -UCNP glass slide sample prepared in this measurement. We have redone the experiment very carefully to avoid sample contamination, and the result is given in Fig. RR9, where no emission from Er^{3+} was observed.

Figure RR9 Upconversion emission spectra of 10% Tm-UCNPs under excitation of 795-nm laser and 810-nm laser with the same power density, respectively.

(6) Figure R11 should provide the sample information, e.g., doping concentration.

Response: We apologize for not providing the composition information in our previous Response Letter. The sample used for the power dependence measurement was NaYF_4 :

18% Yb³⁺, 10% Tm³⁺.

(7) Quite a lot of previous works have proved that the population of ¹D₂ level in Yb-Tm system was related to 4 photon process. I leave this discussion open at this stage.

Response: We appreciate very much the reviewer's open attitude on this debate question.

6. The authors should carefully check the manuscript due to some inconsistent data:

Main text:

The data shown in Figure 1e is different with the related description in the main text.

Page 11 "NaYF₄:18% Yb³⁺, 10% Tm³⁺" should be "NaGdF₄:18% Yb³⁺, 10% Tm³⁺".

Response: We apologize for the errors in our previous manuscript, which have caused inconsistency and confusion. We have corrected the figure legend in Fig. 1e.

"NaYF₄:18% Yb³⁺, 10% Tm³⁺" has also been changed to "NaGdF₄:18% Yb³⁺, 10% Tm³⁺" in the description of 11.8 nm nanoparticles.

Change in the manuscript:

Figure legend changed in Fig. 1e.

Sentence modified in the revised manuscript (lines 291-293, page 13): "High Tm³⁺-doping UCNPs (NaGdF₄:18% Yb³⁺, 10% Tm³⁺) with an average diameter of 11.8(±2.2) nm were synthesized (Supplementary Fig. 4).".

Supporting information the page 4, in the "Preparation of hydrophilic Tm³⁺-UCNPs with diameters around 20 nm" part, there are two methods used for the hydrophilic nanoparticles preparation. Which one has the diameter around 20 nm? Maybe the NaYF₄: 18% Yb, 0.5% Tm one (19.5 nm) is close to this? But it does not seem like this sample.

Also, in the followed description, one method is used for the 20 nm sample, while another method is for the 11.8 nm sample. The authors need to check it carefully.

In the Figure S5 (b), it is better to use 11.8 instead of 11.75.

In the figure S15, where is the 18.5±1.8 nm from? Whether it is the TEM size of the OA free nanoparticles? Why it is a little bigger than that with OA (18.0±1.8 nm). Also, the 18.0±1.8 nm in the figure S19 should be "11.8±2.2 nm".

Response: We apologize for the typos we made in the previous SI which has caused confusion.

We thank the reviewer very much for his/her careful reading and good suggestions.

Actually, the phrase "with diameters around 20 nm" should appear after the heading of another subsection, i.e., "Synthesis of hydrophobic Tm³⁺-NaYF₄ UCNPs (with diameters around 20 nm)" on page 2 in the SI, because the synthesis of NaYF₄ UCNPs with different Tm³⁺ doping concentrations was implemented under the same condition and the resulting diameters of these nanoparticles were close, around 20 nm. This issue has been fixed now.

18.5±1.8 nm was another typo and it should be 18.0±1.8 nm, which is consistent with the TEM image in the Figure 1a of the main text.

Again, the 18.0±1.8 nm in the Figure S19 was also a typo and should be “11.8±2.2 nm” (NaGdF₄).

Changes in the manuscript:

Sub-section heading modified in the revised SI (line 43, page 2): “Synthesis of hydrophobic Tm³⁺-NaYF₄ UCNPs with diameters around 20 nm”.

Sub-section heading modified in the revised SI (line 94, page 4): “Preparation of hydrophilic Tm³⁺-UCNPs ~~with diameters around 20 nm~~”.

Sentence modified in the revised SI (lines 480-481, page 31): “..., which is larger than that observed from TEM measurement (11.8±2.2 nm).”

As suggested by the reviewer, the 11.75 nm has been changed to 11.8 nm in Supplementary Fig. 4 (b) in the revised SI. (line 343, page 17)

In the figure S15 “The bands at 1552 cm⁻¹ and 1460 cm⁻¹ of OA-UCNPs were associated with the asymmetric (ν_{as}) and symmetric (ν_s) stretching vibration of -COOH groups”, while in the figure S19 “OA-UCNPs exhibit a broadband at about 3433 cm⁻¹ and a weak band at 1738 cm⁻¹, associated with the -COOH stretching vibration”. Inconsistent.

Response: We apologize for the inaccurate statement in the previous SI. We have modified the statements to make them clearer. In fact, the bands at 1552 cm⁻¹ and 1460 cm⁻¹ of OA-UCNPs were associated with the asymmetric (ν_{as}) and symmetric (ν_s) stretching vibration of -COO⁻ group, which is consistent with the analysis in the following the references in the original SI.

[5] Zhang, T., et al., *A General Approach for Transferring Hydrophobic Nanocrystals into Water*. Nano Letters, 2007. 7(10): p. 3203;

[21] Liu, C., et al., Monodisperse, size-tunable and highly efficient β -NaYF₄: Yb, Er (Tm) up-conversion luminescent nanospheres: controllable synthesis and their surface modifications. Journal of Materials Chemistry, 2009. 19(21): p. 3546-3553.

In the original Figure S19 in our previous manuscript, the broadband at about 3433 cm⁻¹ and the weak band at 1738 cm⁻¹ correspond to the stretching vibration of O-H and C=O, respectively, also consistent with the reference [5].

Changes in the manuscript:

The original Fig. S15 in the previous supplementary information have been removed together with the original Fig. S16.

Sentence modified in the revised SI (lines 467-470, page 30): “OA-UCNPs exhibit a broadband at about 3433 cm⁻¹ and a weak band at 1738 cm⁻¹, corresponding to the stretching vibration of O-H and C=O (-COOH), respectively, which suggests the presence of trace amount of oleic acid on the

surfaces of nanoparticles [5, 24].”

It is interesting to see that the intensity of the nanoparticles without OA, with OA or PAA have nearly same spectrum. How the emission spectrum are tested in figure S15 and S19? Is this single nanoparticle emission? What’s the laser power used in the test?

Response: We apologize for the unclear statement in the previous SI. Regarding the measurement method, concentrated UCNPs dispersion (without OA, with OA or PAA) was spin-coated onto a glass slide repeatedly to form a uniform film of nanoparticles and then detected under the objective. The spectra were accumulated and collected from an area using the same XY laser scanning setting. As shown in Supplementary Fig. 17 (original Fig. S19) in Supplementary Information, in the two spectral curves of OA-UCNPs and PAA-UCNPs the peak ratios of I_{455}/I_{475} , I_{455}/I_{475} and I_{455}/I_{512} are a little bit different. Probably due to the very high excitation intensity (700 kW/cm²), the surface chemistry effect on the spectral profile was somewhat smeared out. As suggested by the reviewer, we have removed the UCNPs tracking data as well as the related data of the HCl treated nanoparticles (used in particles tracking study in live cells) and characterization (original Fig. S15) in the revised supplementary information.

Change in the manuscripts:

Sentences added and modified in the revised SI (lines 462-464, page 30): “Emission spectra and brightness comparison between the OA-UCNPs and the PAA-UCNPs ($I_{975}=700$ kW/cm²). Concentrated UCNPs dispersion (with OA or PAA) was spin-coated onto a glass slide repeatedly to form a uniform film of nanoparticles and then detected under the objective.”

Other incorrect reference should be double checked through the manuscript:

For example, on page 5 “similar to previous reports in high Tm³⁺ -doping UCNPs under excitation of comparable intensity^{34, 35}”. There is nothing about high Tm doping as well as comparable intensity in Ref. 34 (1%Tm, 15 W/cm²). Also Ref 5 and 24 are the same work.

Response: We apologize for the citing errors. The improper citation of Ref. [28] (Ref. [34] in our previous manuscript) has been removed, and the citing duplicate issue of Ref. [29] has been fixed in the revised manuscript.

Reviewer #3 (Remarks to the Author):

In the revision, the authors have addressed my previous comments adequately. More impressively, they have managed to add the cell skeleton image of UCNPs, by synthesizing much smaller UCNPS (11.8 nm). This has given a clear answer to the long standing question of whether UCNPs can be used as an alternative dye for cellular imaging. With the significant improvement of the revision, I am pleased to accept the manuscript at its current form.

Response: We acknowledge the reviewer for his/her evaluation on our work and recommendation on our manuscript.

Reviewer #4 (Remarks to the Author):

My recommendations and critics were mostly dealt by the authors in the revised version. The reading was substantially improved, although still some ambiguities are present. I am also very grateful for several clarifications and explanations the authors made regarding their work. I still insist on making changes in the manuscript. Main de-excitation mechanism is still not clearly described. Some of my comments are new, based on new decay time measurements, and based on comparison with the work of Reviewer 2 (Nature 2017 doi:10.1038/nature21366), which I was not aware of during my first review process.

Response: We acknowledge the reviewer for his/her general comments, and we address the de-excitation mechanism in response to point 3.

1. In the revised version, authors removed incorrect claim that I_{sat} of their system is orders of magnitude lower than that in common STED cases.

First, I was probably unclear here, but I meant to remove incorrect claim “orders of magnitude lower”, but not the comparison itself. I suggest to leave the comparison of $\text{NaYF}_4:\text{Yb}^{3+}, \text{Tm}^{3+}$ I_{sat} with I_{sat} of previously reported systems (dyes, proteins, lanthanides, vacancies), and mention I_{sat} values from multiple references. In the work of Reviewer 2, I_{sat} is 4.5 smaller. This also should be cited in your work. By the way, such discrepancy (and also others in experimental results) is probably caused not by different NPs grown under different conditions (Fig.S5 points at minimal quantum confinement effect), but rather because of the use of different depletion lasers (810 vs 808 nm) with different emission bandwidths, which might easily mismatch narrow Tm^{3+} absorption-emission bands.

Response: We thank the reviewer very much for his/her advice. As requested, the comparison of I_{sat} of $\text{NaYF}_4:\text{Yb}^{3+}, \text{Tm}^{3+}$ with those of previously reported systems including dyes, proteins, lanthanides, vacancies, has been included in the revised manuscript. The I_{sat} reported in Ref. [20] has been also cited.

Changes in the manuscript:

Sentences added/modified in the revised manuscript (lines 129-133, page 7):

“The saturation power here is significantly lower than those used in CW-STED microscopy using other biomarkers and those used in the depletion of other lanthanide ions, typically tens to hundreds of milliwatts^{13, 37, 38, 39, 40}. Our saturation power is larger than that reported in a similar work about STED microscopy of UCNPs³⁴, probably due to the difference in the nanoparticles.”

We thank the reviewer for seriously considering our mechanism and that reported in Ref. [20] and pointing on possible reasons for the discrepancies. **Herein, we would like to address the key discrepancies in experimental or simulated results in these two works and comment on the different conditions causing these discrepancies.**

The key discrepancies are summarized below.

(1) **We observed obvious emission increase at 1470 nm band (${}^3\text{H}_4 \rightarrow {}^3\text{F}_4$)** in our nanosystems after the addition of the 810 nm laser, which cannot be ascribed to the mechanism discovered in Ref. [20]. According to the simulated results in Ref. [20] shown below, Fig. RR10 (Extended Data Figure 7 in Ref. [20]), the population of the ${}^3\text{H}_4$ state (n_3) significantly decreases after the addition of the 808 nm depletion laser, **indicating emission decrease of the 1470 nm band (${}^3\text{H}_4 \rightarrow {}^3\text{F}_4$)**, which is inconsistent with our observation. This discrepancy implies that the mechanism reported in Ref. [20] is not active in our upconversion nanosystems.

Figure RR10 (Extended Data Figure 7b in Ref. [20]) Emitter populations as a function of time. The 980 nm pumping ($I_{980} = 660 \text{ kW cm}^{-2}$) is turned on at time = 0 s, while the 808 nm probing is turned on at time = 0.5 ms. Figure reproduced with permission from Nature Publishing Group.

(2) In our study, an intensity increase of the 700 nm red emission (${}^3\text{F}_3 \rightarrow {}^3\text{H}_6$) was observed in all UCNPs including those with high (10%), moderate (5%) and low (0.5%) Tm^{3+} -doping concentrations when co-irradiated by an 810 nm CW laser beam (Fig. RR11), which is different from the experimental observation reported in the Nature paper Ref. [21].

(3) We carried out more spectroscopic studies on samples with moderate Tm^{3+} concentration. Interestingly, in $\text{NaYF}_4:18\% \text{Yb}^{3+}, 5\% \text{Tm}^{3+}$, we observed emission enhancement (about 1.5 times enhancement) for the 475 nm band (${}^1\text{G}_4 \rightarrow {}^3\text{H}_6$) while the 455 nm emission was significantly depleted (~75% off) (Fig. RR11 c). However, the mechanism proposed in Ref. [20] would predict significant 475 nm emission depletion, as depicted by n_4 in Fig. RR10 (Extended Data Figure 7b in Ref. [20]), suggesting it is not active in our system.

These results are explained by our theoretical framework. In our theory, the synergistic enhancement effect of the 810 nm laser competes with the ${}^1\text{D}_2 \rightarrow {}^3\text{F}_2$ SE process (with the assistance of the CR1 process), shifting the population center of “gravity” towards lower energy state. **This competition** determines the net effect of the depletion laser on the emission

intensity of both the 455 nm band and 475 nm band. A dominant synergistic enhancement effect leads to an increased emission intensity, and vice versa. The 475 nm band, depleted indirectly, apparently exhibits a higher threshold for emission depletion than the 455 nm band, which is depleted directly by the depletion laser.

Figure RR11 Upconversion emission spectra of 0.5% (a) (Fig. 1e in the revised manuscript), 10% (b) (Fig. 1b in the revised manuscript) and 5% (c) Tm-UCNPs under 975 nm excitation with/without 810 nm irradiation. Power densities: $I_{975}=700 \text{ kW cm}^{-2}$, $I_{810}=17.7 \text{ MW cm}^{-2}$

(4) To carefully examine the relevance of the mechanism reported in Ref. [20] for our system, we carried out more 455 nm emission depletion studies by fixing the intensity of the 810 nm depletion laser (14.2 MW cm^{-2}) while adjusting the intensity of the 975 nm excitation laser in a large range from 175 kW cm^{-2} to 17.5 MW cm^{-2} . It was found that the 455-nm depletion efficiency decreases (increases) when increasing (decreasing) the intensity of the 975 nm laser, as shown in Fig. RR12. These results are consistent with our mechanism and cases of conventional CW-STED, i.e., when increasing the excitation laser intensity the emission

generally increases and it becomes harder to deplete it [21]. According to Ref. [21], the probability of spontaneous decay η_{CW} for conventional CW-STED writes as

$$\eta_{CW} = \frac{k_{ex} + k_s}{k_{ex} + (1 + \gamma)k_s},$$

with k_{ex} the excitation rate and k_s the spontaneous decay rate, and γ the effective saturation factor that accounts for the generally undesired excitation by the STED beam itself, indicating that a larger k_{ex} would cause a smaller depletion efficiency. However, the mechanism reported in Ref. [20] based on the two-photon excited population inversion (**the third excited state** 3H_4 relative to **the ground state** 3H_6) caused by the 975 nm excitation laser would predict the opposite trend, since larger (smaller) excitation intensity causes a more (less) prominent population inversion ($\Delta n = n(^3H_4) - n(^3H_6)$) between these two states, making the 455 nm emission depletion more (less) significant using identical depletion laser intensity (note: the net effect of stimulated emission is determined by the term of $\text{Power}_{808} \cdot (n(^3H_4) - n(^3H_6))$ according to Eq. (19) in Ref. [20], and $n(^3H_4)$ would equal $n(^3H_6)$ after the addition of the depletion laser (Fig. RR6)).

Figure RR12 (a) Dependence of 455-nm depletion efficiency $((I_{455}^{975} - I_{455}^{975\&810})/I_{455}^{975})$ on the 975 nm excitation laser intensity (175 kW cm^{-2} to 17.5 MW cm^{-2}) with the depletion 810-nm laser intensity kept at 14.2 MW cm^{-2} . The measured sample was $\text{NaYF}_4:18\% \text{ Yb}^{3+}, 10\% \text{ Tm}^{3+}$; (b) no population inversion ($^3H_4 \rightarrow ^3H_6$) occurs in our system.

However, inspired by the mechanism proposed by the reviewer in their nanosystem [20] and the spectral overlapping between the transitions of $^1D_2 \rightarrow ^3F_2$ and $^3H_4 \rightarrow ^3H_6$ (possibly also $^1G_4 \rightarrow ^3H_5$), referring to our responses to the Comments 6) and 7) by reviewer #2, we have realized that the stimulated emission process $^3H_4 \xrightarrow{810 \text{ nm}} ^3H_6$ should be also taken into account, which would weaken the synergistic enhancement effect of the 810 nm laser.

Fundamentally, the net effect of the 810 nm laser on the population density of the ${}^3\text{H}_4$ state is determined by the magnitude relationship between two items, i.e., $\sigma_e^{810} N({}^3\text{H}_4)$ and $\sigma_{\text{abs}}^{810} N({}^3\text{H}_6)$, where $\sigma_{\text{abs(e)}}^{810}$ denotes the absorption (stimulated emission) cross section of the transition of ${}^3\text{H}_6 \xrightarrow{810 \text{ nm}} {}^3\text{H}_4$ (${}^3\text{H}_4 \xrightarrow{810 \text{ nm}} {}^3\text{H}_6$), and $N({}^3\text{H}_4)$ and $N({}^3\text{H}_6)$ denote the population densities of the ${}^3\text{H}_4$ and ${}^3\text{H}_6$ states, respectively. Our spectroscopic results of the 1470 nm band and emission depletion efficiency dependence on the 975 nm power points on that the ground state absorption process ${}^3\text{H}_6 \xrightarrow{810 \text{ nm}} {}^3\text{H}_4$ is dominant over the stimulated emission process (${}^3\text{H}_4 \xrightarrow{810 \text{ nm}} {}^3\text{H}_6$), excluding the possibility of ${}^3\text{H}_4$ to ${}^3\text{H}_6$ population inversion in our study. The process of ${}^3\text{H}_4 \xrightarrow{810 \text{ nm}} {}^3\text{H}_6$ SE has been included in our mechanism and in our numerical simulations with the addition of relevant description.

Based on these discrepancies and our responses to the previous comments, we believe that in our work all the observations are consistent with our proposed mechanism and that the mechanism reported in Ref. [20] is not active in our system. Regarding the conditions leading to these discrepancies, we leave the discussion open, but we think nanoparticle crystal microstructure, composition and quality as well as the different laser wavelengths (810 nm v.s. 808 nm) might be involved. Future researchers should choose appropriate theory/mechanism according to their situations.

Second, another misleading claim remained in the manuscript in Lines 197-199: “The power densities for the two NIR beams were several orders of magnitude lower than those used in typical STED imaging (in order of GW/cm^2)⁴³”. In the case of the reference 43, the applied power was indeed huge, but also the resolution achieved was < 25 nm. For 66 nm resolution (achieved by the authors), lower STED-laser powers was used. I insist to remove the statement “several order of magnitude”, be more precise in comparison, and compare with multiple reports.

After all, one of the main abstract claims, lines 17-19 “we show efficient emission depletion ... which significantly lowers the laser intensity requirements of optical depletion” relies on these claims.

Response: We have modified this statement in the revised manuscript and made comparison with multiple reports on saturation powers as suggested.

Change in the manuscript:

Sentence modified (lines 246-249, page 11): “The power densities for the two CW NIR beams were much lower than those used in typical CW-STED imaging^{13, 37, 38, 39, 40, 58}, which is attractive

for biological studies and technique commercialization.”

New references added in the revised manuscript:

Willig, K. I.; Harke, B.; Medda, R.; Hell, S. W., STED microscopy with continuous wave beams. *Nature Methods* **2007**, *4*, 915-918.

Moneron, G.; Medda, R.; Hein, B.; Giske, A.; Westphal, V.; Hell, S. W., Fast STED microscopy with continuous wave fiber lasers. *Optics Express* **2010**, *18*, 1302-1309.

Bianchini, P.; Diaspro, A., Fast scanning STED and two - photon fluorescence excitation microscopy with continuous wave beam. *Journal of microscopy* **2012**, *245*, 225-228.

Beater, S.; Holzmeister, P.; Pibiri, E.; Lalkens, B.; Tinnefeld, P., Choosing dyes for cw-STED nanoscopy using self-assembled nanorulers. *Physical Chemistry Chemical Physics* **2014**, *16*, 6990-6996.

2. Lines 49-50: the work of Reviewer 2 needs to be cited here.

Response: We thank the reviewer for this suggestion. As commented by reviewer #2, our work is an independent work from theirs and was not motivated by the progress reported in the Nature article. Thus, we think it is not appropriate to cite this article in the Introduction section.

But we do cite Ref. [20] when we discuss the I_{sat} and make comparison with multiple reports as suggested. In addition, we also cite Ref. [20] in the Discussion section of our revised manuscript when discussing the advantage of the low excitation power density of upconversion nanoscopy.

Referenced added/modified in the manuscript:

34. Liu Y, *et al.* Amplified stimulated emission in upconversion nanoparticles for super-resolution nanoscopy. *Nature* **543**, 229-233 (2017).

3. Claims in the abstract Lines 17-19, “with assistance of interionic cross relaxation, which significantly lowers the laser intensity requirements” and in the introduction, lines 66-72, “We establish an efficient optical depletion approach...by enhancing the inter-Tm³⁺ interaction.” are apparently leading the reader into the wrong direction.

Based on authors' explanation of the experimental results on pages 7-9, I came to the conclusion that the solely depleting action of the 810nm laser – is de-excitation of the ¹D₂ state via stimulated emission to the ³F₂ state. From their response to my comment (1) “With high Tm³⁺ concentrations, the highly efficient CR1 process suppresses the synergistic excitation effect and simultaneously amplifies the depletion effect of the 810 nm beam by transferring the electrons at ³H₄ state to the ³F₄ state”, it is clear that CR1 does not de-excite ¹D₂ state, but de-excites ³H₄ state, facilitating faster de-population of the latter. Main point is, either with only 975 nm or with both 975+810 nm beams applied, the CR1 process will have the same effect on the de-/re-population rates of the energy states involved. The ¹D₂ will be re-populated at the same or

higher rate in the presence of 810 nm beam.

If the only process de-exciting the 1D_2 state is the stimulated emission to the 3F_2 state, then there should not be any claims that the depletion efficiency in $\text{NaYF}_4:\text{Yb}^{3+},\text{Tm}^{3+}$ is better compared to conventional STED (light-matter interaction), due to matter-matter interaction. The depletion is more efficient then solely due to better combination of $\text{Tm}^{3+} ^1D_2 \rightarrow ^3F_2$ SE cross-section and radiative emission lifetime compared to those of STED-proteins and dyes. Then it is advisable to put some literature survey on those values for Tm^{3+} transitions. The claims in lines 17-19 and 66-72 need to be re-formulated accordingly.

If there are more 1D_2 de-excitation mechanisms involved, they need to be clearly stated and explained. Basically, this is the same comment as I originally did.

Response: We thank the reviewer very much for his/her valuable comments. We agree that the $^1D_2 \rightarrow ^3F_2$ SE provides the only **direct optical** de-excitation process caused by the depletion laser, but we think **the CR1 suppresses the repopulation of the 1D_2 state from the 3H_4 state, which affects the threshold.**

(1) In the case of low Tm^{3+} doping (without the CR1 process), the relaxing electrons from 1D_2 to 3F_2 via stimulated emission and then to 3H_4 would be readily re-pumped back to the 1D_2 state via the co-excitation of the two lasers, compromising the depletion efficacy. On the contrary, in the case of elevated Tm^{3+} doping (with the CR1 process), the electrons at the 1D_2 state are first transferred to the 3H_4 state (via 3F_2) by the depletion laser, and then the CR1 process transfers a portion of the electrons at 3H_4 state to the 3F_4 state, where the CR1 process introduces a second energy dissipating pathway $^3F_4 \rightarrow ^3H_6$. **Such a shift of the population center of “gravity” towards lower energy state** (as indicated by the decreased ratio I_{455}/I_{800} with increasing the Tm^{3+} doping, as shown in Figure 1c in Ref. [15]) **helps suppress the re-population rate of the emitting state, thus enhancing stimulated emission induced depletion efficacy somewhat** (Figs. 2a-2b).

Figure RR13 (a) The proposed interionic interaction assisted STED in $\text{Yb}^{3+}, \text{Tm}^{3+}$ -co-doped UCNP system employs cross relaxation to suppress the synergistic excitation effect and enhanced the stimulated emission depletion effect of the depletion laser, reducing the requirement on depletion laser intensity; (b) without interionic cross relaxation.

(2) In the depletion process, the depletion laser has an unwanted enhancement side effect (for emission depletion), due to the undesired matching of the depletion wavelength with a ground state absorption process (${}^3\text{H}_6 \rightarrow {}^3\text{H}_4$). **A highly efficient CR1 process with high Tm^{3+} doping can also help suppress the synergistic excitation effect to the ${}^1\text{D}_2$ of the depletion laser by transferring the electrons at ${}^3\text{H}_4$ state to the ${}^3\text{F}_4$ state, again shifting the population center of “gravity” towards lower energy states, which is beneficial for 455 nm emission depletion (Fig. RR13).**

As advised, a literature survey on SE cross-section and lifetime values for Tm^{3+} transitions has been included along with changes to claims in the manuscript.

Changes in the manuscript:

As suggested, we have removed the original Figure S1 in the revised SI.

Sentences added in the revised manuscript (line 163-166, page 8):

“Given that the emission cross sections of Tm^{3+} transition are typically in the range of 10^{-21} - 10^{-20} cm^2 ^{45, 54, 55, 56}, current depletion laser intensity yields a stimulated emission lifetime from the ${}^1\text{D}_2$ state, τ_{STED} , in the order of a few μs (Supplementary Fig. 5)”

Sentences added in the revised manuscript (lines 210-225, page 10):

“Furthermore, it should be noted that the CR1 process can simultaneously enhance the STED effect of the 810 nm beam and thus lowers the depletion laser intensity requirements. As shown in Fig. 2b, in the case of low Tm^{3+} doping (without the CR1 process), the relaxing electrons

from 1D_2 to 3F_2 via stimulated emission and then to 3H_4 would be readily re-pumped back to the 1D_2 state via the synergistic excitation of the two lasers. On the contrary, in the case of high Tm^{3+} doping the electrons at the 1D_2 state are first transferred to the 3H_4 state (via 3F_2) by the depletion laser, and then the CR1 process consecutively transfers the relaxing electrons from the 3H_4 state further to the 3F_4 state, where the CR1 process in fact introduce a second energy dissipating pathway $^3F_4 \rightarrow ^3H_6$. The cooperation of the STED effect of the 810 laser beam and the CR1 process makes the population center of “gravity” shift towards lower energy states and helps suppress the re-population rate of the emitting state under two laser irradiation, thus enhancing the stimulated emission induced depletion efficacy of the depletion laser. This downward shift of the population center of “gravity”, yielding a new equilibrium among the population densities of different energy states, can also explain the passive depletion of the 1G_4 state (475 nm emission) just below the 1D_2 state at high Tm^{3+} doping (Fig. 1b).”

4. New decay time measurements of Fig. S6 and Table S1 reveal interesting details. I agree with other reviewers that the decay times under the additional 810 nm irradiation should be substantially shorter (say ~ 10 μs for 95% SE depletion, assuming ~ 200 μs radiative lifetime of the 1D_2 state). Step-wise pumping nature might affect the measured decay time, but the drop of the decay time from 200 μs to 8 μs with increase in Tm concentration points on a strong cross-relaxation process, the one that is described in the work of Reviewer 2 as CR3. If cumulative decay time due to radiative emission and cross-relaxation is $\tau = 200 \mu s / 8 \mu s$ for 0.5%/20% Tm, the decay time due to those + 810 nm depletion is $150 \mu s / 2.3 \mu s$ for 0.5%/20% Tm, then following the $1/\tau = 1/\tau_1 + 1/\tau_2$ equation, the decay time due to the 810-depletion alone drastically drops from 630 μs to 3 μs for 0.5%/20% Tm, whereas it should be independent on Tm concentration. I doubt that step-wise pumping nature can explain this 200-fold variation.

Response: We thank the reviewer’s the valuable comments.

First of all, we would like to emphasize that the step-wise pumping nature of upconversion emission could **significantly** affect the measured decay time. Güdel et al. investigated the excitation dynamics of a simplified three-level ensemble upconversion system composed of identical ions through numerical simulations based on a rate equation model [14], and ended up with the following findings. In a system where only the GSA/ETU is active, the decay of the upper level population lasts **substantially longer** than the natural decay of this state, and has a rate constant exactly twice that of the intermediate state under low-power conditions. **This actually means that the measured lifetime of the emitting state by indirect excitation could be a half of that of the intermediate state in specific conditions ($k_2 \gg k_1$)**, where $k_{1(2)}$ denotes the decay rate constant of the intermediate (upper) state. Although these conclusions are obtained based on a simplified three-level system describing the upconversion process of

Er^{3+} singly doped systems, they can be translated to other sophisticated upconversion systems involving more types of ions and energy states. In those cases, the decay time of the upper state would be affected by the lifetimes of many long-lived intermediate states in a complicated way. In $\text{Yb}^{3+}, \text{Tm}^{3+}$ -codoped nanoparticles, the decay rate of the $^1\text{D}_2$ state is significantly larger than those of intermediate states [6], thus the measured lifetime using an indirect pumping approach would be influenced by those of intermediate states implicitly. According to the above discussion, a direct pumping approach (e.g., 355 nm ultraviolet excitation for $^1\text{D}_2$) is desired when measuring the lifetime of the upper emitting state. Unfortunately, due to instrument limitation, we could not manage to perform such measurements (EX/DE/EM: 355/810/455 nm) in our lab.

The measured lifetime using step-wise excitation is obviously dependent on concrete upconversion pathway possibly involving many mechanisms, including light absorption, energy transfer, and cross-relaxation. When the doping concentration changes, a significant change could occur in the upconversion pathway by manifesting strong cross relaxations at high concentrations. For example, our experimental results have shown that with the increase of Tm^{3+} doping concentration (from 0.5% to 10%), the ratio I_{1470}/I_{1800} shows a very significant decrease (from about 5:1 to 1:32), indicating a **160 fold enhancement for I_{1800} with respect to I_{1470} (Fig. 2e, Supplementary Fig. 9 in the main text)**. The enhancement would be larger than 160 if comparing the data of 0.5% Tm^{3+} and 20% Tm^{3+} . Thus, the decay time of the upper state would be influenced by the lifetimes of intermediate state quite differently. In this situation, it would be rather rough to use an equation like $1/\tau=1/\tau_1+1/\tau_2$ to quantitatively evaluate the decay time due to the 810-depletion alone.

In addition, in the present nanosystem, the depletion laser has an adverse synergistic enhancement effect weakened by CR1 process, which was not considered in this equation ($1/\tau=1/\tau_1+1/\tau_2$). This raises questions about the validity of this equation in our case, although it is valid for conventional STED.

In summary, this dramatic change in the lifetime equation is plausibly consistent with our theory. We thank the reviewer very much for the comment on the drop of the decay time from 200 μs to 8 μs with increase in Tm concentration, which points on a strong cross-relaxation process ($^1\text{D}_2+^3\text{H}_6 \rightarrow ^3\text{F}_2+^3\text{H}_4$). We have realized the occurrence of this CR process and have included it in our theory and numerical simulations in order to make our theoretical description more accurate.

Figure RR14 (Fig. 2a in the revised manuscript) Proposed optical emission depletion mechanism of the 455 nm upconversion band of Tm^{3+} of $\text{NaYF}_4:18\% \text{Yb}^{3+}, 10\% \text{Tm}^{3+}$ nanoparticles.

Changes in manuscript:

Sentences added in the revised manuscript (lines 163-172, page 8):

“Given that the emission cross sections of Tm^{3+} transition are typically in the range of 10^{-21} - 10^{-20} cm^2 ^{45, 54, 55, 56}, current depletion laser intensity yields a stimulated emission lifetime from the $^1\text{D}_2$ state, τ_{STED} , in the order of a few μs (Supplementary Fig. 5), which are three orders of magnitude longer than stimulated emission lifetime in fluorescent organic dyes (ns to ps). The less dramatic lifetime change in some samples than expected was probably ascribed to the multiple effects of the depletion laser (discussed below), as well as the stepwise upconversion pumping approach in the lifetime measurement, where the measured decay lifetimes are often not solely determined by the lifetime of the emitting state but also reflect the radiative properties of many long-lived intermediate states ⁵⁷.”

5. Regarding the response of the authors to my comment (4) “Our model and numerical simulation are not intended to quantitatively evaluate the absolute optical depletion efficiency induced by the 810 nm laser, due to the difficulty of experimentally obtaining the parameter values needed (thus they have to be estimated based on the literature).”

I agree that it is difficult to measure those constants, even literature values are often vary quite a lot. Nevertheless, I recommend tabulating all the constants the authors used in these simulations, and where possible, compare those constants with literature values.

Response: We thank the reviewer very much for the advice, according to which we have modified our manuscript by e.g. tabulating all the constants we used in these simulations (some constants are estimated from literature).

We have made some modifications of the mechanism, mainly the inclusion of the $^1D_2 + ^3H_6 \rightarrow ^3F_2 + ^3H_4$ CR process and the $^3H_4 \xrightarrow{810 \text{ nm}} ^3H_6$ SE process (Fig. RR15). Inspired by the mechanism discovered in Ref. [20] and our recognition of the spectral overlap between the transitions of $^1D_2 \rightarrow ^3F_2$ and $^3H_4 \rightarrow ^3H_6$, we have realized that the stimulated emission process $^3H_4 \xrightarrow{810 \text{ nm}} ^3H_6$, weakening the synergistic enhancement effect of the 810 nm laser, should be taken into account in the simulation, since this GSA and SE between the 3H_6 and 3H_4 states coexist. In addition, the excited state absorption process $^3F_2 \xrightarrow{810 \text{ nm}} ^1D_2$ should be also taken into account. In this case, in our simulation model we have included all the six processes (Fig. RR15, excitation and stimulated emission for the three pairs of transitions), which theoretically occur during 810-nm laser and Tm-UCNPs interaction regardless of their efficiency.

Figure RR15 (Figure S10 in the revised Supplementary information) The energy diagram used in the simulation of the proposed 455-nm luminescence depletion of NaYF₄: 18% Yb³⁺, 10% Tm³⁺.

Fundamentally, the direct (net) effect of the 810 nm laser on the population density of the 3H_4 state is determined by the magnitude relationship between $\sigma_{se}^{810} N(^3H_4)$ and $\sigma_{abs}^{810} N(^3H_6)$, where $\sigma_{abs(se)}^{810}$ denotes the absorption (stimulated emission) cross section of the transition of $^3H_6 \xrightarrow{810 \text{ nm}} ^3H_4$ ($^3H_4 \xrightarrow{810 \text{ nm}} ^3H_6$), and $N(^3H_4)$ and $N(^3H_6)$ denote the population densities of the 3H_4 and 3H_6 states, respectively. **Our spectroscopic results of the 1470 nm band ($^3H_4 \rightarrow ^3F_4$) and emission depletion efficiency dependence on the 975 nm power (Fig. RR12) indicates that the GSA process $^3H_6 \xrightarrow{810 \text{ nm}} ^3H_4$ is dominant over the stimulated emission process**

(${}^3\text{H}_4 \xrightarrow{810 \text{ nm}} {}^3\text{H}_6$). The stimulated emission process (${}^1\text{D}_2 \xrightarrow{810 \text{ nm}} {}^3\text{F}_2$) is dominant over the ESA process ${}^3\text{F}_2 \xrightarrow{810 \text{ nm}} {}^1\text{D}_2$, due to a larger population density at the ${}^1\text{D}_2$ state than the ${}^3\text{F}_2$ state and a larger emission cross section than the absorption cross section at this wavelength. For simplicity, the net effects were plotted in mechanism diagram of the main text (Figure RR14, Figure 2a).

As requested, we have tabulated all the constants used in our modelling and simulation, as shown in Table RR2. We have re-performed simulations based on the above modifications. The results are presented in Figure RR16. More details about the simulations can be found in the revised Supplementary Information.

Table added in the Supplementary Information:

Table RR2 (Supplementary Table 2 in the revised Supplementary Information) The values of key constants and rate parameters used in the simulations for 455-nm emission modulation of UCNP at low and high Tm^{3+} -doping.

	c_1 ($\text{cm}^3 \text{ s}^{-1}$)	c_2 ($\text{cm}^3 \text{ s}^{-1}$)	c_3 ($\text{cm}^3 \text{ s}^{-1}$)	c_4 ($\text{cm}^3 \text{ s}^{-1}$)	w_1 ($\text{cm}^3 \text{ s}^{-1}$)	w_2 ($\text{cm}^3 \text{ s}^{-1}$)
10%	$2.0 \times 10^{-16\text{a}}$	$5.3 \times 10^{-16\text{a}}$	$5.3 \times 10^{-16\text{a}}$	$6.0 \times 10^{-16\text{a}}$	$6.0 \times 10^{-17\text{b}}$	$3.0 \times 10^{-16\text{b}}$
0.5%	$5.0 \times 10^{-19\text{a}}$	$1.3 \times 10^{-18\text{a}}$	$1.3 \times 10^{-18\text{a}}$	$1.5 \times 10^{-18\text{a}}$	$1.0 \times 10^{-17\text{b}}$	$5.0 \times 10^{-17\text{b}}$
	w_3 ($\text{cm}^3 \text{ s}^{-1}$)	w_4 ($\text{cm}^3 \text{ s}^{-1}$)	$\sigma_{\text{d}1}^{\text{a}}$ (cm^2)	$\sigma_{\text{d}2}^{\text{a}}$ (cm^2)	$\sigma_{\text{d}3}^{\text{a}}$ (cm^2)	$\sigma_{\text{d}1}^{\text{se}}$ (cm^2)
10%	$2.5 \times 10^{-16\text{b}}$	$3.0 \times 10^{-16\text{b}}$	$9.0 \times 10^{-22\text{c}}$	$1.0 \times 10^{-24\text{c}}$	$2.0 \times 10^{-23\text{c}}$	$3.0 \times 10^{-21\text{c}}$
0.5%	$4.2 \times 10^{-17\text{b}}$	$5.0 \times 10^{-17\text{b}}$	$9.0 \times 10^{-22\text{c}}$	$1.0 \times 10^{-24\text{c}}$	$2.0 \times 10^{-23\text{c}}$	$3.0 \times 10^{-21\text{c}}$
	$\sigma_{\text{d}2}^{\text{se}}$ (cm^2)	$\sigma_{\text{d}3}^{\text{se}}$ (cm^2)	$\tau_{\text{yb}1}$ (s)	τ_1 (s)	τ_3 (s)	τ_4 (s)
10%	$5.0 \times 10^{-24\text{c}}$	$9.0 \times 10^{-21\text{c}}$	$3.3 \times 10^{-4\text{d}}$	$1.9 \times 10^{-3\text{d}}$	$3.7 \times 10^{-4\text{d}}$	$4.7 \times 10^{-4\text{d}}$
0.5%	$5.0 \times 10^{-24\text{c}}$	$9.0 \times 10^{-21\text{c}}$	$5.0 \times 10^{-4\text{d}}$	$2.8 \times 10^{-3\text{d}}$	$5.6 \times 10^{-4\text{d}}$	$7.0 \times 10^{-4\text{d}}$
	τ_7 (s)	τ_6 (s)	τ_7 (s)	τ_8 (s)	β_2 (s^{-1})	β_4 (s^{-1})
10%	$8.0 \times 10^{-4\text{d}}$	$2.5 \times 10^{-4\text{d}}$	$1.5 \times 10^{-4\text{d}}$	$3.0 \times 10^{-5\text{d}}$	$3.4 \times 10^{4\text{e}}$	$1.0 \times 10^{5\text{e}}$
0.5%	$1.2 \times 10^{-3\text{d}}$	$3.8 \times 10^{-4\text{d}}$	$2.3 \times 10^{-4\text{d}}$	$4.5 \times 10^{-5\text{d}}$	$3.4 \times 10^{4\text{e}}$	$1.0 \times 10^{5\text{e}}$
	b_{30}	b_{31}	b_{60}	b_{61}	b_{62}	b_{63}
10%	0.85^{f}	0.15^{f}	0.65^{f}	0.15^{f}	0.16^{f}	0.14^{f}
0.5%	0.85^{f}	0.15^{f}	0.65^{f}	0.15^{f}	0.16^{f}	0.14^{f}
	b_{70}	b_{71}	b_{73}	b_{74}	b_{75}	b_{81}
10%	0.48^{f}	0.40^{f}	0.03^{f}	0.03^{f}	0.06^{f}	0.58^{f}
0.5%	0.48^{f}	0.40^{f}	0.03^{f}	0.03^{f}	0.06^{f}	0.58^{f}
	b_{82}					
10%	0.42^{f}					
0.5%	0.42^{f}					

^a Estimated from Ivanova *et al.* [30] and Tkachuk *et al.* [31]

^b Estimated from Braud *et al.* [32]

^c Estimated from Peterka *et al.* [33] Medoidze *et al.* [34] and Smith *et al.* [35]

^d Estimated from Villanueva-Delgado *et al.* [16] and Walsh *et al.* [36]

^e From Ivanova *et al.* [30]

^f From Villanueva-Delgado *et al.* [16]

Figure RR16 (Supplementary Fig. 11 in the revised Supplementary Information) Simulated dependence of the population changes of the 1D_2 states on the power of the 810 nm depletion beam for the case of (a) high and (b) low Tm^{3+} -doping UCNPs.

Changes in the manuscript:

Sentences added and modified in the revised SI (lines 271-309, pages 12-13):

“Due to the matching of the 810 nm laser (FWHM 4 nm) with multiple transitions, including $^3H_6 \xrightarrow{810\text{ nm}} ^3H_4$, $^3H_5 \xrightarrow{810\text{ nm}} ^1G_4$ and $^3F_2 \xrightarrow{810\text{ nm}} ^1D_2$ [11-13], three light absorption and three stimulated emission processes at 810 nm are all considered in this model, with different cross sections (Supplementary Table 2). Regarding the net effect of the 810 nm laser on the population density of the 3H_4 state, fundamentally determined by the magnitude relationship between two items, namely, $\frac{\sigma_{d1}^a I_d}{h\nu_d} n_0$ and $\frac{\sigma_{d1}^{se} I_d}{h\nu_d} n_3$, our spectroscopic result of the 1470 nm band ($^3H_4 \rightarrow ^3F_4$) (Fig. 2d in the main text) indicates that the GSA process $^3H_6 \xrightarrow{810\text{ nm}} ^3H_4$ is dominant over the stimulated emission process ($^3H_4 \xrightarrow{810\text{ nm}} ^3H_6$). In addition, the stimulated emission process ($^1D_2 \xrightarrow{810\text{ nm}} ^3F_2$) is supposed to be dominant over the ESA process $^3F_2 \xrightarrow{810\text{ nm}} ^1D_2$, due to a larger population density at the 1D_2 state than the 3F_2 state and a larger emission cross section than the absorption cross section at this wavelength. For simplicity, only the net effects of the 810 laser on the $^3H_6 \xrightarrow{810\text{ nm}} ^3H_4$, $^3H_5 \xrightarrow{810\text{ nm}} ^1G_4$ and

${}^3F_2 \xrightarrow{810 \text{ nm}} {}^1D_2$ transitions are illustrated in the mechanism diagram (see Fig. 2a in the main text).

The Tm^{3+} concentration dependence of the depletion efficiency of the 455 nm emission was investigated by varying the parameter values, particularly those for cross relaxation processes. The values used for the main parameters for both low and high Tm^{3+} -doping samples are tabulated in Supplementary Table 2. The simulated results are presented in Supplementary Fig. 11.”

The rate equations of each energy state modified in the revised SI (pages 11-13)

$$\text{Tm}^{3+}({}^3H_6): \frac{dn_0}{dt} = -\sum_{i=1}^8 \frac{dn_i}{dt}$$

$$\text{Tm}^{3+}({}^3F_4): \frac{dn_1}{dt} = \beta_2 n_2 + 2c_1 n_0 n_3 + c_2 n_3 n_5 + c_3 n_3 n_5 + b_{31} \frac{n_3}{\tau_3} + b_{51} \frac{n_5}{\tau_5} + b_{61} \frac{n_6}{\tau_6} + b_{71} \frac{n_7}{\tau_7} + b_{81} \frac{n_8}{\tau_8} - w_2 n_{Yb1} n_1 - \frac{n_1}{\tau_1}$$

$$\text{Tm}^{3+}({}^3H_5): \frac{dn_2}{dt} = \frac{\sigma_{d2}^{se} I_d}{h\nu_d} n_6 - \frac{\sigma_{d2}^a I_d}{h\nu_d} n_2 + w_1 n_{Yb1} n_0 + b_{62} \frac{n_6}{\tau_6} + b_{82} \frac{n_8}{\tau_8} + \beta_3 n_3 - \beta_2 n_2 - \frac{n_2}{\tau_2}$$

$$\text{Tm}^{3+}({}^3H_4): \frac{dn_3}{dt} = \frac{\sigma_{d1}^a I_d}{h\nu_d} n_0 - \frac{\sigma_{d1}^{se} I_d}{h\nu_d} n_3 + b_{63} \frac{n_6}{\tau_6} + b_{73} \frac{n_7}{\tau_7} + \beta_4 n_4 + c_4 n_0 n_7 - c_1 n_0 n_3 - c_2 n_3 n_6 - c_3 n_3 n_6 - w_3 n_{Yb1} n_3 - \beta_3 n_3 - \frac{n_3}{\tau_3}$$

$$\text{Tm}^{3+}({}^3F_3): \frac{dn_4}{dt} = \beta_5 n_5 + b_{74} \frac{n_7}{\tau_7} - \beta_4 n_4 - \frac{n_4}{\tau_4}$$

$$\text{Tm}^{3+}({}^3F_2): \frac{dn_5}{dt} = \frac{\sigma_{d3}^{se} I_d}{h\nu_d} n_7 - \frac{\sigma_{d3}^a I_d}{h\nu_d} n_5 + w_2 n_{Yb1} n_1 + c_4 n_0 n_7 + b_{75} \frac{n_7}{\tau_7} + \beta_6 n_6 - \beta_5 n_5 - \frac{n_5}{\tau_5}$$

$$\text{Tm}^{3+}({}^1G_4): \frac{dn_6}{dt} = \frac{\sigma_{d2}^a I_d}{h\nu_d} n_2 - \frac{\sigma_{d2}^{se} I_d}{h\nu_d} n_6 + w_3 n_{Yb1} n_3 - c_2 n_3 n_6 - c_3 n_3 n_6 - \beta_6 n_6 - \frac{n_6}{\tau_6}$$

$$\text{Tm}^{3+}({}^1D_2): \frac{dn_7}{dt} = -\frac{\sigma_{d3}^{se} I_d}{h\nu_d} n_7 + \frac{\sigma_{d3}^a I_d}{h\nu_d} n_5 + c_2 n_3 n_6 + c_3 n_3 n_6 - w_4 n_{Yb1} n_7 - c_4 n_0 n_7 - \frac{n_7}{\tau_7}$$

$$\text{Tm}^{3+}({}^1I_6): \frac{dn_8}{dt} = w_4 n_{Yb1} n_7 - \frac{n_8}{\tau_8}$$

$$\text{Yb}^{3+}({}^2F_{7/2}): \frac{dn_{Yb0}}{dt} = -\frac{dn_{Yb1}}{dt}$$

$$\text{Yb}^{3+}({}^2F_{5/2}): \frac{dn_{Yb1}}{dt} = \frac{\sigma_p^a I_p}{h\nu_p} n_{Yb0} - (w_1 n_0 + w_2 n_1 + w_3 n_3 + w_4 n_7) n_{Yb1} - \frac{n_{Yb1}}{\tau_{Yb1}}$$

New references added in the revised SI:

- Zhang, H., et al., Mechanisms of the blue emission of $\text{NaYF}_4:\text{Tm}^{3+}$ nanoparticles excited by an 800 nm continuous wave laser. *Phys. Chem. Chem. Phys.*, 2016. **18**(37): p. 25905-25914.

12. Loiko, P. and M. Pollnau, Stochastic Model of Energy-Transfer Processes Among Rare-Earth Ions. Example of $\text{Al}_2\text{O}_3:\text{Tm}^{3+}$. *J. Phys. Chem. C*, 2016. **120**(46): p. 26480-26489.
13. Simpson, D.A., et al., Visible and near infra-red up-conversion in $\text{Tm}^{3+}/\text{Yb}^{3+}$ co-doped silica fibers under 980 nm excitation. *Optics Express*, 2008. **16**(18): p. 13781-13799.
26. Ivanova, S.E., et al., Spectroscopic study of thulium-activated double sodium yttrium fluoride $\text{Na}_{0.4}\text{Y}_{0.6}\text{F}_{2.2}:\text{Tm}^{3+}$ crystals:I. Intensity of spectra and luminescence kinetics. *Optics & Spectroscopy*, 2008. **105**(2): p. 228-241.
27. Tkachuk, A.M., et al., Luminescence self-quenching in Tm^{3+} : YLF crystals: II. The luminescence decay and macrorates of energy transfer. *Optics & Spectroscopy*, 2001. **90**(1): p. 78-88.
28. Braud, A., et al., Energy-transfer processes in Yb:Tm-doped KY_3F_{10} , LiYF_4 , and BaY_2F_8 single crystals for laser operation at 1.5 and 2.3 μm . *Physical Review B*, 2000. **61**(8): p. 5280.
29. Peterka, P., et al., Theoretical modeling of fiber laser at 810 nm based on thulium-doped silica fibers with enhanced $^3\text{H}_4$ level lifetime. *Optics Express*, 2011. **19**(3): p. 2773.
30. Medoidze, T.D. and Z.G. Melikishvili, Ultraviolet and visible emission cross-sections for $\text{Tm}^{3+}:\text{YLiF}_4$ laser system. *Laser Physics Letters*, 2010. **1**(2): p. 65–68.
31. Smith, A.V. and J.J. Smith, Mode instability thresholds for Tm-doped fiber amplifiers pumped at 790 nm. *Optics Express*, 2015. **24**(2): p. 975.
32. Villanueva-Delgado, P., D. Biner, and K.W. Krämer, Judd–Ofelt analysis of $\beta\text{-NaGdF}_4:\text{Yb}^{3+}, \text{Tm}^{3+}$ and $\beta\text{-NaGdF}_4:\text{Er}^{3+}$ single crystals. *Journal of Luminescence*, 2016. **189**: p. 84-90.
33. Walsh, B.M. and N.P. Barnes, Comparison of Tm:ZBLAN and Tm:silica fiber lasers; Spectroscopy and tunable pulsed laser operation around 1.9 μm . *Applied Physics B*, 2004. **78**(3-4): p. 325-333.

6. Line 179 (Fig.2b). Is it a misspell, did you mean Fig. 2e,f?

Response: We thank the reviewer for his/her reminder. We intended to direct the readers to the schematic shown in Fig. 2b. The cross reference is correct here.

References:

- [1] G. Liu, *Spectroscopic Properties of Rare Earths in Optical Materials*, Springer Berlin Heidelberg 2005.
- [2] G.H. Dieke, H.M. Crosswhite, H. Crosswhite, *Spectra and energy levels of rare earth ions in crystals*, Interscience Publishers 1968.
- [3] J.B. Gruber, W.F. Krupke, J.M. Poindexter, *The Journal of Chemical Physics*, 41 (1964) 3363-3377.
- [4] S. Ivanova, A. Tkachuk, A. Mirzaeva, F. Pellé, *Optics and Spectroscopy*, 105 (2008) 228-241.
- [5] A. Malakhovskii, I. Edelman, A. Sokolov, V. Temerov, S. Gnatchenko, I. Kachur, V. Piryatinskaya, *Journal of Alloys and Compounds*, 459 (2008) 87-94.
- [6] S.E. Ivanova, A.M. Tkachuk, A. Mirzaeva, F. Pelle, *Optics and Spectroscopy*, 105 (2008) 228-241.
- [7] R.W. Schwartz, T.R. Faulkner, F.S. Richardson, *Molecular Physics*, 38 (1979) 1767-1780.
- [8] J.B. Gruber, R.P. Leavitt, C.A. Morrison, *The Journal of Chemical Physics*, 74 (1981) 2705-2709.
- [9] P. Peterka, I. Kasik, A. Dhar, B. Dussardier, W. Blanc, *Opt. Express*, 19 (2011) 2773-2781.
- [10] F. Wang, Y. Han, C.S. Lim, Y. Lu, J. Wang, J. Xu, H. Chen, C. Zhang, M. Hong, X. Liu, *Nature*, 463 (2010) 1061-1065.
- [11] G. Wang, W. Qin, J. Zhang, L. Wang, G. Wei, P. Zhu, R. Kim, *Journal of Alloys and Compounds*, 475 (2009) 452-455.
- [12] D.A. Simpson, W.E.K. Gibbs, S.F. Collins, W. Blanc, B. Dussardier, G. Monnom, P. Peterka, G.W. Baxter, *Optics Express*, 16 (2008) 13781-13799.
- [13] H. Zhang, T. Jia, X. Shang, S. Zhang, Z. Sun, J. Qiu, *Phys. Chem. Chem. Phys.*, 18 (2016) 25905-25914.
- [14] D. Gamelin, H. Gudel, *Upconversion Processes in Transition Metal and Rare Earth Metal Systems*, in: H. Yersin (Ed.) *Transition Metal and Rare Earth Compounds*, Springer Berlin / Heidelberg 2001, pp. 1-56.
- [15] J. Zhao, D. Jin, E.P. Schartner, Y. Lu, Y. Liu, A.V. Zvyagin, L. Zhang, J.M. Dawes, P. Xi, J.A. Piper, E.M. Goldys, T.M. Monro, *Nat. Nano.*, 8 (2013) 729-734.
- [16] P. Villanueva-Delgado, D. Biner, K.W. Krämer, *Journal of Luminescence*, 189 (2016) 84-90.
- [17] B.M. Walsh, N.P. Barnes, *Applied Physics B*, 78 (2004) 325-333.
- [18] I.R. Martín, V.D. Rodríguez, Y. Guyot, S. Guy, G. Boulon, M.F. Joubert, *Journal of Physics: Condensed Matter*, 12 (2000) 1507-1507.
- [19] M. Dulick, G.E. Faulkner, N.J. Cockroft, D.C. Nguyen, *Journal of Luminescence*, 48 (1991) 517-521.
- [20] Y. Liu, Y. Lu, X. Yang, X. Zheng, S. Wen, F. Wang, X. Vidal, J. Zhao, D. Liu, Z. Zhou, C. Ma, J. Zhou, J.A.

- Piper, P. Xi, D. Jin, *Nature*, 543 (2017) 229-233.
- [21] M. Leutenegger, C. Eggeling, S.W. Hell, *Opt. Express*, 18 (2010) 26417-26429.
- [22] K.I. Willig, B. Harke, R. Medda, S.W. Hell, *Nat Meth*, 4 (2007) 915-918.
- [23] A.V. Smith, J.J. Smith, *Opt. Express*, 24 (2016) 975-992.
- [24] T.D. Medoidze, Z.G. Melikishvili, *Laser Physics Letters*, 1 (2004) 65-68.
- [25] J. Hanne, H.J. Falk, F. Gorlitz, P. Hoyer, J. Engelhardt, S.J. Sahl, S.W. Hell, *Nat Commun*, 6 (2015) 7127.
- [26] C. Toma, M.F. Pittenger, K.S. Cahill, B.J. Byrne, P.D. Kessler, *Circulation*, 105 (2002) 93-98.
- [27] B.N.G. Giepmans, T.J. Deerinck, B.L. Smarr, Y.Z. Jones, M.H. Ellisman, *Nat Meth*, 2 (2005) 743-749.
- [28] F. Wang, R. Deng, J. Wang, Q. Wang, Y. Han, H. Zhu, X. Chen, X. Liu, *Nat. Mater.*, 10 (2011) 968-973.
- [29] B. Zhou, B. Shi, D. Jin, X. Liu, *Nat. Nano.*, 10 (2015) 924-936.
- [30] S.E. Ivanova, A.M. Tkachuk, A. Mirzaeva, F. Pellé, *Optics & Spectroscopy*, 105 (2008) 228-241.
- [31] A.M. Tkachuk, I.K. Razumova, E.Y. Perlin, M.F. Joubert, R. Moncorge, *Optics & Spectroscopy*, 90 (2001) 78-88.
- [32] A. Braud, S. Girard, J.L. Doualan, M. Thuau, R. Moncorgé, A.M. Tkachuk, *Physical Review B*, 61 (2000) 5280.
- [33] P. Peterka, I. Kasik, A. Dhar, B. Dussardier, W. Blanc, *Optics Express*, 19 (2011) 2773.
- [34] T.D. Medoidze, Z.G. Melikishvili, *Laser Physics Letters*, 1 (2010) 65-68.
- [35] A.V. Smith, J.J. Smith, *Optics Express*, 24 (2015) 975.
- [36] B.M. Walsh, N.P. Barnes, *Applied Physics B*, 78 (2004) 325-333.

Reviewers' comments:

Reviewer #2 (Remarks to the Author):

In the revised version, authors have fixed most of our suggestions, and the rebuttal letter is getting very long now, but the outstanding argument is around the emission wavelength of $1D2 \rightarrow 3F2,3$ transitions and depletion mechanism. According to our work (Nature Nano. 2013, 8, 729-734), if the authors agree that the 740 nm emission belongs to the $1D2 \rightarrow 3F3$ transition, the small energy gap (difference) between the $3F3$ and the $3F2$ would be too small for the other peak ($1D2 \rightarrow 3F2$ transition) hidden behind 800 emission bands. We would rather leave this analysis and all the discussed literature open for future discussions.

Instead, it would be more straightforward for the authors to support their proposed mechanism by running a 'pump-probe' experiment, under 975 excitation, and scan the probe laser from 700 nm to 810 nm to check the emission and depletion spectrum. This would be one more step by tuning the laser output wavelength from a Ti:sapphire laser (Laser 2, Mira 900, Coherent) used in this work.

Some additional comments on the FTIR characterizations: the authors reported that "OA-UCNPs exhibit a broadband at about 3433 cm^{-1} and a weak band at 1738 cm^{-1} , corresponding to the stretching vibration of O-H and C=O (-COOH), respectively, which suggests the presence of trace amount of oleic acid on the surfaces of nanoparticles [5, 24]." Actually, in the Ref 24, Liu et al reported "the weak stretching mode of the -COOH group at 1709 cm^{-1} suggests the presence of trace amounts of free oleic acid on the surface of nanocrystals before ligand exchange. This band is significantly enhanced and shifted to 1732 cm^{-1} after ligand exchange by PAA, suggesting an increase in the quantity of the -COOH groups on the particle surface." This means the 1732 cm^{-1} peak belongs to PAA modified UCNPs, so why a weak band at 1738 cm^{-1} suggests the presence of trace amount of oleic acid? I suggest the authors should focus on the main peak shift after the PAA modification, which is common in the references. In this work, the main peak around 1560 cm^{-1} should belong to the COO- of OA- on the surface of OA-UCNPs, while this peak shift to 1647 cm^{-1} after the PAA modification.

Reviewer #4 (Remarks to the Author):

In the revised version, the authors properly addressed most of my recommendations. Now the article more precisely cites previous research, and adequately compares its results with common STED - microscopy performance and the work of reviewer #2. I do not question the differences in the experimental results of the authors work and the work of reviewer #2 due to different experimental conditions. The models used by both groups to describe the depletion mechanisms are also different, which does not seem questionable to me at this stage. Taking into account novelty of the demonstration of two-color super-resolution imaging and super-resolution cellular structure imaging using UCNPs the article is worth publishing in Nature Communications.

Manuscript No. NCOMMS-17-01804B

Title: Achieving high-efficiency emission depletion nanoscopy by employing cross relaxation in upconversion nanoparticles

Response Letter to Reviewers

Dear Reviewers,

Many thanks for your recommendation and positive evaluation on our manuscript as well as your further careful consideration and valuable comments on our revised manuscript, which again have helped us improve the manuscript. In the following, we provide a point-by-point response to the comments, together with the corresponding changes in the manuscript. As below, the reviewers' comments are written in **black** and our responses to them in **blue**. The important amendments or changes to the manuscript are given after the response in **red**.

Reviewers' comments:

Reviewer #2 (Remarks to the Author):

(1) In the revised version, authors have fixed most of our suggestions, and the rebuttal letter is getting very long now, but the outstanding argument is around the emission wavelength of $^1D_2 \rightarrow ^3F_{2,3}$ transitions and depletion mechanism. According to our work (Nature Nano. 2013, 8, 729-734), if the authors agree that the 740 nm emission belongs to the $^1D_2 \rightarrow ^3F_3$ transition, the small energy gap (difference) between the 3F_3 and the 3F_2 would be too small for the other peak ($^1D_2 \rightarrow ^3F_2$ transition) hidden behind 800 emission bands. We would rather leave this analysis and all the discussed literature open for future discussions.

Response: We thank the reviewer for his/her comments. We fully agree that the emission centered at around 740 nm originates from the $^1D_2 \rightarrow ^3F_3$ transition of Tm^{3+} . Although the energy gap between the 3F_3 and the 3F_2 states is not that big, we have thoroughly discussed how it affects the spectral range around 800 nm.

We appreciate the reviewer's open attitude on this discussion.

(2) Instead, it would be more straightforward for the authors to support their proposed mechanism by running a 'pump-probe' experiment, under 975 excitation, and scan the probe laser from 700 nm to 810 nm to check the emission and depletion spectrum. This would be one more step by tuning the laser output wavelength from a Ti:sapphire laser (Laser 2, Mira 900, Coherent) used in this work.

Response: We thank the reviewer for his/her comments. As requested, we have carried out a 'pump-probe' experiment with a fixed 975 nm excitation laser output while scanning the probe laser from 700 nm to 815 nm (constant power, FWHM $\Delta\lambda=4$ nm). The depletion-laser-

wavelength dependent depletion efficiency data (Left, Fig. 1) well supports the proposed optical depletion mechanism of the 455 nm emission of high Tm³⁺-doping (NaYF₄:18% Yb³⁺,10% Tm³⁺) nanoparticles, where a depletion efficiency peak at around 810 nm (corresponding to the ¹D₂ → ³F₂ transition) was observed. This is consistent with the observed excitation spectrum of 455-nm emission under sole irradiation of the probe laser beam (Right, Fig. 1; the inset is an enlarged view), where a probe laser beam at around 810 nm gives rise to weaker luminescence background than other wavelengths. Interestingly, an extra depletion region centered at around 750 nm was observed (Left, Fig. 1), which is associated with the ¹D₂ → ³F₃ transition. This reveals that the population of the ¹D₂ state can also be efficiently depleted to the ³F₃ state via stimulated emission with the laser irradiation of proper wavelength, analogous to our proposed stimulated emission pathway by an 810 nm laser associated with the ¹D₂ → ³F₂ transition.

Figure 1 Left (Fig. 2f in the revised manuscript): Dependence of 455-nm depletion efficiency (DE, $(I_{455}^{975} - I_{455}^{975\&810})/I_{455}^{975}$) of NaYF₄:18% Yb³⁺,10% Tm³⁺ on the depletion laser wavelength ($\Delta\lambda=4$ nm) ranging from 700 nm to 815 nm. Negative depletion efficiency represents emission enhancement. The intensity of the 975-nm excitation laser and the depletion laser were kept at 700 kW cm⁻² and 17.7 MW cm⁻², respectively. Right (Fig. 2g in the revised manuscript): The excitation spectrum of the 455-nm emission of NaYF₄:18% Yb³⁺,10% Tm³⁺ solely excited by the depletion laser with wavelength ranging from 700 nm to 815 nm; the inset is an enlarged view.

Figure 2 (Supplementary Figure 13 in the revised SI) Proposed energy level mechanism for the depletion-laser-wavelength (700-815 nm) dependent depletion/enhancement effect on the 455-nm emission of NaYF₄:18% Yb³⁺/10% Tm³⁺ UCNPs. (a) The laser wavelengths in the range of 700 - 730 nm (indicated by letter “a”) can match the ground state absorption (GSA) process of the ³H₆ → ³F₃ transition [1, 2]; (b) The laser wavelengths in the range of 720 - 770 nm (indicated by letter “b”) can match the emission spectrum of the ¹D₂ → ³F₃ transition (centered at around 745 nm) [3, 4]. (c) The laser wavelengths in the range of 730-790 nm (indicated by letter “c”) can match the excited state absorption (ESA) spectrum of the ³H₅ → ¹G₄ transition (centered at around 765 nm) [5]; (d) The laser wavelengths in the range of 770 - 815 nm (indicated by letter “d”) can match both the GSA spectrum of the ³H₆ → ³H₄ transition (centered at around 780 nm) and the emission spectrum of the ¹D₂ → ³F₂ transition [5].

Change in the manuscript:

Paragraph added in the revised manuscript (Line 238-261, Page 11-12): “The proposed optical depletion mechanism for the 455 nm emission of high Tm³⁺-doping (NaYF₄:18% Yb³⁺,10% Tm³⁺) nanoparticles is also well supported by the depletion-laser-wavelength dependent depletion efficiency data, which were obtained by fixing the wavelength (975 nm) and power of the excitation laser and scanning the wavelength of the second beam from 700 nm to 810 nm while keeping the power, as shown in Fig. 2f. A depletion efficiency peak at around 810 nm (corresponding to the ¹D₂ → ³F₂ transition) was observed. This is consistent with the observed excitation spectrum for 455-nm emission under sole irradiation of the second beam (Fig. 2g), where a second laser beam at around 810 nm gives rise to weaker luminescence background than other wavelengths. Interestingly, an extra depletion region centered at around 750 nm was observed (Fig. 2f), which is associated with the ¹D₂ → ³F₃ transition³⁴. This reveals that the population of the ¹D₂ state can also be efficiently depleted

to the 3F_3 state via stimulated emission with laser irradiation of proper wavelength, analogous to our proposed stimulated emission pathway by an 810 nm laser associated with the $^1D_2 \rightarrow ^3F_2$ transition (Supplementary Fig. 13). The difference in the depletion efficiency at these two depletion peaks could be due to the different emission cross-sections of the $^1D_2 \rightarrow ^3F_2$ and $^1D_2 \rightarrow ^3F_3$ transitions. Notable 455 nm emission enhancement was induced by addition of a second beam approaching 700 nm (Fig. 2f), which could be caused by the action of the $^3H_6 \rightarrow ^3F_3$ ground state absorption process that increases the population of the 3F_3 state, in favor of multiphoton upconversion of Tm^{3+} ions (Supplementary Fig. 13). In addition, the 455 nm emission enhancement observed under the co-irradiation of a ~ 765 nm laser beam and the 975 nm excitation laser (Fig. 2f) could be due to the matching of the former with the $^3H_5 \rightarrow ^1G_4$ excited state absorption process^{34, 53, 58}, facilitating multiphoton upconversion luminescence by increasing the population of the 1G_4 state (Supplementary Fig. 13).”.

(3) Some additional comments on the FTIR characterizations: the authors reported that “OA-UCNPs exhibit a broadband at about 3433 cm^{-1} and a weak band at 1738 cm^{-1} , corresponding to the stretching vibration of O-H and C=O (-COOH), respectively, which suggests the presence of trace amount of oleic acid on the surfaces of nanoparticles [5, 24].” Actually, in the Ref 24, Liu et al reported “the weak stretching mode of the -COOH group at 1709 cm^{-1} suggests the presence of trace amounts of free oleic acid on the surface of nanocrystals before ligand exchange. This band is significantly enhanced and shifted to 1732 cm^{-1} after ligand exchange by PAA, suggesting an increase in the quantity of the -COOH groups on the particle surface.” This means the 1732 cm^{-1} peak belongs to PAA modified UCNPs, so why a weak band at 1738 cm^{-1} suggests the presence of trace amount of oleic acid? I suggest the authors should focus on the main peak shift after the PAA modification, which is common in the references. In this work, the main peak around 1560 cm^{-1} should belong to the COO- of OA- on the surface of OA-UCNPs, while this peak shift to 1647 cm^{-1} after the PAA modification.

Response: We thank the reviewer very much for his/her comments and advice. As suggested, we reanalyzed the experimental FTIR data and modified its description

In principle, most of the oleic acid molecules are chemically attached on the nanoparticles’ surface through the coordination (no C=O) between $-COO^-$ and RE^{3+} [6, 7], which extinguishes the band of C=O at 1738 cm^{-1} . On the surface of our prepared OA-UCNPs, there were possibly a few free oleic acid molecules, since a weak band at 1738 cm^{-1} corresponding to the stretching vibrations of C=O of free oleic acid (-COOH) was observed. And thus, in the original SI we thought it would suggest the presence of free oleic acid, which disappeared after PAA modification. To avoid any misunderstanding, we have deleted the sentences in the revised SI.

The FTIR spectrum of pure PAA was added in the revised SI to better demonstrate the successful functionalization of PAA on the UCNPs' surface, as shown in Fig. 3. The bands at 1552 and 1460 cm^{-1} are associated with the asymmetric (ν_{as}) and symmetric (ν_{s}) stretching vibration of $-\text{COO}^-$ groups, as shown in Fig. 3. After reaction with PAA, the band at 1552 cm^{-1} corresponding to asymmetric stretching vibration of carboxylate anions became inconspicuous, and at the same time, the band of PAA at 1671 cm^{-1} was shifted to 1647 cm^{-1} . These suggest the successful PAA modification on the surface of nanoparticles.

Figure 3 (the revised Supplementary Figure 17 (a) in the SI) Fourier transform infrared spectroscopy (FTIR) spectra of (i) OA-UCNPs, (ii) PAA-UCNPs and (iii) pure PAA.

Change in the manuscript:

Modified figure caption in the revised SI (Supplementary Figure 16): "...The bands at 1552 and 1460 cm^{-1} are associated with the asymmetric (ν_{as}) and symmetric (ν_{s}) stretching vibration of $-\text{COO}^-$ groups, since the oleic acid molecules are chemically attached on the UCNP surface through the coordination between the $-\text{COO}^-$ group and the RE^{3+} ions [3, 5]. After reaction with PAA, the band at 1552 cm^{-1} corresponding to asymmetric stretching vibration of carboxylate anions became inconspicuous, and the band of PAA measured at 1671 cm^{-1} was shifted to 1647 cm^{-1} , suggesting successful PAA modification on the surfaces of nanoparticles [29, 30] ..."

(Line 483-490, Page 31, SI)

Reviewer #4 (Remarks to the Author):

In the revised version, the authors properly addressed most of my recommendations. Now the article more precisely cites previous research, and adequately compares its results with common STED-microscopy performance and the work of reviewer #2. I do not question the differences in the experimental results of the authors work and the work of reviewer #2 due to different experimental conditions. The models used by both groups to describe the depletion mechanisms are also different, which does not seem questionable to me at this stage. Taking into account novelty of the demonstration of two-color super-resolution imaging and super-resolution cellular structure imaging using UCNPs the article is worth publishing in Nature Communications.

Response: We really appreciate the reviewer for his/her recommendation for publication of our revised manuscript and all his/her previous comments, which have helped us improve our work significantly.

References

- [1] J.B. Gruber, W.F. Krupke, J.M. Poindexter, *Journal of Chemical Physics*, 41 (1964) 3363-3377.
- [2] V.A. Antonov, P.A. Arsenev, K.E. Bienert, A.V. Potemkin, *Physica Status Solidi*, 19 (1973) 289-299.
- [3] Y. Liu, Y. Lu, X. Yang, X. Zheng, S. Wen, F. Wang, X. Vidal, J. Zhao, D. Liu, Z. Zhou, C. Ma, J. Zhou, J.A. Piper, P. Xi, D. Jin, *Nature*, 543 (2017) 229-233.
- [4] J. Zhao, D. Jin, E.P. Schartner, Y. Lu, Y. Liu, A.V. Zvyagin, L. Zhang, J.M. Dawes, P. Xi, J.A. Piper, *Nature Nanotechnology*, 8 (2013) 729.
- [5] S.E. Ivanova, A.M. Tkachuk, A. Mirzaeva, F. Pellé, *Optics & Spectroscopy*, 105 (2008) 228-241.
- [6] N. Bogdan, F. Vetrone, G.A. Ozin, J.A. Capobianco, *Nano Letters*, 11 (2011) 835.
- [7] T. Zhang, J. Ge, A. Yongxing Hu, Y. Yin, *Nano Letters*, 7 (2007) 3203.

REVIEWERS' COMMENTS:

Reviewer #2 (Remarks to the Author):

The authors followed my recent suggestion and conducted "pump-probe" experiment by fixing 975 nm excitation and scanning the depletion laser from 700 nm to 815 nm. This new result is interesting, and definitely adds additional value to this work.

This new depletion spectrum in fact exactly shows the efficient depletion over the rather broad band of from 780 nm to over 815 nm band ($3H6 \rightarrow 3H4$ transition), while the depletion efficiency at 1D2 level is rather weak (but good to know).

Again according to the small energy gap between 3F2 and 3F3, I would rather believe the 740 nm band emission belongs to the mix of the $1D2 \rightarrow 3F2$ transition and the $1D2 \rightarrow 3F3$ transition, the stretch to 810 nm is too large to be true. The mechanism described by this work could be an additional pathway for the depletion of 455 nm emissions at 740 nm ($1D2 \rightarrow 3F2/3F3$), but much weaker compared with 800 nm band depletion ($3H6 \rightarrow 3H4$ transition, centred at around 800 nm, not 780 nm).

The data presented now is clear enough, I leave this interesting discussion open for the community to judge. As I suggested earlier, due to many other merits of this work, I now suggest this work should be accepted for a timing publication in Nature Communications.

Manuscript No. NCOMMS-17-01804C

Title: Achieving high-efficiency emission depletion nanoscopy by employing cross relaxation in upconversion nanoparticles

Response Letter to Reviewer

Dear Reviewer,

Many thanks for your recommendation for publication of our manuscript and your open attitude on the discussion as well as your further valuable comments on our revised manuscript. In the following, we provide a point-by-point response to the comments, together with the corresponding changes in the manuscript. As below, the reviewer's comments are written in **black** and our responses to them in **blue**. The important amendments or changes to the manuscript are given after the response in **red**.

Reviewer's comments:

Reviewer #2 (Remarks to the Author):

(1) The authors followed my recent suggestion and conducted “pump-probe” experiment by fixing 975 nm excitation and scanning the depletion laser from 700 nm to 815 nm. This new result is interesting, and definitely adds additional value to this work.

This new depletion spectrum in fact exactly shows the efficient depletion over the rather broad band of from 780 nm to over 815 nm band ($^3\text{H}_6 \rightarrow ^3\text{H}_4$ transition), while the depletion efficiency at $^1\text{D}_2$ level is rather weak (but good to know).

Response: We thank the reviewer for his/her comments.

As we discussed in our previous Response Letter (2nd revision), the net effect of the 810 nm STED beam on the transitions between the $^3\text{H}_6$ and $^3\text{H}_4$ states is light absorption, which increase the population density of the $^3\text{H}_4$ state, rather than stimulated emission. In fact, this net effect of light absorption degrades the depletion efficiency to a large extent. This argument is well supported by previously presented experimental results, including the observed enhancement of the 1470 nm band ($^3\text{H}_4 \rightarrow ^3\text{F}_4$) (Figure 2d in the manuscript), and that the 475 nm emission ($^1\text{G}_4 \rightarrow ^3\text{H}_6$) was enhanced in the 5% Tm^{3+} sample while the 455 nm emission was depleted (75% off) (Figure RR7 in the 2nd revision Response Letter).

In addition, the previously presented results (2nd revision Response Letter) of our studies on the 455 nm emission depletion, by fixing the intensity of the 810 nm depletion laser (14.2 MW cm^{-2}) while adjusting the intensity of the 975 nm excitation in a large range (175 kW cm^{-2} - 17.5 MW cm^{-2}), indicate particularly that the two-photon excited population inversion (the $^3\text{H}_4$ excited state relative to the ground state $^3\text{H}_6$) did not occur in our study and the mechanism

reported in Ref. [1] was not active in our system.

As discussed in our manuscript (version NCOMMS-17-01804C) and previous Response Letter (3rd revision), the depletion region centered at around 750 nm in the depletion spectrum can be associated with the $^1D_2 \rightarrow ^3F_3$ transition and the depletion region from 780 nm to over 815 nm can be associated with the $^1D_2 \rightarrow ^3F_2$ transition. This broad depletion region associated with the $^1D_2 \rightarrow ^3F_2$ transition is in consistence with many previous reports [2-5], indicating this transition can have a rather broad emission band (758-860 nm) due to various spectrum broadening effects [2], referring to detailed analysis in our previous Response Letter (2nd revision). The depletion efficiency at 1D_2 level is very efficient, since it induces 96% emission off (Fig. 1 in the manuscript) even after compensating the synergistic enhancement effect of the STED beam. As we discussed in our manuscript, the depletion efficiency at these two depletion regions could be due to the different emission cross-sections of the $^1D_2 \rightarrow ^3F_2$ and $^1D_2 \rightarrow ^3F_3$ transitions, supported by previous reports stating that the $^1D_2 \rightarrow ^3F_2$ transition has a larger branching ratio than the $^1D_2 \rightarrow ^3F_3$ transition [6-9]. In addition, laser light at around 750 nm could have a larger synergistic enhancement effect (cooperating with the 975 nm excitation light) on the 455 nm emission than the 810 nm light, due to its perfect matching with, e.g., the $^3H_5 \rightarrow ^1G_4$ transition [2-4], which would also degrade the depletion efficiency at this wavelength (supplementary Fig. 13).

In order to make these more clearer, we have accordingly added several sentences in the manuscript, see below.

Change in the manuscript:

Sentences added (page 11): “This broad depletion region is in consistence with the $^1D_2 \rightarrow ^3F_2$ emission spectrum^{49, 54, 55, 56}, indicating this transition can have a rather broad emission band in various host materials due to spectrum broadening effects.”

Sentences modified/added (page 12): “In addition, laser light at around 750 nm could have a larger synergistic enhancement effect (cooperating with the 975 nm excitation light) on the 455 nm emission than the 810 nm light, due to its matching with other energy gaps, for example, the $^3H_5 \rightarrow ^1G_4$ transition^{49, 54, 55}, which would also degrade the depletion efficiency at this wavelength (supplementary Fig. 13).”

(2) Again, according to the small energy gap between 3F_2 and 3F_3 , I would rather believe the 740 nm band emission belongs to the mix of the $^1D_2 \rightarrow ^3F_2$ transition and the $^1D_2 \rightarrow ^3F_3$ transition, and the stretch to 810 nm is too large to be true. The mechanism described by this work could be an additional pathway for the depletion of 455 nm emissions at 740 nm ($^1D_2 \rightarrow ^3F_2/^3F_3$), but much weaker compared with 800 band depletion ($^3H_6 \rightarrow ^3H_4$ transition, centered at around 800 nm, not 780 nm).

Response: We thank the reviewer for his/her comments.

As discussed in our previous Response Letter (2nd revision) based on an intensive literature summary and analysis, we insist that the $^1D_2 \rightarrow ^3F_2$ transition generates an emission band significantly longer than 740 nm due to various spectrum broadening effects, readily approaching and exceeding 810 nm (758-860 nm) [2-5], overlapping with the $^3H_4 \rightarrow ^3H_6$ emission transition.

The difference in the depletion efficiency in the two depletion pathways could be mainly due to two reasons, referring to our response to Comment (1).

As discussed in our response to Comment (1), the net effect of the 810 nm STED beam on the transitions between the 3H_4 and 3H_6 states is light absorption, which increases the population density of the 3H_4 state, rather than stimulated emission.

As discussed in our manuscript, the $^1D_2 \xrightarrow{810\text{ nm}} ^3F_2$ stimulated emission process together with the $^3H_4 + ^3H_6 \rightarrow ^3F_4 + ^3F_4$ cross-relaxation process are the mechanism for the depletion of 455 nm emissions in our system, taking into account all the spectroscopic results.

In addition, the $^3H_6 \rightarrow ^3H_4$ transition (absorption) is centered around 780 nm and the $^3H_4 \rightarrow ^3H_6$ transition (emission) is centered around 800 nm. There is a blue shift of the absorption band relative to the emission band.

(3) The data presented now is clear enough, I leave this interesting discussion open for the community to judge. As I suggested earlier, due to many other merits of this work, I now suggest this work should be accepted for a timing publication in Nature Communications.

Response: We appreciate the reviewer's open attitude on the discussion and his/her recommendation for publication of our manuscript.

References

- [1] Y. Liu, Y. Lu, X. Yang, X. Zheng, S. Wen, F. Wang, X. Vidal, J. Zhao, D. Liu, Z. Zhou, C. Ma, J. Zhou, J.A. Piper, P. Xi, D. Jin, Nature, 543 (2017) 229-233.
- [2] S.E. Ivanova, A.M. Tkachuk, A. Mirzaeva, F. Pelle, Optics and Spectroscopy, 105 (2008) 228-241.
- [3] J.B. Gruber, W.F. Krupke, J.M. Poindexter, The Journal of Chemical Physics, 41 (1964) 3363-3377.
- [4] J.B. Gruber, R.P. Leavitt, C.A. Morrison, The Journal of Chemical Physics, 74 (1981) 2705-2709.
- [5] R.W. Schwartz, T.R. Faulkner, F.S. Richardson, Molecular Physics, 38 (1979) 1767-1780.
- [6] P. Villanueva-Delgado, D. Biner, K.W. Krämer, Journal of Luminescence, 189 (2016) 84-90.
- [7] B.M. Walsh, N.P. Barnes, Applied Physics B, 78 (2004) 325-333.
- [8] I.R. Martín, V.D. Rodríguez, Y. Guyot, S. Guy, G. Boulon, M.F. Joubert, Journal of Physics: Condensed Matter, 12 (2000) 1507-1507.
- [9] M. Dulick, G.E. Faulkner, N.J. Cockroft, D.C. Nguyen, Journal of Luminescence, 48 (1991) 517-521.